

# Robotic observations of high wintertime carbon export in California coastal waters

J. K. B. Bishop[1,2], M. B. Fong[3], T. J. Wood[2]

[1]Department of Earth and Planetary Science, UC Berkeley, Berkeley, CA 94720, USA.
[2]Earth Sciences Division, Lawrence Berkeley National Laboratory, Berkeley, CA, 94720, USA
[3]Department of Chemistry, Scripps Institution of Oceanography, La Jolla, CA, USA.

*Correspondence to*: J. K. B. Bishop (jkbishop@berkeley.edu)

**Abstract.**

Biologically mediated particulate organic and inorganic carbon (POC and PIC) export from surface waters is the
principal determinant of the vertical oceanic distribution of pH and dissolved inorganic carbon and thus sets the
conditions for air sea exchange of $CO_2$; exported organic matter also provides the energy fuelling communities in the
mesopelagic zone. However, observations are temporally and spatially sparse. Here we report first hourly-resolved
optically-quantified POC and PIC sedimentation rate time series from autonomous Lagrangian Carbon Flux
Explorers (CFEs), which monitor particle flux using imaging at depths below 140 m in the Santa Cruz Basin, CA in
May 2012, and in January and March 2013. Highest POC vertical flux (~100-240 mmol C $m^{-2}$ $d^{-1}$) occurred in
January, when most settling material was mm to cm-sized aggregates, but when surface biomass was low; fluxes
were ~18 and 6 mmol C $m^{-2}$ $d^{-1}$, respectively in March and May, under high surface biomass conditions. An
unexpected discovery was that January 2013 fluxes measured by CFE were 20 times higher than simultaneously
deployed surface-tethered sediment traps and which multiple lines of evidence indicate strong under sampling of
aggregates larger than 1 mm. Furthermore, the Jan 2013 CFE fluxes were about 10 times higher compared to highest
previous nearby multi year sediment trap observations. The strength of carbon export in biologically dynamic
California coastal waters is likely underestimated by a factor of between 3 and 20.

**Key words.** Biological Carbon Pump, Ocean Carbon Cycle, Twilight Zone Export, Coastal Sedimentation, Optical
Sedimentation Recorder, Carbon Flux Explorer.

## 1. Introduction

Phytoplankton account for half of global net photosynthesis (Field et al., 1988), or about 50 Pg C $y^{-1}$, yet they live for
a week before being consumed by grazers. While most phytoplankton carbon is recycled in the surface layer, recent
model and observation based estimates (Henson *et al*., 2011; Yao and Schlitzer 2013; Siegel *et al*., 2014) suggest that
globally 5 to 12 Pg C $y^{-1}$ is exported below the euphotic zone as sinking particulate organic and inorganic carbon
(POC and PIC) in fecal pellets, amorphous large aggregates, and as independently sinking carapaces and calcareous
shells (e.g. Bishop, Ketten, and Edmond, 1978; Alldredge and Silver 1988; Turner, 2015). Many large amorphous
aggregate particles (e.g. greater in size than 0.5 mm) would also be classified as marine snow by virtue of their being
visible to the naked eye or camera in reflected light; such aggregates have been shown to originate as products of




feeding (e.g. ejected boluses of fecal material, or discarded mucus feeding webs, or appendicularian houses) or from turbulence driven aggregation of biogenic organic and inorganic particles (Alldredge and Silver 1988). Important to POC settling is particulate inorganic carbon (i.e. PIC), in the form of calcite foraminifera shells and coccoliths plates and aragonite pteropod shells and shell fragments. Foraminifera shells preserved in sediments in particular are key to

unravelling past climate and ocean chemistry, yet little of their dynamics in the water column has been understood (Be *et al.*, 1985). Together with siliceous diatom frustrules, biogenic carbonates are a major factor governing the excess density, and hence sinking rate, of aggregate particles carrying POC downward from the surface layer (Armstrong et al., 2002). As much of exported organic matter is consumed in the water column or at the sediment sea water interface, little POC is preserved in sediments. The varied processes of production, grazing, recycling, export,

and subsurface remineralisation constitute the ocean's biological carbon pump (Volk and Hoffert, 1985).

Direct observations of POC and PIC sedimentation fluxes in the upper thousand meters of the ocean are sparse, of short duration (days), and with few exceptions, are mostly during summertime. This is because all observations to date have required ships to be present or near by (e.g. Martin *et al.,* 1987; Buesseler *et al.,* 2000; Stanley *et al.,* 2004; Lam and Bishop, 2007; Lampitt *et al.,* 2008; Lee *et al.*, 2009; Owens *et al.,* 2013).

Such direct observations are often used to calibrate models used to estimate global carbon export (e.g. Henson *et al.*, 2011, Siegel *et al.*, 2014). Henson *et al.* (2011) estimated an export of 5 Pg C y$^{-1}$, based on the synthesis of a decade of ship observations of the ratio of dissolved radioactive $^{234}$Th to parent $^{238}$U, $^{234}$Th/POC ratios measured in shallow sediment trap or pump sampled particulates, and satellite-based estimates of primary productivity. Siegel *et al.* (2014) estimated a global carbon export of 6 Pg C y$^{-1}$ using the combination of food web models – calibrated with

ship-board observations of production, grazing, and sedimentation -- and global satellite retrievals of size-dependent phytoplankton biomass. In contrast, Yao and Schlitzer (2013) estimated an export of 12 Pg C y$^{-1}$ by inversion, within an ocean circulation model, of ocean nutrient fields averaged over five to six decades. It is unknown whether the factor of two difference in export numbers are a reflection of a strong recent weakening biological carbon pump, or of differences in methodology. Either way, there is major uncertainty in both the strength and stability of the ocean's

biological carbon pump and of consequent feedbacks to atmospheric CO$_2$ trends; furthermore, there remains major uncertainty as to the magnitude of the energy flow carried by sinking particles to sustain mesopelagic communities (Burd, *et al.* 2010; Banse, 2013).

Eppley and Peterson (1979; E&P) pointed out the importance of near-shore waters, defined as coastal waters deeper than 200 m, to the ocean carbon cycle. In their estimate, near-shore waters account for over 40% of global new

production and hence particle export. While there have been arguments that this number may be somewhat high in the modern context (Henson *et al.*, 2011), all recent calculations of global export (although not broken down as done as in E&P 1979) indicate a lower contribution from coastal waters.

In this paper, we describe direct in-situ observations of carbon export in biologically dynamic ocean waters near the California coast obtained using a fully autonomous ocean profiling Carbon Flux Explorer (CFE; Fig. 1A) we have

developed. The Carbon Flux Explorer is designed to perform sustained high-frequency observations of POC and PIC sedimentation within the upper kilometer (or twilight zone) of the ocean for seasons to years and to operate in an observational context not dependent on ships. The CFE melds the concept of current-following, sample-collecting





neutrally-buoyant sediment traps (Buesseler *et al.,* 2000; Stanley *et al.*, 2004, Lampitt et al., 2008, Owens *et al.*, 2013) with photographic imaging of the particles as they are deposited in a sediment trap (Asper, 1987). The current-following approach aims to avoid the hydrodynamic biases suffered by surface tethered sediment traps (Gardner, 2000; Buesseler *et al.,* 2007). With imaging, the CFE is able to gain detail of the biological process of sedimentation,

thus it extends significantly beyond the simple Carbon Flux Index (CFI) optical sedimentation proxy from transmissometer-equipped Lagrangian floats (Bishop et al. 2004, 2009; Estapa et al., 2013).

Below we present observations from CFE development testing during three expeditions off the coast of southern California aboard R/V New Horizon: May 29-June 4 2012 (NH1204), Jan. 18-21 2013 (NH1301) and March 27-31 2013 (NH1304). These observations, like past ship-tended studies using neutrally buoyant and surface-tethered

sediment traps, are of relatively short duration (days); however, they are the first carried out in a productive coastal environment. CFE data are analysed below in the framework of remotely sensed surface chlorophyll and POC, *in-situ* water column optical properties, and subsurface currents. The study site is a 150 km$^2$ region centred at 33° 43'N 119° 33'W in the 1900 m deep Santa Cruz Basin (SCB) near the California coast (Figs. 1B & C).

## 2. Methods

### 2.1 Carbon Flux Explorer (CFE).

The CFE is comprised of an Optical Sedimentation Recorder (OSR) interfaced to a CTD-equipped Sounding Oceanographic Lagrangian Observer (SOLO) float (Davis, Sherman and Dufor, 2001). The CFE dives repeatedly (daily in this study) below the surface to obtain OSR observations at three target depths. The CFE's Optical Sedimentation Recorder (OSR; Figure A1, Appendix Sec. 1.1) awakes when the CFE reaches a target depth. Particles

settle through a hexagonal celled baffle (1 cm opening, 5 cm length) into a high-aspect titanium funnel and settling column before depositing on a glass sample stage. Particles are imaged in three lighting modes: transmitted, transmitted-cross-polarized, and dark field. On first wake up of a given dive, the sample stage is flushed with water and images of the particle-free stage are obtained. At timed intervals (~20 min in data described here) the OSR repeats image sets, which register the sequential build-up of particles. After the pre-determined number of image sets

over ~1.8 hours, cleaning occurs and a new reference image set is obtained. After ~5 hours at a target depth the OSR performs a final image/cleaning cycle and reference image set, and the SOLO surfaces to report GPS position, CTD profile data, and OSR data, and dives to its next target depth. Every 3 dive cycles, the depth order is repeated; in the current study, target depths were 150, 300 and 500m. A ballasting error in May 2012 led to the CFE operating considerably deeper than planned. In the present form of the CFE, all images are stored aboard the OSR for post

recovery detailed analysis.

### 2.2 OSR Image data reduction.

Images were taken under transmitted, transmitted-cross-polarized, and dark field illumination, and were processed to yield three parameters that characterize particle abundance, according to the scheme depicted in Fig. 2. The parameters are particle attenuance, cross-polarized photon yield, and dark field counts. Attenuance is the best



measure of particle loading as light is reduced exponentially as it propagates through the sample. Cross-polarized photon yield takes advantage of the birefrigent property of calcium carbonate, and is a measure of particulate inorganic carbon. Dark field counts reveal the colour of particles. Details of the treatment of transmitted light and transmitted cross-polarized light images are covered in sections 2.2.1 and 2.2.2, respectively. Dark-field images are

normalized by illumination intensity and then transformed to a consistently scaled reflectance in counts above background. A full-resolution post-processing image of an aggregate particle from January 2013 is shown for all modes of illumination in Figure 3.

### 2.2.1 Attenuance (ATN).

Transmitted light (TRA) images are normalized by an in-situ particle free image of the sample stage area under

transmitted light illumination. Image attenuance (ATN), the sum of both light scattering loss and particle absorption of the primary beam, is calculated as the negative $\log_{10}$ of transmittance and is a measure of both aggregate thickness and transparency. The effects of overlaying particles are additive in attenuance units. We calculate attenuance in $\log_{10}$ units rather than in natural log units as order-of-magnitude changes of transmission are easy to infer.

In order to detect particles, we set the lowest threshold of pixel attenuance to be 0.02. The attenuance of all pixels

identified as particles is integrated across sample stage area and then divided by the total number of pixels of the sample stage area to yield the average particle load of the sample stage (in attenuance units). For convenience attenuance is multipled by 1000 and reported in units of milli-attenuance (mATN). The stage load, when multiplied by sample stage area yields units of mATN-cm$^2$. Because the light source is directly above the sample stage, the stage load (mATN-cm$^2$ ) is an optical 'volume' of material on the sample stage. Attenuance flux (in units of mATN-

cm$^2$ cm$^{-2}$ d$^{-1}$) is calculated by taking differences of successive stage loads, divided by the mouth opening of the trap, and further divided by the time (in days) between image sets. Attenuance is never saturated in our images (e.g. transmission is always greater than 0).

### 2.2.2 Cross-Polarized Photon Yield (POL).

Transmitted, cross-polarized light images are processed in three steps: compensation for the attenuation effects of

particles, subtraction of the particle-free blank and normalization for light source intensity. Due to the isolation of sub-stage polarizer in the pressure case of the OSR, hydrostatically induced stress on the glass pressure case window at depth induces an interference seen by the camera; thus, absorbing particles appear dark in the uncorrected image (Fig. 2B – RAW POL Image). The absorption effect is compensated by division of observed image counts by a polarization compensation factor (PCF), an empirically derived function of attenuance: PCF = 1/(2.1 * ATN). This

stress polarization interference has been eliminated in later builds of the CFE by moving the sub-stage polarizer outside of the pressure case.

Following correction, image counts at each pixel (for particles identified using the 0.02 attenuance threshold) are integrated across all particle pixels and normalized by the total number of pixels covering the stage area. Sample counts are normalized to lamp brightness and scaled by the difference in exposure time for POL vs. TRA images to



yield the quantity cross-polarized photon yield, expressed as a fraction of incident beam intensity. Results are scaled up by $10^6$ and data are reported in ppm. This optical measure of flux is in units of ppm-cm$^2$ cm$^{-2}$ d$^{-1}$.

### 2.2.3 Interference by Swimmers.

In the PIC flux time series, there were several instances when a calcified swimming organism was detected. This only affected data were from expedition NH1304, and Cypriod barnacle larvae were the predominant interfering organism. When the appearance and disappearance of these organisms in the image series led to a strong positive then negative flux, the interfering flux estimate was eliminated from the 1.8 hour average. In one other case during NH1304, a calcified amphipod carapace (Fig. 4-B) was removed from the 1.8 hour average PIC flux. This carapace was clearly part of the flux but was excluded in our consideration of aggregate sedimentation. Excluded points in Fig. 5B (below) are indicated by circled (+) symbols.

### 2.2.4 POC and PIC flux proxy.

We hypothesize that attenuance is the best proxy for the POC in particles by analogy to the finding that transmissometer measured particle beam attenuation coefficient ($c_P$) is highly correlated to POC concentration (e.g., Bishop and Wood, 2008). We note that particle beam attenuation coefficient is expressed in units of m$^{-1}$ while our attenuance values are dimensionless. This is because in transmissometry, beam attenuation occurs continuously along the seawater path that the light beam passes through. In our case, the path length from light to sample stage is unimportant as all attenunance of light occurs in the thin layer of particles on the sample stage. Our use of the image of the light as a transmission reference removes the minor effects of light loss through the water path.

Cross-polarized photon yield is a measure of PIC, which is comprised of biogenic calcite and aragonite polymorphs of calcium carbonate. The photon yield under cross-polarized illumination (or birefringence) of carbonates has been shown to scale linearly PIC concentration; furthermore, biogenic carbonates both dominate all mineral material in the water column and have 10 times greater photon yield compared with other birefringent minerals (Guay and Bishop, 2002).

### 2.3 Conversion of POC$_{ATN}$ flux to POC flux.

Calibration samples from the BUOY-OSR system were not useful due to a 20-fold lower particle collection rate compared to that of the CFE (described below in Sec. 3.3). We thus estimate the amount of POC in our images in order to translate attenuance flux to POC flux.

Bishop et al. (1978) report information on the dry weight density and geometric properties of particles of similar morphology as sampled here. Particularly, we use particle dry weight density of 0.087 g cm$^{-3}$, and the Bishop et al. (1978) equation, which estimates aggregate thickness (h, in cm) as a function of equivalent circular diameter (d, in cm):

$$h = 0.052d + 0.0045. \tag{1}$$

We analysed a series of 5 images collected on January 20 2013 from 11:11:47 to 12:39:47 UTC (just before cleaning). The last image of this set had 65% of the sample stage covered with particles. These images are provided in supplemental material for readers to independently analyse. The software package ImageJ (National Institutes of Health, USA – http://rsb.info.nih.gov/ij) was used to measure the equivalent circular diameter of aggregates. As aggregates overlapped in successive images due to the



unexpectedly high sedimentation observed, successive images were stepwise subtracted in pairs prior to size analysis. Aggregate volume was calculated as the product of cross sectional area and h.

A total of 127 aggregates from 500 μm to 8.3 mm in size were enumerated in all five images. A size cutoff of 800 μm was used to minimize contamination of the analysis due to the slight expansion of aggregate dimension over time as they rested on the

stage. Although, size distributions were initially quantified to a lower limit of 20 μm, it is important to note that 97% of the volume attenuance of the sample was accounted for by particles >800 μm in this sample. For the remaining 63 particles larger than 800 μm, we calculated an aggregate volume of 0.113 cm$^3$. To estimate the POC flux, we assume an aggregate dry weight density of 0.087 g cm$^3$ (e.g. From Fig. 22 in Bishop et al. 1978) and thus calculate the weight of aggregate matter as 0.0098 g. Organic matter at depths between 100 m and 500 m is typically 60% (range 50-70%) of dry weight (Fig. 11 in Bishop et al.

1978). Thus, organic matter weight is 0.0059g. We estimate a POC (as carbon) by dividing this mass by the OM:C conversion factor, 1.88 from Hedges et al. (2002). We divide by 12.011, the atomic weight of carbon, to yield a POC loading of 0.26x10$^{-3}$ moles. The opening of the OSR funnel has a diameter of 15.4 cm, which yields an assumed trap collection area of 186.3 cm$^2$. The time interval for collection of this sample was 1.84 hours, or 0.0766 days. Combining this information yields a carbon flux of 183 mmol m$^{-2}$ d$^{-1}$. The average POC$_{ATN}$ flux for all particles >13 μm in size during the same time interval was 66.2 mATN-

cm$^2$ cm$^{-2}$ d$^{-1}$. As mentioned above, over 97% of cumulative sample attenuance was in the >800 μm size fraction. The conversion factor for POC$_{ATN}$ flux to POC flux is the ratio of 183 to 66.2*0.97 = 2.8. This conversion factor allows estimation of POC fluxes observed for compare with other data from near by waters.

We stress that the factor 2.8, used for conversion of POC$_{ATN}$ flux to POC flux is approximate. Equation (1) for aggregate thickness is the only published study that we are aware of. The dry weight density factor (0.087 g cm$^{-3}$) was taken for aggregates

that were similarly optically dense as observed in this sample. Bishop et al. (1978) also reported dry density values as high as 0.24 g cm$^{-3}$. The organic matter percentage for large particulates used in the calculation above, is typical of values obtained by our group using in-situ filtration. We note that Collins et al. (2011) report upper water column POC flux and mass flux data from surface tethered sediment traps in the San Pedro Basin, closer to shore, which yield values closer to 30% with the rest being contributed by inorganic terrigenous and biogenic phases. This would require our estimate of dry weight particle density to be

proportionally adjusted upwards to account for the greater fraction of inorganic ballasting material – and thus compensate for the reduced organic percentage. A factor of two adjustment of the POC:POC$_{ATN}$ ratio either way will not change our conclusions regarding the high magnitude of the sedimentation rate observed in January 2013 by the CFE. Nor will it change the relative difference observed between CFE and BUOY-OSR presented below.

**2.4 Satellite data processing.**

Retrievals of chlorophyll and POC from observations made by the Moderate-Resolution Imaging Spectroradiometer (MODIS) on NASA's Aqua satellite are provided on the NASA DAAC. Level 2 Local Area Coverage (L2-LAC) data at 1 km resolution for the period of the three expeditions were downloaded on April 4 2013 and processed using SEADAS software. Chlorophyll and POC were retrieved for pixels within a 2 km search radius of 33.72°N 119.5°W and were selected using the standard ocean colour processing quality flags. Means and standard deviations were

calculated and data were further selected based on the requirement of 5 valid pixels. Imagery for January 2013 from 4 km gridded L3 products is included in supplemental materials.

Surface photosynthetically active radiation (PAR) estimated from MODIS Aqua data for Jan., Mar., and May expeditions was 25, 45, and 60 Einsteins m$^{-2}$ d$^{-1}$, respectively. Corresponding values for diffuse attenuation





coefficient for PAR ($k_{PAR}$) from beam attenuation coefficient profiles (Bishop and Wood, 2009) were 0.068, 0.20, and 0.18 m$^{-1}$. Euphotic zone depths corresponding to the 1% light level were 68, 23, and 26 m. The euphotic zone depths based on the 1 E m$^{-2}$ d$^{-1}$ isolume were 50, 20, and 25 m.

## 3. Results

### 3.1 Flux time-series, Hydrography, and Remote sensing data

Figure 5 shows the time series of POC$_{ATN}$ flux and PIC$_{POL}$ flux. In contrast to surface-tethered sediment traps (Martin et al., 1987), we found that the influence by/presence of swimming organisms was rarely significant in our image time series (see supplemental animations of CFE imagery). The highest POC$_{ATN}$ flux was measured in Jan. 2013, intermediate flux was observed in March 2013, and lowest flux was observed in May 2012. Similar temporal/depth systematics was observed for PIC$_{POL}$ flux (*Fig. 5B*); however, the relative ratio of PIC/POC flux was highest in March 2013.

The hydrographic context for our observations was established by CTD/transmissometer casts made to full water depth 3-4 times per day during each expedition. We found that mixed layer depths were mostly shallower than 10 m during all expeditions, reflective of the California coastal upwelling regime and net divergence of surface waters in the Santa Cruz Basin. We calculate properties of the upper 20 m, as these are most closely comparable to remotely sensed surface properties. Upper 20 m temperatures for January, March, and May averaged 12.6, 13.5, and 15 °C, respectively, reflecting seasonal warming. Over each cruise, averaged 0-20 m POC concentrations derived from transmissometer profiles (± s.d; Appendix A, Sec. A1.2 Fig. A2) were 4.1±0.7 (Jan.), 17.5±3.4 (March), 15.4± 5.7 µM (May). POC was most variable in May 2012, with concentrations rising from ~10 µM to 25 µM over the period of observations reflecting the progression of a plankton bloom southward from Point Conception. Corresponding euphotic zone depths (EZD) based on the 1 Einstein m$^{-2}$ d$^{-1}$ isolume (EZD for 1% light level in parentheses) were 50 (68), 20 (23), and 25 (26) m, respectively. January and March 2013 showed little temporal POC trend indicating a quasi steady state balance of photosynthesis and loss processes (Appendix A, Sec. 1.2 Fig. A2).

Figure 6 shows MODIS Aqua Chlorophyll and POC time series, averaged MODIS Chlorophyll and POC values for each cruise period and the previous week, averaged upper 20 m POC from CTD casts, and depth profiles of POC$_{ATN}$ and PIC$_{POL}$ flux. Satellite retrieved POC values agreed well with transmissometer estimates for March and May, and were lowest in January 2013, albeit approximately 2.5 times higher than transmissometer estimates. Imagery of the area in January the week prior to our cruise showed patchy (rather than organized structure) in chlorophyll distribution. During the time of the expedition, distributions were nearly spatially invariant (supplemental materials). From all perspectives, January 2013 had the lowest phytoplankton biomass but by far the highest fluxes of POC and PIC.

### 3.2 Particle Flux Profiles

Averaged POC$_{ATN}$ and PIC$_{POL}$ flux profiles provide systematic quantification of depth trends of sedimentation. In Jan. 2013, POC$_{ATN}$ fluxes at depths 150, 300 and 500 m were 67, 85, and 35 mATN-cm$^2$ cm$^{-2}$ d$^{-1}$, respectively; in




Mar. 2013 at similar depths, flux was between 6 to 7 mATN-cm$^2$ cm$^{-2}$ d$^{-1}$; values in May 2012 for the 450 – 900 m interval fell between 2 and 3 mATN-cm$^2$ cm$^{-2}$ d$^{-1}$ with no depth trend. PIC$_{POL}$ flux trends roughly followed those for POC$_{ATN}$ flux. PIC$_{POL}$ fluxes and at 150, 300 and 500 m depths were 96, 122, and 63 ppm- cm$^2$ cm$^{-2}$ d$^{-1}$ (Jan. 2013) and 33, 15, and 16 ppm-cm$^2$ cm$^{-2}$ d$^{-1}$ (Mar. 2013); The apparent decrease of PIC flux with depth from 2.8 to 0.5

ppm-cm$^2$ cm$^{-2}$ d$^{-1}$ in May 2012 may be an artefact of the low flux condition and high scatter of points.

For both POC$_{ATN}$ and PIC$_{POL}$ flux, it is also notable that the 1.8 hour averaged fluxes for each depth ranged over a factor of 10 relative to their average for the May and March periods, but ranged by less than a factor of two in the high flux period January 2013. This suggests that sedimentation is more uniform when large aggregates dominate.

The profile of PIC$_{POL}$/POC$_{ATN}$ ratio (Fig. 7) shows a depth dependent increase from 1.4 to 2.4 in January, indicating

preferential loss of POC with sinking. In March, PIC$_{POL}$/POC$_{ATN}$ ratios slightly decreased with depth from 4.6 to 3.7. In May 2012, PIC/POC ratio decreased from 1.5 to 0.3 between 430 and 900 m.  These observations confirm that the relative contribution of carbonates to aggregate ballasting varies with time.

### 3.3 Comparison of Surface Tethered OSR flux with CFE Flux

In January 2013, a sample-collecting OSR system, with the same OSR model as the one in the CFE, was deployed at

237 m tethered below a buoy system free-drifting at the surface (Appendix, Section 1.3, Fig. A3) to gain samples of particulates for comparison with the POC$_{ATN}$ and PIC$_{POL}$ data.

The ATN time series from CFE and BUOY-OSR are compared in Fig. 8. The BUOY-OSR yielded POC$_{ATN}$ and PIC$_{ATN}$ fluxes of 3.6 mATN-cm$^2$ cm$^{-2}$ d$^{-1}$ and 2.2 ppm-cm$^2$ cm$^{-2}$ d$^{-1}$, respectively – approximately 25 fold and 50 fold lower than CFE fluxes at 144 and 320 m depths. At the same time, BUOY-OSR and CFE images showed that the

two systems collected aggregates of completely different morphology and size distribution (Fig. 9). The CFE images showed abundant 5 mm- to cm-sized marine snow aggregates. The BUOY-OSR collected only fragments of these aggregates and few particles larger than 2 mm.

Figure 10 shows cumulative size distributions of volume attenuance corresponding to samples collected by the CFE and BUOY-OSR. Two cases are contrasted. First, we selected CFE images for depths 144 m (Fig 7), 320, and 507 m.

We also selected the BUOY-OSR image corresponding to Fig. 7 and another taken at the exact same time as the first CFE image. Other BUOY-OSR images were analysed for the rest of the time series. CFE images were either the first or second image after stage cleaning since subsequent images had multiple cases of particle overlap. BUOY-OSR images analysed were from just before stage cleaning since no particles were overlapping. Size distribution analysis was performed using ImageJ software, with an attenuance threshold >0.02, and a size threshold of at least two pixels

for a particle. Particles were classified according to their equivalent circular diameter (ECD), with the lowere size limit of 20 μm. Volume attenuance for each particle was computed (the product of particle area and attenuance). Results were summed cumulatively from largest to smallest particle size and then normalized by total volume attenuance (Table 1; Fig. 10). In January 2013, >96 % of cumulative volume attenuance (CVA) was found in the >1000 μm size range in CFE data at all depths. In contrast, BUOY-OSR results typically show less than half of the

cumulative volume attenuance for aggregates larger than 1000 μm, consistent with the particles being fragments of the larger aggregates and consistent with the factor of 20 difference in observed attenuance flux. If we were to



eliminate all aggregates larger than 1500 μm from January CFE data, then there is close agreement with the CVA – size-distributions derived from BUOY-OSR data. During the period when current flow across the BUOY-OSR fell below 2 cm s$^{-1}$, analysis of the BUOY-OSR image shows that nearly 50% of the aggregate volume attenuance was in the 2000-5000 μm, i.e. large, size range. Although, BUOY-OSR and CFE were operating at different depths, and at

slightly different locations in our study area (Fig. 1), there was no indication in CTD/optics profiles that there was any discontinuity or advective feature at 237 m that would lead to biasing the BUOY-OSR results low. Nor was there any indication in remote sensing imagery of near by fronts or eddy structures. Figure 11, showing CFE data from all three expeditions with January BUOY-OSR further demonstrates that CFE images always captured a greater loading of large particles than the BUOY-OSR in January. All lines of evidence indicate a major bias against aggregates

larger than millimetre in size by the surface-tethered BUOY-OSR.

### 3.4 POC flux estimates

In absence of calibration sample data, calculations based on particle geometry, thickness, dry weight density, and organic matter percentages (Sec. 2.3 based on Bishop *et al.*, 1978) yield a scaling factor of 2.8, which can be used to convert the optical $POC_{ATN}$ flux (mATN-cm$^2$ cm$^{-2}$ d$^{-1}$) to POC flux (mmol m$^{-2}$ d$^{-1}$). POC flux was thus estimated to

be 190, 240, and 100 mmol C m$^{-2}$ d$^{-1}$ in Jan. 2013 (at 144, 320 and 506 m), ~18 mmol C m$^{-2}$ d$^{-1}$ in Mar. 2013 (from 170 to 508 m), and ~6 mmol C m$^{-2}$ d$^{-1}$ in May 2012 (430 to 900 m). We similarly estimate ~10 mmol C m$^{-2}$ d$^{-1}$ at 237 m during the January 2013 BUOY-OSR deployment.

### 4. Discussion

#### 4.1 Surface Biomass and Export in Coastal Waters

The high January 2013 POC flux values of 190 and 240 mmol C m$^{-2}$ d$^{-1}$ to depths of 144 and 320 m are remarkable from several perspectives. Firstly, cases where the imaging stage was almost completely covered with material in 1.8 hours had been never before encountered in our previous deployments of CFEs for missions totalling over 100 days (mostly in summer time conditions). Secondly, these high particle accumulation rates were observed in the wintertime at a time when both remotely-sensed and in-water optical measures of particle stocks were low. Thirdly,

the magnitude of the January 2013 POC flux, estimated from CFE data, was approximately 8 times higher than the highest sedimentation rates measured using surface-tethered (at 100 and 200 m) and moored sediment traps (near 500 m) in the nearby Santa Barbara (Thunnel, 1998) and San Pedro Basins (Collins *et al.*, 2011). In both these studies, POC flux in moored trap samples collected over 4 years at a depth of 500 m were typically 3 mmol C m$^{-2}$ d$^{-1}$ with rare values as high as 15 mmol m$^{-2}$ d$^{-1}$. The much sparser surface-tethered trap observations (Collins *et al.*, 2011)

were no higher than 28 mmol C m$^{-2}$ d$^{-1}$. The finding of high sedimentation rate during wintertime low-biomass conditions in California coastal waters has not been reported in multi-year moored sediment trap observations.

The May 2012 fluxes of 6 mmol m$^{-2}$ d$^{-1}$ observed by CFE were the lowest of the study, at a time when surface waters had high particle concentrations, thus implying high recycling and shallow remineralisation. The finding of low




export beneath plankton blooms in coastal waters reinforces the findings in the open ocean that high surface biomass does not imply high export to depth (Bishop and Wood, 2009; Lam and Bishop 2007; Ebersbach et al., 2011).

The low surface biomass observed in January 2013 could indicate either a state of nutrient limitation of photosynthesis or a state of efficient transfer of primary produced organic matter to grazers and efficient particle

export through the upper 500 m. We rule out nutrient limitation in our case, as mixed layers consistently shallower than 10 m and surface temperatures of 12.5 °C indicated that upwelling conditions were prevalent. Furthermore, we observed intense activity of marine mammals and sea birds feeding on abundant macro invertebrates and fish during this period – which suggests that consumers were the source of the sinking aggregated material in the water column at this time. Many of the large aggregates seen in both Jan. and Mar. 2013 closely resemble abandoned

appendicularian houses described by Alldredge and Silver (1988). Animations of CFE imagery are provided in supplemental materials for further exploration. Physically driven aggregative processes were not likely in January and March 2013, as winds were below 5 m s$^{-1}$ and averaged 2.5 m s$^{-1}$. In May 2012, winds were below 10 m s$^{-1}$ and averaged 3 m s$^{-1}$. Surface waves and swell were almost completely absent in January and March. ADCP records showed currents were dominated by tidal fluctuations and not by eddy structures. The low biomass condition

observed in January is consistent therefore with biologically mediated export.

We note that 5-day running mean of satellite chlorophyll and POC levels appear to be nearly halved in a week (Fig. 4A,B) just before the study period in January. The rapid decrease suggests that active grazing and export were in progress.  Satellite imagery from Jan 2013 shows a patchy POC/chlorophyll distribution without obvious eddy structures or fronts near by.

Our observations show that POC$_{ATN}$ flux profiles (Fig. 6) attenuate with depth, albeit not following the classic Martin et al. (1987) curve fit ($\Phi z = \Phi_{100}/(Z/100)^b$, Z is depth and $\Phi$ is flux, and b = 0.858).  Using light based euphotic depths of 50, 20 and 25 m for January, March, and May as a reference for particle remineralisation (Buesseler and Boyd (2009), we derive Martin's 'b' factors as 0.36, 0.22, and 0.17, respectively, far lower than the accepted b value. In January 2013, the export may have been high enough to overwhelm the capacity of detritus feeders to reduce the

flux. In May and March, all evidence is consistent with high recycling of the sinking POC in waters shallower than our 400 and 125 m observation points, respectively.

In the case of PIC$_{POL}$ flux, we note that living foraminifera and empty foraminifera shells were predominantly associated with large aggregates in January. While the occurrence of empty shells in aggregates would not be exceptional, the finding of living foraminifera suggests an active association with these particles.

**4.1 Surface tethered BUOY-OSR vs. Lagrangian CFE fluxes.**

Our comparison of fluxes recorded by the CFE and a surface tethered BUOY-OSR in January 2013 found a factor of 20 difference in collection rate (Fig. 9), consistent with the loss of all aggregates >1500μm in size. CFE collection rates, ignoring the >1500 μm fraction in March 2013 and May 2012, would be approximately 4 times lower and 3 times lower, respectively (Fig. 11). This was surprising in light of reports that there are less than 'factor of two'

differences in sedimentation measured using surface tethered particle interceptor traps (Martin et al. 1987; PITS) and neutrally-buoyant sediment traps (NBST) deployed in oligotrophic waters near Bermuda (Buesseler *et al.,* 2000;





Stanley *et al.*, 2004, Owens *et al.*, 2013). In our study, the particles mostly absent from the BUOY-OSR were large aggregates in the millimetre to cm size class (Figs. 9, 10, 11). The only time when large aggregates appeared in the BUOY-OSR imagery was during periods when the flow of water across the mouth of the BUOY-OSR was below 2 cm s$^{-1}$ (Fig. 8). Given that marine snow aggregates typically settle at rates of 0.1-0.2 cm s$^{-1}$ (Alldredge and Silver,

1988), the trajectory of the aggregates across the BUOY-OSR funnel would be almost horizontal, even in currents as weak as 2 cm/s. The large aggregates are thus likely discriminated against entering the cm-sized trap baffle openings, which are of the same size as the particles and we hypothesize that the aggregates simply bounce back into the flow after shedding fragments, rather than enter the trap.

Further support for a high coastal sedimentation than is presently recognized is provided by Stukel et al. (2011) who

worked just offshore of our study area in the California Current. They compared the new production (Eppley and Peterson, 1979) "f-ratio" – the fraction of primary production supported by upwelled nitrate vs. the $^{234}$Th disequilibrium based "e-ratio" – the fraction of primary production exported as particles, and the two should be equivalent in a food web in equilibrium. While methods agreed 300 km offshore, the "f-ratio" exceeded the "e-ratio" by factors of 5-7 at locations within 50 and 100 km from the coast. Simply stated, there is insufficient time for the

$^{234}$Th (half life 24 days) deficit relative to its parent $^{238}$U to reach steady state in coastal waters. Subsequent surface-tethered Particle Interceptor Trap deployments (Stukel et al., 2013) in the same area showed consistently low exports in this region where f-ratios are typically high (Eppley and Peterson, 1979). Our location is closer to shore. Even taking into account factors such as food web and lateral transport processes, multiple lines of evidence evidence strongly indicates that large > 1 mm sized aggregates are severely under-sampled by surface-tethered baffled traps.

It is important to note that sediment traps have been the sole method to date for the assessment of carbon export in coastal waters. The single profile $^{234}$Th/$^{238}$U method is not applicable in the coastal environment since it relies on an assumption of a steady state deficit and requires a time scale of multiple half-lives (24.1 d) to reach this condition. Although the $^{234}$Th/$^{238}$U method may be used to infer export from time series sampling, such a study has not been undertaken in near-shore California waters to our knowledge. Also, e-ratio estimates are calibrated with large

particles sampled either using traps or pumps and the assumption is that the Th:C ratio of all large particle phases is the same. Both methods have been shown to suffer serious biases; in-situ pumps are discussed by Bishop, Lam and Wood (2012), and results presented here show a serious issue of under sampling of large aggregates by surface tethered baffled traps. As there is scant literature on the size dependence of Th-C in different large particle size classes in the coastal environment, it is thus impossible to know the accuracy of e-ratio export estimates. Coordinated

at sea studies are required to inter compare the various methods of measurement of carbon export.

### 5. Conclusions

The new continuous observations of particulate flux off the coast of Southern California by autonomous CFE's show high fluxes in winter when surface chlorophyll is low. The finding is consistent with high primary productivity,

efficient grazing, and high export of resulting aggregate material. The finding is also consistent with Eppley and



Peterson's (1979) estimate of high new production and their inferred strength of particle export in near-shore waters, and as noted above, is higher by a factor of ~8 than the highest observations made with surface tethered and moored sediment traps in waters close to our study site. Analysis of the images of accumulated particles suggest that surface-tethered sediment traps have under-sampled a major class of large aggregates contributing to carbon export from

surface waters.  At this writing, it is premature to revise the carbon export attributable to coastal waters up by a factor of 20 as we do not yet know the year-round contribution of large aggregates to sedimentation. In other seasons, our POC flux estimates are closer – but still elevated by greater than a factor of two compared with moored traps, and likely a minimum of a factor of three based on discussion of size frequency distribution results presented in Sec. 4.1. Missing contributions of large aggregates in sediment trap collections may explain the reported imbalance between

POC flux to the seafloor and benthic respiratory demand seen at nearby station M ($34^o$ 50'N, $123^o$ 00'W; Smith et al., 2013). We also show that particulate fluxes estimated using remotely sensed surface biomass may be overestimated sometimes - or underestimated at other times, depending on the population of grazers and the efficiency of export. We show that autonomous sampling technologies may be deployed to address many questions about the workings of the biological pump both offshore and onshore, including whether or not its strength has changed over the past

several decades and whether or not it will change in the future in the face of anthropogenic $CO_2$ warming and acidification, and exploitation of living resources of the ocean.

**Data Availability.** Data for plots, representative images used in this paper, and animations of image time series (at 1/2 resolution) are available as supplemental materials and are available from the author on request. Shipboard data sets: including

meteorological, surface hydrography, CTD cast data, are available through the Go Ship (http://www.go-ship.org) archives or in the case of ADCP data, are available through the University of Hawaii.

**Acknowledgements.**

The CFE was developed in close collaboration with Mike McClune and Russ Davis of the Scripps Institution of Oceanography Instrument Development Group. OSR design and construction had critical contributions from the

Electrical and Mechanical engineering groups and machine shop facilities at Lawrence Berkeley National Laboratory. Alex Morales (LBNL) and 25 UC Berkeley undergraduates facilitated CFE and BUOY-OSR deployments and recoveries at sea. In particular, Gabrielle Weiss, Amelia Weiss, Andrew Bower, and Christina Hamilton contributed both at sea and in the laboratory; we also thank the resident technicians, crews and captains of the R/V New Horizon for assistance. Jules Hummon (U Hawaii) assisted with ADCP data reprocessing. Hannah

Bourne (UC Berkeley) cross-checked ADCP and BUOY-OSR current shear calculations. The CFE data sets and animated image videos are posted as supplemental Information. Undergraduate participation in Cruises was facilitated by the UC Berkeley Undergraduate Research Apprentice Program and by the Earth and Planetary Science Department Ramsden fund. All work reported here was sponsored by National Science Foundation grant OCE-0936143 (JKBB).





**Author contributions.**

JKBB initiated the CFE program, was chief scientist at sea, and led the analysis and writing of this paper. TJW played a key engineering role in all aspects of development, deployment, programming, and operation of the CFE. MBF analysed MODIS Aqua satellite data.

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

**Appendix A. Methods Detail**

**A1.1 Optical Sedimentation Recorder on CFE.**

The Optical Sedimentation Recorder (OSR; *Fig. A1*) is independently powered from the SOLO float to which it is attached and the two communicate via a hard-wired link. The SOLO dives to its target depth and signals the OSR to begin operation. During time at depth, particles settle into a baffled (5 cm long x 1 cm wide hex cell) high aspect polished titanium funnel (height 23.6 cm; slope 75°; 15.4 cm diameter opening) and settling column (height ~11 cm) before depositing on a 2.5 cm diameter glass sample stage. The baffle is typical to the dimensions used in surface

tethered Particle Interceptor Traps (e.g. Martin et al. 1987, Collins et al., 2011, Stukel et al., 2013).

A downward focused 5 cm diameter white (color temperature 2700 K) light emitting diode (LED) source with bonded linear polarizer (Meadowlark Optics, Frederick, CO) is axially aligned with the funnel and provides transmitted light (or backlit) illumination of the sample.

An annular array of 12 white LEDs provides dark field illumination. This light source is integrated within the body of

a 3-D printed form designed to provide both a hydrodynamic cleaning function and to secure a 27 mm diameter dashed cross-hair reticle (Edmond Optics Inc., Barrington, NJ) to the pressure case window. A Seabird Inc. (Bellevue WA) pump outfitted with an isolation valve is used to flush the sample stage during cleaning.

A substage polarizer rotator is rotated either parallel or perpendicular to the axis of the downlight polarizer. In the case of CFE001, this polarizer rotator was housed within the instrument pressure case.

Under computer control, particles are imaged with a 5M pixel Sumix (SMX-11M5C; Oceanside, CA) colour imager in transmitted, transmitted – cross polarized, and dark field lighting modes which are designated as TRA, POL, DRK, respectively. Gain settings for each of the red, green, blue (RGB) channels were adjusted so that images of the down-light yield similar count value histogram averages (to within several %). The same gain settings were used for cross-polarized transmitted light illumination. For dark field, RGB channel balance was achieved by imaging salt crystals

dispersed across the sample stage. The camera lens was set to F16. Shutter timing was set for each of the three modes to prevent image saturation.

**A 1.2 Surface conditions, Hydrography, and Transmissometer data.**

Sea and wind conditions during the three study periods were benign. For example, In January and March 2013, winds monitored by the ship were less 5 m s$^{-1}$ and averaged 2.5 m s$^{-1}$. In May 2012, winds were less than 10 m s$^{-1}$ and





averaged 3 m s$^{-1}$. Surface waves and swell were almost completely absent in January and March. In January 2013, the weather was completely cloud free which contributed to excellent satellite imagery of the area.

During all expeditions, a CTD/rosette system with a C-Star (WETLabs, Inc. Philomath, OR) transmissometer and Seapoint Inc. scattering sensor was repeatedly deployed to full water column depth. Approximately 20 casts were

completed during each cruise. Only transmissometer data are reported here. The transmissometer cleaning, calibration and data processing to particle beam attenuation coefficient ($c_p$) follow protocols we have published (Bishop and Wood, 2008, Bishop, Lam and Wood, 2012). POC was calculated by multiplying particle beam attenuation coefficient by 27 (Bishop and Wood, 2008).

Figure A2 (A-C) shows $c_p$ profiles for the three expeditions. Also shown is 0-20 m averaged POC from these casts.

The time line for plot (D) is relative to the start of deployment of the CFE. Average and standard deviation values of 0-20 m averaged POC are quoted in the text.

The effects of internal waves with periods of 20-30 minutes were usually seen in CTD cast data. In March 2013, internal wave amplitudes of approximately 40 m were observed during CTD tow-yo experiments. During other expeditions, wave amplitudes were less than 20 m.

**A 1.3 BUOY-OSR configuration, January 2013.**

Figure A3 depicts the deployment configuration of the sample collecting BUOY-OSR used in this study. A SBE 19-plus logging CTD (Sea Bird Electronics, Bellevue WA) recorded pressure, temperature and salinity at 5 second intervals. One OSR (Buoy instrument 84) was configured identically to CFE001. The funnel assembly of the second OSR (Buoy instrument 88) had a sample stage leak enabling strong water circulation through the funnel leading to

particle loss as the system was pumped up and down due to wave action. Buoy instrument 84 is considered reliable as particles remained in place where they fell as they accumulated.



**Table 1. Volume Attenuance Size Distributions for CFE and BUOY-OSR Samples from January 2013.**

| Diameter[1] | 144 m CFE | | 320 m CFE | | 507 m CFE | | 237 m BUOY[2] | | | 237 m BUOY[3] | |
|---|---|---|---|---|---|---|---|---|---|---|---|
| (µm) | 20130120 11:11 | | 20130120 16:42 | | 20130120 05:56 | | 20130120 AVG | | | 20130120 18:08 | |
| >5000 | 0.462[4] | 0.462[5] | 0.000 | 0.000 | 0.776 | 0.776 | 0.000 | 0.000 | ±0.000 | 0.000 | 0.000 |
| 2000-5000 | 0.441 | 0.903 | 0.978 | 0.978 | 0.150 | 0.926 | 0.065 | 0.065 | ±0.130 | 0.494 | 0.494 |
| 1000-2000 | 0.059 | 0.962 | 0.000 | 0.978 | 0.055 | 0.981 | 0.353 | 0.418 | ±0.090 | 0.169 | 0.662 |
| 500-1000 | 0.014 | 0.976 | 0.007 | 0.984 | 0.003 | 0.983 | 0.250 | 0.668 | ±0.070 | 0.153 | 0.815 |
| 200-500 | 0.015 | 0.992 | 0.005 | 0.989 | 0.011 | 0.994 | 0.245 | 0.913 | ±0.022 | 0.144 | 0.959 |
| 100-200 | 0.005 | 0.996 | 0.007 | 0.996 | 0.003 | 0.997 | 0.064 | 0.977 | ±0.006 | 0.030 | 0.989 |
| 50-100 | 0.003 | 0.999 | 0.003 | 0.999 | 0.002 | 0.999 | 0.017 | 0.994 | ±0.001 | 0.008 | 0.997 |
| 20-50 | 0.001 | 1.000 | 0.001 | 1.000 | 0.001 | 1.000 | 0.006 | 1.000 | ±0.000 | 0.003 | 1.000 |

1. Diameter is Equivalent Circular Diameter of particle with the same area;

2. Averages (excluding 18:08 data) for BUOY data, third column (±) is s.d. of cumulative data.

3. Currents below 3 cm/sec for 60 min.

4. Volume Attenuance for indicated size interval.

5. Cumulative volume attenuance for particles greater than lower limit of size interval





**Figure Captions.**

Figure 1. (A) Carbon Flux Explorer (CFE). (B) Santa Cruz Basin Study Area with bathymetry from Google Earth. The SCB is 1900 m deep and exchanges water with the open ocean at depths shallower than 1000 m. C) Detail of CFE surfacing positions (large symbols), CTD – optics casts (small symbols), and drift track of the BUOY-OSR system (blue line). CFE and BUOY-OSR deployments took near 33.75°N 119.5°W. CTD profiles were taken within several km of CFE and BUOY-OSR positions.

Figure 2. Detail of Image Analysis Scheme for CFE and BUOY-OSR images. Images from NH1301 CFE001 deployment. (A) Raw Backlit (TRA) images are divided by the in-situ TRA lighting reference (Cross hair reticle has 1 mm sized dashes) to get transmittance. This image is transformed to units of attenuance. (B) Raw crossed polarized transmitted light [POL] image is first corrected for attenuation effects of optically dense particles by dividing by PCF, an empirically derived factor proportional to attenuation (ATN). The POL lighting REF is then subtracted and then divided by the TRA Lighting ref to yield the normalized POL image. The contrast and brightness of the final image is enhanced. (C). Raw Dark Field (DRK) images are normalized by dividing by the in-situ lighting reference intensity.

Figure 3. Magnified detail of a marine snow aggregate from the upper left hand quadrant after processing to normallized Dark Field (DRK) counts, Attenuance (ATN), and polarized (POL) counts. The scale of the aggregate is ~1 cm. Image resolution is 13 μm. An empty 600 μm Pteropod shell is at the bottom left of the POL image. The 10 bright spherical regions in the POL image are ~200 μm sized Foraminifera shells; haze in aggregates is likely due to coccoliths.

Figure 4. Results from CFE deployments in the Santa Cruz Basin. CFE depths shown in italics. (A) sample attenuance from transmitted light images. (B) polarized photon yield in parts per million from cross-polarized light images

Figure 5. $POC_{ATN}$ (A) and $PIC_{POL}$ (B) flux systematics from CFE deployments in the Santa Cruz Basin. CFE depths are shown in italics in (A). Bars and small circles denote average flux over 1.8 and 0.3 hour intervals, respectively.

Figure 6. A) Chlorophyll-a and POC from the MODIS Aqua satellite; lines are 5 point running means. Color scheme for May 2012 (red), Jan. 2013 (blue) and March 2013 (green). Open and filled graphics in A and B correspond to the week previous to and during each expedition. (B) From left to right successive bar pairs depict satellite chlorophyll and satellite POC. Crosshatched bars denote 0-20 m transmissometer POC. (C) POCATN flux for January (blue circles), March (green squares), and May (red triangles). The large filled symbols are a grand average for a particular depth, small open symbols denote 1.6 hour averaged data. (D) PICPOL flux profiles. Symbols as in C.

Figure 7. Depth profiles of the ratio of PIC and POC flux proxys. Results show highest relative contribution of PIC occurred in March 2013. January and May values near 500 m are comparable.

Figure 8. Time series of image attenuation from CFE and BUOY-OSR deployments in the Santa Cruz Basin during January 2013. Time axis is elapsed days since January 1 at 0000 UTC. In (A) and (B), the green lines depict the ATN time series. In (B), red symbols depict currents relative to the BUOY-OSR at a depth of 237 m that were calculated using ship broad band acoustic current profiler data (RDI Instruments, Ocean Surveyor 75; methods described by Hummon and Firing (2003) and 30 minute BUOY-OSR GPS positions. The current dropped briefly below 1 cm s-1 and the rate of accumulation of particles in the BUOY-



OSR increased. This moment was also the time when the first large aggregates were seen. Also shown in (B) in blue is the temperature time series from the logging CTD. The most obvious effect on particle collection rate by the BUOY-OSR was the relative lull in currents near day 19.68. Images from samples depicted by * are compared in Figure 9. (C) Time series of CFE and BUOY-OSR tilt in degrees. CFE tilt averaged 3 degrees, whereas BUOY-OSR tilt averaged 0.5 degrees.

Figure 9. Comparison of representative Dark Field Images of accumulated settling particles over a 2.5 hour period from an Optical Sedimentation Recorder (OSR) on the Carbon Flux Explorer (CFE, upper left) vs. that from a surface BUOY-tethered OSR (upper right) during NH1301 (Jan. 2013). The CFE was observing sedimentation at 320 m (image 21:14 UTC). The BUOY-OSR was deployed at 237 m (image 18:07 UTC). Each image is 2.5 cm across and pixel resolution is 14 μm. The Large

fluffy 0.5 – 1.0 cm sized marine snow aggregates were almost completely missed by the BUOY OSR. Winds were near calm (< 5 m s$^{-1}$) and surface waves and swell were almost entirely absent. The BUOY system had ~0.5° of tilt. The only difference between the CFE and BUOY was the absence/presence of tidally fluctuating horizontal currents of up to 8 cm sec$^{-1}$. Only fragments of the marine snow particles were seen in the BUOY samples.

Figure 10. Comparison of Normalized Cumulative Attenuance – Size Distribution (NCA-SD) for images from CFE and BUOY-OSR systems, January 2013. Over 96% of the cumulative volume attenuance loading was in the >1000 μm fraction in CE samples. The BUOY system images at 11:26 and 18:08 corresponded to times when current velocity above the mouth of the trap was 6 cm s$^{-1}$ and less than 2 cm s$^{-1}$, respectively. The cumulative distributions for the BUOY-OSR can be matched by eliminating all of the >1500 μm sized material from the CFE size distributions.

Figure 11. Comparison of Normalized Cumulative Volume Attenuance – Size Distributions (CVA-SD) from CFE deployments in May 2012 (red lines and symbols), January 2013 (blue), and March 2013 (green). Shown in grey is the average CVA-SD from BUOY-OSR deployments. No CFE data from any of the three expeditions over lapped BUOY-OSR results from January 2013; in all cases, larger particles were observed by the CFE.

Figure A1. Drawing depicting the configuration of CFE001 and detail of lighting/imaging elements of the Optical Sedimentation Recorder (OSR).  During operation, the OSR uses a 3 axis accelerometer to monitor system tilt.

Figure A2. (A-C) Particle beam attenuation coefficient ($c_p$) profiles corresponding to NH1301 (Jan 2013), NH1304 (Mar. 2013),
and NH1204 (May/June 2012) Expeditions. (D) POC calculated for the 0-20 m depth interval using $c_p$ multiplied by 27. The x axis is in days relative to the start of each CFE deployment.

Figure A3. Photograph of the surface tethered BUOY-OSR as deployed in January 2013. Also shown is the mooring configuration used in this study. The seven small floats spaced 1 m apart are intended to minimize the effects of surface wave
action on the up and down motion of the subsurface sediment trap. During the January 2013 deployment depth was 237 m (±0.25 m S.D.) – Maximum peak to trough vertical motion was 1 m.





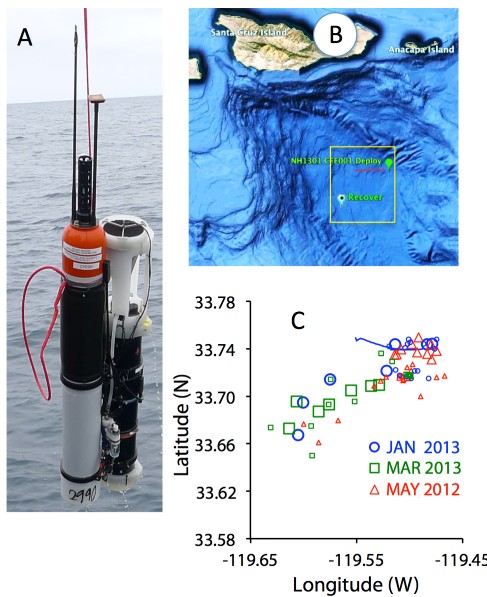

Figure 1. (A) Carbon Flux Explorer (CFE). (B) Santa Cruz Basin Study Area with bathymetry from Google Earth. The SCB is 1900 m deep and exchanges water with the open ocean at depths shallower than 1000 m. C) Detail of CFE surfacing positions (large symbols), CTD – optics casts (small symbols), and drift track of the BUOY-OSR system (blue line). CFE and BUOY-OSR deployments took near 33.75°N 119.5°W. CTD profiles were taken within several km of CFE and BUOY-OSR positions



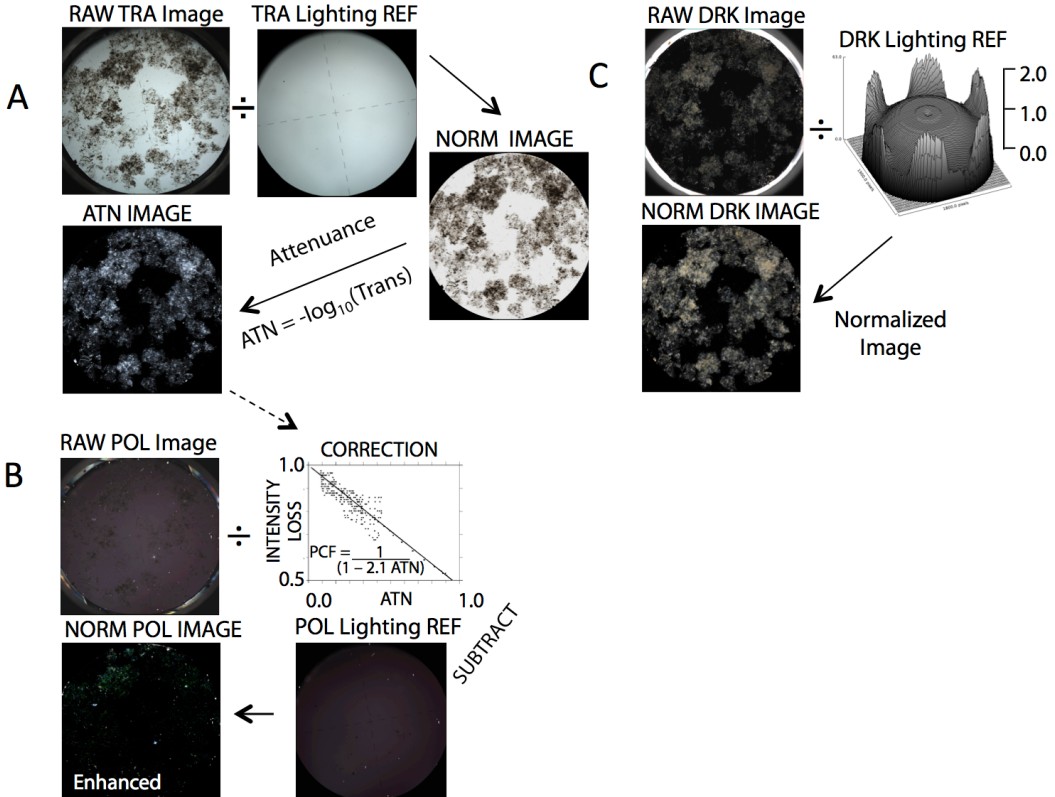

Figure 2. Detail of Image Analysis Scheme for CFE and BUOY-OSR images. Images from NH1301 CFE001 deployment. (A) Raw Backlit (TRA) images are divided by the in-situ TRA lighting reference (Cross hair reticle has 1 mm sized dashes) to get transmittance. This image is transformed to units of attenuance. (B) Raw crossed polarized transmitted light [POL] image is first corrected for attenuation effects of optically dense particles by dividing by PCF, an empirically derived factor proportional to attenuance (ATN). The POL lighting REF is then subtracted and then divided by the TRA Lighting ref to yield the normalized POL image. The contrast and brightness of the final image is enhanced. (C). Raw Dark Field (DRK) images are normalized by dividing by the in-situ lighting reference intensity.





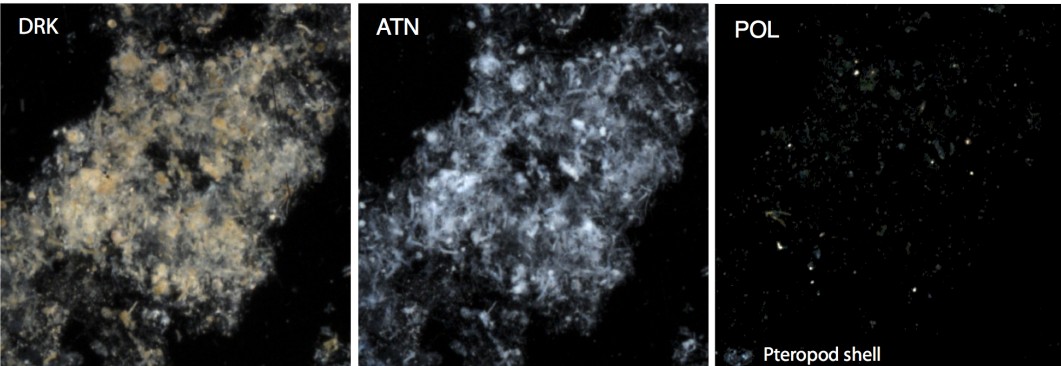

Figure 3. Magnified detail of a marine snow aggregate from the upper left hand quadrant after processing to normallized Dark Field (DRK) counts, Attenuance (ATN), and polarized (POL) counts. The scale of the aggregate is ~1 cm. Image resolution is better than 15 µm. An empty 600 µm Pteropod shell is at the bottom left of the POL image. The 10 bright spherical regions in the POL image are ~200 µm sized Foraminifera shells; haze in aggregates is likely due to coccoliths.

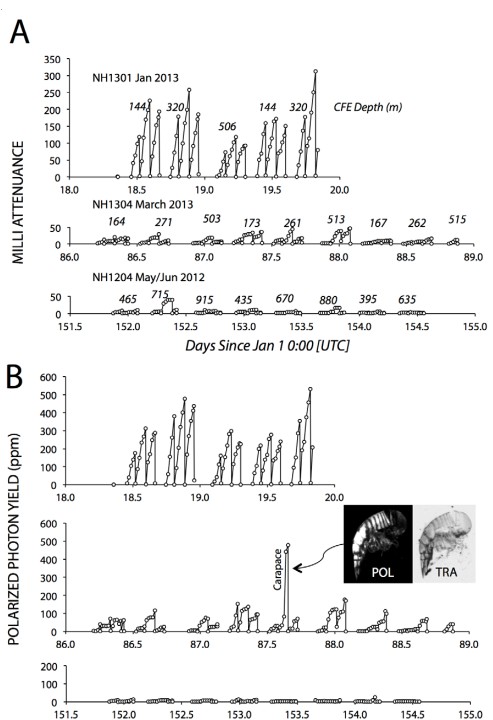

Figure 4. Results from CFE deployments in the Santa Cruz Basin. CFE depths shown in italics. (A) sample attenuance from transmitted light images. (B) polarized photon yield in parts per million from cross-polarized light images.





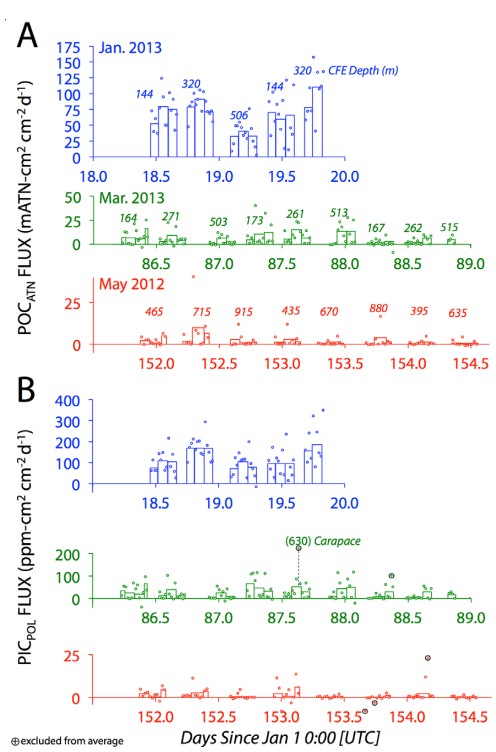

Figure 5. $POC_{ATN}$ (A) and $PIC_{POL}$ (B) flux systematics from CFE deployments in the Santa Cruz Basin. CFE depths are shown in italics in (A). Bars and small circles denote average flux over 1.8 and 0.3 hour intervals, respectively.





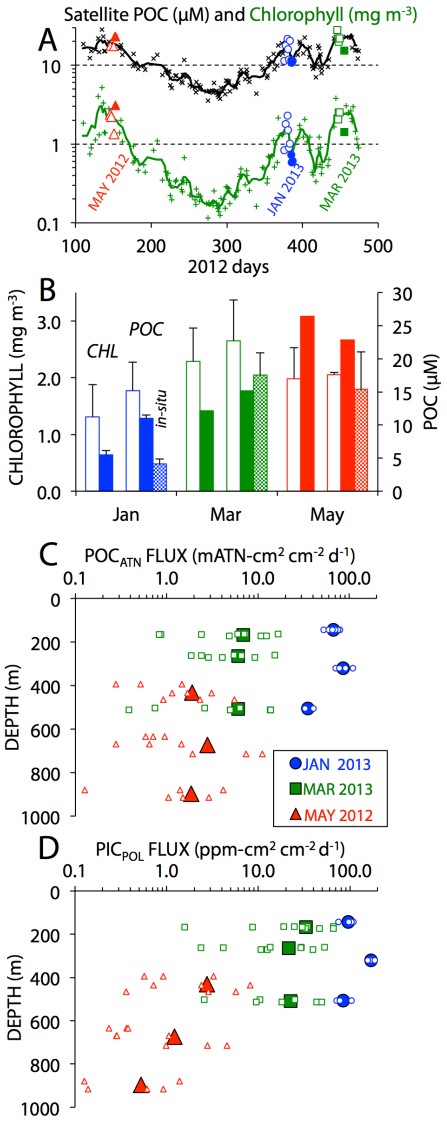

Figure 6. A) Chlorophyll-a and POC from the MODIS Aqua satellite; lines are 5 point running means. Color scheme for May 2012 (red), Jan. 2013 (blue) and March 2013 (green). Open and filled graphics in A and B correspond to the week previous to and during each expedition. (B) From left to right successive bar pairs depict satellite chlorophyll and satellite POC. Crosshatched bars denote 0-20 m transmissometer POC. (C) $POC_{ATN}$ flux for January (blue circles), March (green squares), and May (red triangles). The large filled symbols are a grand average for a particular depth, small open symbols denote 1.6 hour averaged data. (D) $PIC_{POL}$ flux profiles. Symbols as in C.





Figure 7.

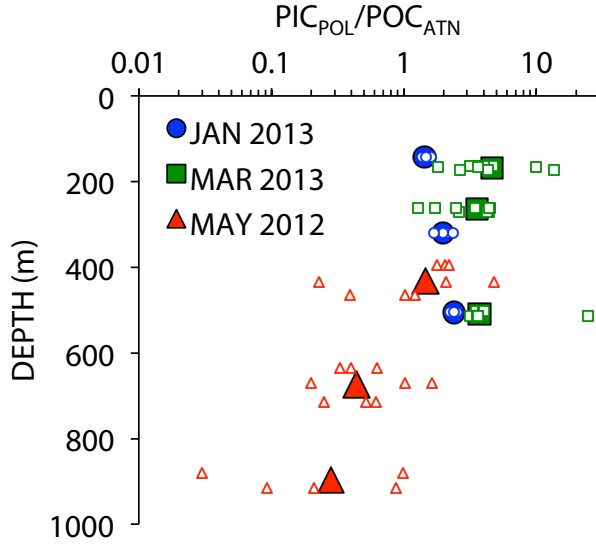

Figure 7. Depth profiles of the ratio of PIC and POC flux proxys. Results show highest relative contribution of PIC occurred in

5   March 2013. January and May values near 500 m are comparable.



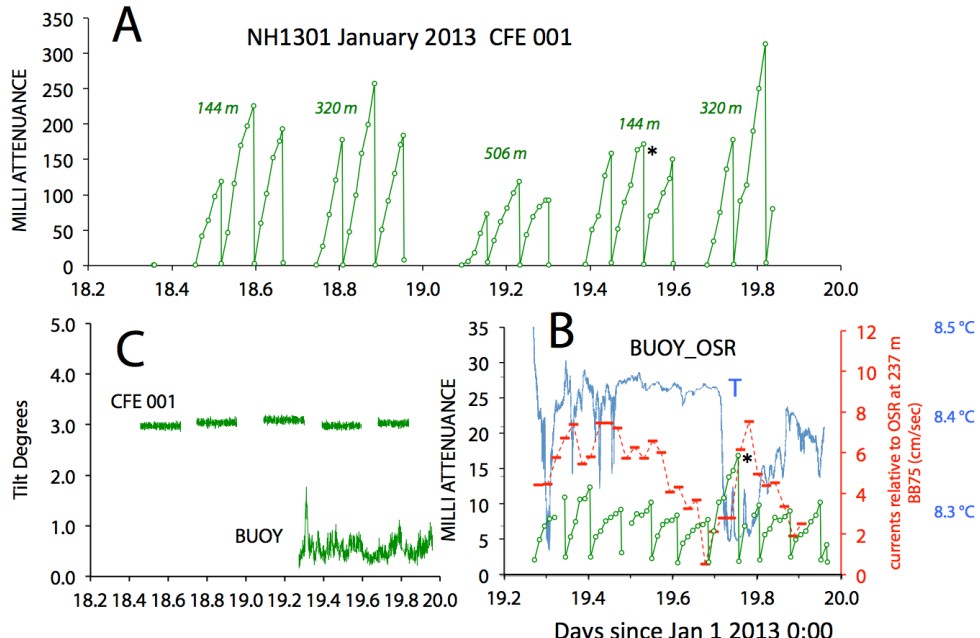

Figure 8. Time series of image attenuance from CFE and BUOY-OSR deployments in the Santa Cruz Basin during January 2013. Time axis is elapsed days since January 1 at 0000 UTC. In (A) and (B), the green lines depict the ATN time series. In (B), red symbols depict currents relative to the BUOY-OSR at a depth of 237 m that were calculated using ship broad band acoustic current profiler data (RDI Instruments, Ocean Surveyor 75; methods described by Hummon and Firing (2003) and 30 minute BUOY-OSR GPS positions. The current dropped briefly below 1 cm s$^{-1}$ and the rate of accumulation of particles in the BUOY-OSR increased. This moment was also the time when the first large aggregates were seen. Also shown in (B) in blue is the temperature time series from the logging CTD. The most obvious effect on particle collection rate by the BUOY-OSR was the relative lull in currents near day 19.68. Images from samples depicted by * are compared in Figure 9. (C) Time series of CFE and BUOY-OSR tilt in degrees. CFE tilt averaged 3 degrees, whereas BUOY-OSR tilt averaged 0.5 degrees.





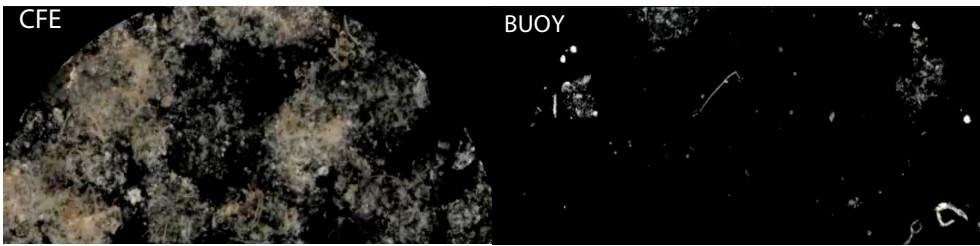

Figure 9. Comparison of representative Dark Field Images of accumulated settling particles over a 2.5 hour period from an Optical Sedimentation Recorder (OSR) on the Carbon Flux Explorer (CFE, upper left) vs. that from a surface BUOY-tethered OSR (upper right) during NH1301 (Jan. 2013). The CFE was observing sedimentation at 320 m (image 21:14 UTC). The BUOY-OSR was deployed at 237 m (image 18:07 UTC). Each image is 2.5 cm across and pixel resolution is 14 μm. The Large fluffy 0.5 – 1.0 cm sized marine snow aggregates were almost completely missed by the BUOY OSR. Winds were near calm (< 5 m s$^{-1}$) and surface waves and swell were almost entirely absent. The BUOY system had ~0.5° of tilt. The only difference between the CFE and BUOY was the absence/presence of tidally fluctuating horizontal currents of up to 8 cm sec$^{-1}$. Only fragments of the marine snow particles were seen in the BUOY samples.



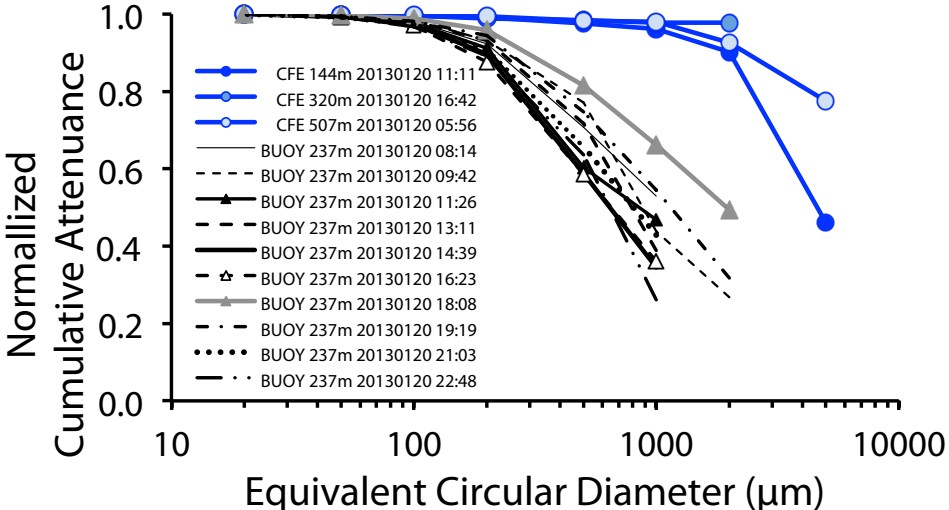

Figure 10. Comparison of Normalized Cumulative Attenuance – Size Distribution for images from CFE and BUOY-OSR systems, January 2013. Over 96% of the cumulative volume attenuance loading was in the >1000 µm fraction in CE samples. The BUOY system images at 11:26 and 18:08 corresponded to times when current velocity above the mouth of the trap was 6 cm s$^{-1}$ and less than 2 cm s$^{-1}$, respectively. The cumulative distributions for the BUOY-OSR can be matched by eliminating all of the >1500 µm sized material from the CFE size distributions.

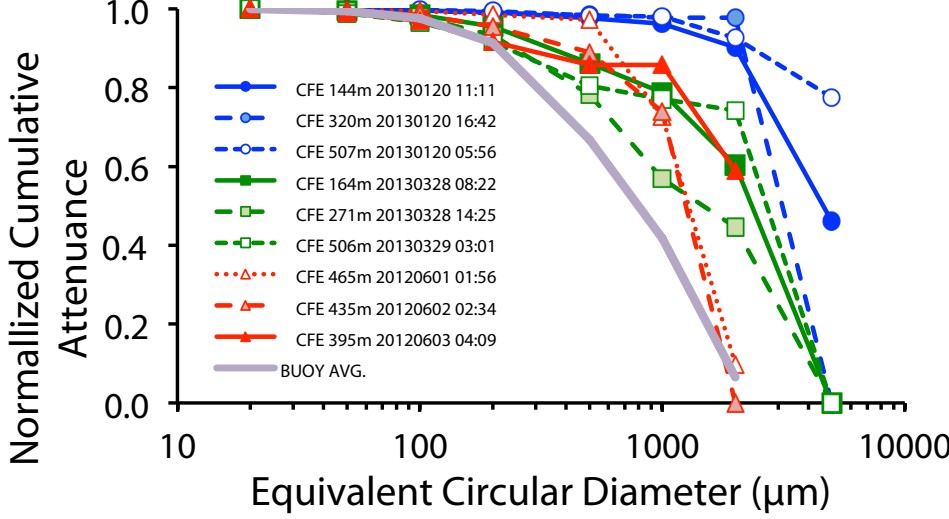

Figure 11. Comparison of Normalized Cumulative Volume Attenuance – Size Distributions (CVA-SD) from CFE deployments in May 2012 (red lines and symbols), January 2013 (blue), and March 2013 (green). Shown in grey is the average CVA-SD from BUOY-OSR deployments. No CFE data from any of the three expeditions over lapped BUOY-OSR results from January 2013; in all cases, larger particles were observed by the CFE.





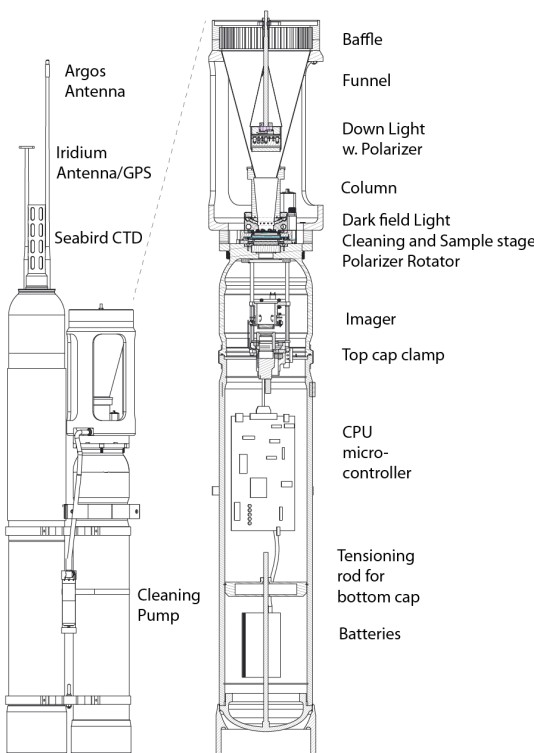

Figure A1. Drawing depicting the configuration of CFE001 and detail of lighting/imaging elements of the Optical Sedimentation Recorder (OSR). During operation, the OSR uses a 3 axis accelerometer to monitor system tilt.



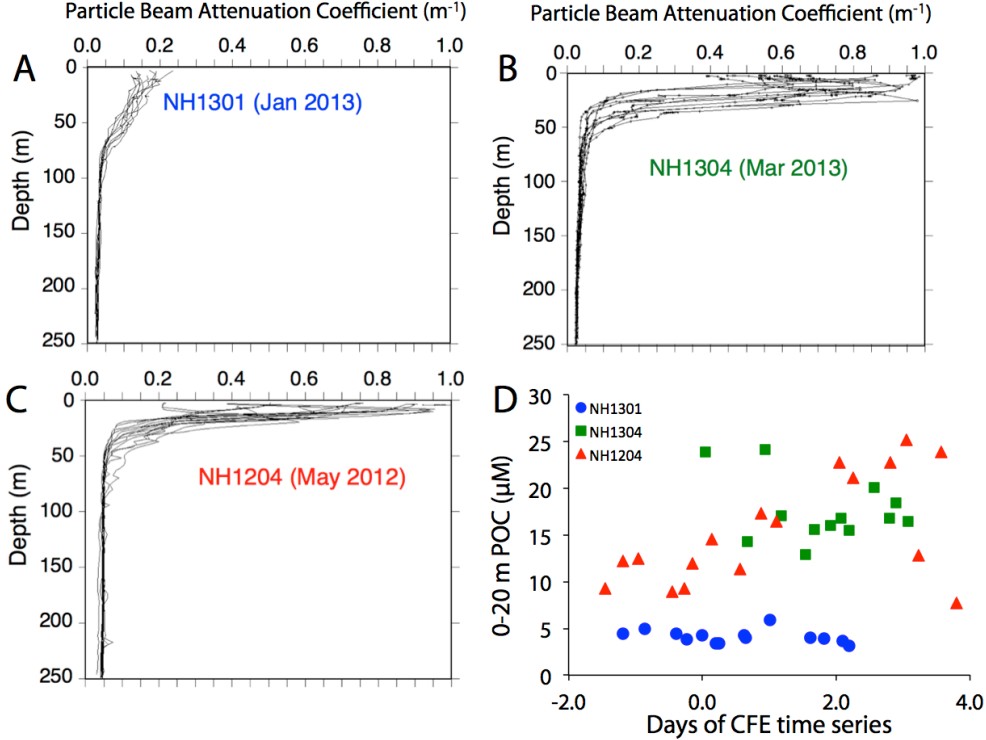

Figure A2. (A-C) Particle beam attenuation coefficient ($c_p$) profiles corresponding to NH1301 (Jan 2013), NH1304 (Mar. 2013), and NH1204 (May/June 2012) Expeditions. (D) POC calculated for the 0-20 m depth interval using $c_p$ multiplied by 27. The x axis is in days relative to the start of each CFE deployment.





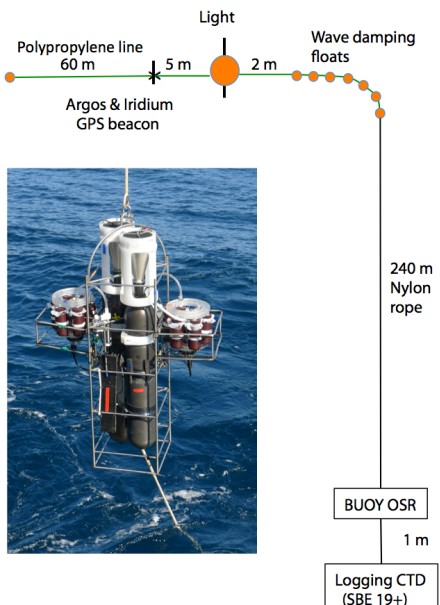

Figure A3. Photograph of the surface tethered BUOY-OSR as deployed in January 2013. Also shown is the mooring configuration used in this study. The seven small floats spaced 1 m apart are intended to minimize the effects of surface wave action on the up and down motion of the subsurface sediment trap. During the January 2013 deployment depth was 237 m (±0.25 m S.D.) – Maximum peak to trough vertical motion was 1 m.