# Peer review of "Robotic observations of high wintertime carbon export in California coastal waters"

_Biogeosciences, 2016_

## Referee Comment (RC1) · Anonymous Referee #1 · 11 Apr 2016

This paper presents results from robotic observations of carbon export flux from the Santa Cruz Basin located within the coastal upwelling waters off California. The authors report on high resolution POC and PIC fluxes derived from autonomous optical measurements, i.e. Carbon Flux Explorers (CFEs). This method captures particulate organic carbon fluxes below 140 m in the Santa Cruz Basin, CA, in 2 different years and 3 different seasons. One major finding is that the optical data reveal carbon fluxes 20 times higher than fluxes derived from surface tethered sediment traps. Obviously, the traps under sample larger marine snow aggregates of the size class larger than 1mm. Further, optically-derived fluxes were about one order of magnitude higher than previously measured multi-year sediment trap fluxes.

Carbon export from the photic zone is an important issue and only a few methods provide reliable flux data from the upper ocean (e.g. neutrally buoyant traps (NBST),

other free-drifting systems, Martin et al., 1987). These data points determine carbon attenuation curves which may be variable both in space and time depending on the local biochemical conditions and ecology. Additionally, this export sets the conditions for deep ocean carbon flux classically measured with moored sediment traps.

The ms is well and clearly written and fits with no doubt into the concept of BG. I have some major and minor comments/suggestions outlined below.

Following the abstract and introduction, the methods were described in great detail. Sometimes I feel that this is more a methodological manuscript. This can be seen in the high number of figures describing the methods. Even in the result and discussion sections, methodological aspects can be found (e.g. results: chapter 3.3., third para-graph, chapter 3.4. . . .). I wonder whether and how the methods described here are different from the ones presented in earlier papers, e.g. in Bishop et al., 2004 ('Robotic observations. . . . . .). This should be clarified and then, the method sections could be shortened and the focus concentrated on the gathered data. If the methods differ, the changes/improvements could be briefly outlined. Some figures could also be moved to the appendix.

My major concern, however, is the comparison between optically- and trap-derived (surface tethered and moored traps) fluxes. Both approaches are fundamentally dif-ferent and comparisons are limited by seasonal, interannual and regional variability. It would be helpful to have more information on the locations of the different types of sediment traps used for comparison (e.g. in Fig. 1) and the exact seasons/years and the water depths. In chapter 4.1. the CFE fluxes from the Santa Cruz Basin were compared to fluxes from the nearby Santa Barbara and San Pedro Basins which were derived from different water depths and different sediment trap types. Flux data from Martin et al. (1987) were captured from more open ocean sites in the far north of the Pacific in different years rather than from a more local basin. To facilitate reading and to follow the presented argumentation and evaluate this comparison more critically, the authors should show a table with optical and sediment trap fluxes but with detailed information on locations, seasons, years, water depths, surface or bottom-tethered arrays with traps, type of traps and openings.

Further, the authors cannot explain the large differences (10-20 times) of fluxes in a satisfactory way. It is hard to believe that the CFE's optical sedimentation recorded (OSR) with 1 cm opening was able to better sample large marine snow aggregates of several mm size (which are rather rare in the water column) than a cylindrical tube of about a decimeter or larger in size of a free-drifting (surface tethered) sediment trap. I would expect this to be the other way round. It is hard to believe that the baffles of sedimentation tubes are so small to destroy fragile marine snow aggregates of a few mm in size (page 11, uppermost chapter). By the way, the opening of the tube of the OSR (1 cm) is not larger than the cm-sized trap baffle openings (page 11 lines 6-9).

Sediment traps have their limitations as well, depending for instance on the type of array (surface vs bottom tethered). In particular, bottom-fixed moorings with shallow water sediment traps seem to be critical to record fluxes in the upper few hundred meters of the water column. Shallow traps may not provide an accurate measure of particle fluxes and differ by a factor of 3-10 (Buesseler 1991). Part of the discrepancies discussed here may be attributed to these uncertainties in trap-derived fluxes (e.g. page 9, chapter 4.1. of discussion). Particle fluxes from sediment traps (NBST, surface tethered, moored) on the other hand, measure carbon fluxes more directly and apply less assumptions than the optical methods. There are less than a factor of two differences of fluxes between NBST and surface tethered traps as the authors mention at the end of page 10.

Summarizing this, I suggest to be more careful with this kind of comparison and the conclusions. The problems associated with this comparison of fluxes (optical vs traps) should be clearly mentioned in the discussion.

Minor issues

- page 1 , line 28: . . .by grazers or settle down as larger marine snow particles.

- The optical methods to estimate carbon fluxes which are described here need several assumptions, e.g. conversation factors (chapter 2.3.). What are the errors of the individual methods and the potential cumulative errors? Is there any estimation/quantification? Something written in earlier papers?

- page 5, chapter 2.2.4. Do the authors only hypothesize that attenuance is the best proxy for POC? What is the basis for this assumption, please clarify

- page 8, chapter 3.3., sometimes hard to read due to many abbreviations

- Fig. 1, show surroundings of the Santa Cruz Basin to provide more info on the general setting of the study site and the other sites used for comparison

―――――――――――――――――――――――――

---

## Referee Comment (RC2) · Anonymous Referee #2 · 13 Apr 2016

General comments This manuscript describes the first use of a powerful new observational technology. Pending some revisions detailed below, it will make an important contribution to the literature on methods for observing the ocean's biological carbon pump. These data are the first of their kind, and if the CFE technology continues to be robust in the field, these types of measurements could revolutionize our understanding of the ocean biological pump. Aside from the new method, this paper's scientific findings include high sinking POC flux under apparent low-productivity conditions in an upwelling, coastal system, and undersampling of sinking POC by surface-tethered relative to neutrally-buoyant sediment traps. Most of the specific comments below pertain to instances where strong, rather general statements are not fully supported by the data as presented. In some cases, the data just need to be clarified or more technical detail added to the text. In other cases, more nuance should be added to the interpre-

tations. A theme through several of the comments below is that more attention should be paid to spatial variability in the measured C fluxes, particularly in comparisons of in situ observations to satellite data. In general, the necessary changes should be straightforward for the authors to address.

Specific comments

p. 2, L. 4. The specificity of the sentence about foraminifera shells, etc. suggests that perhaps a reference is necessary.

p. 2, L. 36. Can the CFE really operate for years? It would be more useful to most readers to point out the demonstrated length of deployment so far (months?) or give the nominal number of profile cycles that can generally be achieved.

Methods section. I suggest you tabulate the cruise numbers, deployment locations/times/depths, and retrieval locations/times/depths to help the reader keep track of the different observations.

Section 2.1. It may be a good idea to move the basic trap funnel and stage dimensions from Appendix A to here. Otherwise, you don't state the collection area anywhere. I noticed that another reviewer has assumed the OSR has a 1 cm diameter trap opening, which is not the case and may have led to some misunderstanding.

P. 3, L. 20. Use "thickness" instead of "length" to describe the vertical dimension of the baffle.

P. 3 L. 28-30. If any of the OSR data are transmitted, they must be pre-processed on board prior to doing so, correct? If this is something that has been implemented, a brief description of the on-board processing steps should be given here. Otherwise, please remove "OSR data" from the list of things that are transmitted during surfacing cycles.

P. 3 L. 29. State the actual dive depth rather than "considerably deeper than planned."

P. 4, L. 12-13: While it is ultimately the author's call, I suggest a change from base-10

to base-e. I believe that this is the convention for most of the optical oceanography community when describing optical properties.

P. 4, L. 20 (and related discussion in Section 2.3): How have you dealt with particles that overlap? Would stepwise subtraction lead to a possible underestimation of flux? Please address.

P. 4, L. 30: Is there a poster or meeting presentation which has shown the elimination of the stress polarization interference, which you can refer to here? If not, this sentence should be reworded to indicate that the problem is surmountable, but the discussion of later CFE builds should be removed as they are not relevant to the data presented here.

Section 2.2.3 and P. 7 L. 7-8, and supplemental videos: Presumably swimmer interference should also affect analysis of the POC flux data, not just the PIC flux. Please revise this discussion so that it applies to both proxies. The supplemental videos clearly show several instances of "disappearing/moving particles" (e.g., halfway through 1204 and maybe again at the end; also about halfway through 1301). Unlike direct sediment traps, you can actually detect and correct for the presence of swimmers with the OSR, and I think this should be discussed in more detail.

P. 5, L. 21: Please add a reference for the statement that birefringence scales linearly with PIC concentration.

P. 6, L. 6-16: Please add an equation or two summarizing the calculation you have described here in words. It will greatly clarify the procedure.

P. 6, L. 18-28: It would be illustrative to also compute fluxes using models in the literature for aggregate carbon content as a function of size. For instance, Alldredge (1998) contains useful relationships for several categories of marine snow. At the very least, you should mention the existence of such models and their relevance to interpretation of image data such as that collected by the OSR.

P. 6, L. 33-34 and Figure 1. Add a distance scale bar to Fig. 1c. You state that all satellite data points used for comparison were within a 2km radius of 33.72°N, 119.5°W, but that appears to be the center of a "150 km2" study region, and does not correspond to the actual surfacing locations of the CFE or optics cast locations. If Chl and POC were patchy, then the changes seen in the surface optical properties at a fixed point in the center of the box are unlikely to correspond with the observations on a quasi-Lagrangian platform drifting tens of kilometers from this point. Comparisons to satellite data should attempt to match up with the actual locations of the CFE tracks. Once this is done, please adjust the text accordingly.

Section 3.2: How close in time and space were the different depths measured on each deployment? These details should be made clear in the text or in Figure 1. That is, it is possible the CFEs were sampling different sinking particle pools at each of the different depths? It may not be correct to assume a single attenuation model.

p. 10 L. 11-14. You invoke certain physical conditions here in order to support the hypothesis that low-biomass conditions were caused by consumer-driven export and not by physical aggregation or advection. However, the minimal wind and current shears you describe are inconsistent with your other major finding that your surface-tethered BUOY-OSR undersampled by a factor of 20 relative to the CFE due to strong hydrodynamic effects felt by the different platforms. Please reconcile these ideas.

p. 10, L 16-17 and Figure 6: The trend is hard to pick out from Figure 6. You need to add an inset that shows the January 2013 period. Otherwise it looks like the 1-week-prior points are scattered, not necessarily decreasing, and in any case they cover up the running-mean line so it cannot be seen.

P. 10 L. 18-19. "Satellite imagery from Jan 2013 shows a patchy POC/chlorophyll distribution without obvious eddy structures or fronts near by." This statement is not possible to evaluate from the satellite chlorophyll images included in the Supplementary Information. It certainly appears as though there are potential eddy structures and fronts in

these images. However, the images do not have latitude and longitude marked, there is no color bar (is it log or linear? What are the scale limits?) nor are the CFE deployment locations marked. These images need to be clearly annotated so it is obvious that they support the claim that there were no nearby eddy structures or fronts. Otherwise it is not possible to differentiate a rapid temporal change in POC from a rapid spatial change, and this assertion should be removed.

P. 10, L. 22-23: What are the uncertainties on your derived "Martin" b values? Are they even significant? (If not, add a statement to this effect – it adds strength to your finding that the Martin curve is an inappropriate model for these data). However, you should also mention again the time and distance separations among the different depth measurements – if export was patchy, then it could be that each sampled depth is too far from the others to infer a continuous attenuation profile.

Section 4.1: The difference between surface tethered and neutrally buoyant traps may be more pronounced in the presence of large aggregates such as the ones you have observed here. The studies you cite comparing PITS and NBST traps were conducted in an oligotrophic region where in situ camera profiles showed low concentrations of particles larger than 1500 $\mu$m (McDonnell and Buesseler, 2012). Your findings in coastal California are quite striking, but there may be site-specific differences in the relative efficiencies of tethered/neutrally-buoyant traps at collecting aggregates. Please revise lines 34-36 to address the differences between the different types of environments.

P. 11, L. 5-6: Near-horizontal approach of particles to tethered traps has been described in detail by Siegel et al. 2008; I suggest you include a citation to this reference.

P. 11 L. 21: Replace "the single profile 234Th/238U method" with "the 234Th/238U method with a steady-state assumption", which is clearer.

P. 11 L. 23-24: Similarly, replace "time series sampling" (which is less specific) with something like, "multiple reoccupations of a water parcel assuming non steady-state

conditions".

P. 11 L. 22: Replace "is not applicable" with "may not be applicable". If it can be established that a coastal system is in steady state and advection is minimal, then the steady-state assumption can be used.

Figure 9: Please put a thin margin between these panels.

Movies in Supplement. Do these represent multiple depths and profile cycles? Can you make this information clearer in the "readme" file? If there are multiple cycles represented in the videos, please insert "marker" frames so it's clear where the breaks are. Also, the movies occasionally show "disappearing" particles. Are these zooplankton? How are they treated in the flux estimation calculation? (see also the comment on section 2.2.3)

Technical comments

p. 1, L. 14, change "monitor" to "monitored" p. 1, L. 19, Break into two sentences. Start 2nd sentence with "Multiple lines of evidence indicate..." p. 1., L. 19. Remove space from "under sampling" p. 1, L. 20. Change "compared to" to "than the" p. 2, L. 3. "coccoliths" should be singular p. 2 L. 13, change "near by" to "nearby" P. 2 L. 13, should be "strong, recent weakening" P. 2 L. 18. It is unnecessary to abbreviate Eppley and Peterson 1979 because you only cite it once more. p.2 L.34. Insert "that" before "we have developed". p.3 L. 4. "gain detail of the" is awkwardly-worded. p. 8 L. 5. "artifact" is misspelled. P. 10 L. 17: Should refer to Figure 6, not Figure 4.

References:

Alldredge, A. 1998. The carbon, nitrogen and mass content of marine snow as a function of aggregate size. Deep Sea Research Part I: Oceanographic Research Papers 45: 529–541.

McDonnell, A. M. P., and K. O. Buesseler. 2012. A new method for the estimation of sinking particle fluxes from measurements of the particle size distribution, average

sinking velocity, and carbon content. Limnol. Oceanogr. Methods 10: 329–346.

---

## Referee Comment (RC3) · Anonymous Referee #3 · 16 Apr 2016

Bishop and coworkers describe a novel technological development (carbon Flux Explorer, CFE) that promises to revolutionize research on the biogeochemical cycling of carbon within the ocean thermocline, while also offering new insights into the factors that control the spatial and temporal patterns of the flux of particulate organic carbon (POC) exported from the euphotic zone. This publication is timely in that the CFE will be a valuable asset to the emerging EXPORTS program (Siegel et al., 2016. Prediction of the Export and Fate of Global Ocean Net Primary Production: The EXPORTS Science Plan. Frontiers in Marine Science, 3, doi 10.3389/fmars.2016.00022).

Fluxes and transformation of carbon in the ocean thermocline have long been a topic of discussion and debate, and the need for more quantitative characterization of these fluxes, and of the processes that regulate them, has been recommended in a number of community planning docu-

ments (e.g., OCTET http://www.somas.stonybrook.edu/1999/10/18/octet/ and OCCC http://www.us-ocb.org/documents/occc_is_2004.pdf). Motivating factors include both the desire to understand the supply of nutrition that fuels the mesopelagic ecosystem and the need to characterize the environmental factors within the thermocline that control the fraction of carbon exported from the euphotic zone that ultimately reaches abyssal depths before being regenerated, where it may be sequestered from the atmosphere for centuries to millennia. The CFE will enable investigators to characterize and quantify these POC fluxes with unprecedented accuracy and resolution, both spatial and temporal.

The manuscript is well written, and it could be published with only modest revision, as already described by Referees 1 and 2. However, in the spirit of exploring multiple working hypotheses, I would ask the authors to consider an alternative interpretation of their principal scientific finding described in the paper.

Bishop et al. interpret the high flux of POC in January, a time of low surface biomass and low POC concentration, to reflect the rapid loss of POC by grazing and export. By contrast, they interpret the opposite end-member condition of high surface biomass and low export, in May, to reflect the much greater efficiency of biological recycling (consumption and regeneration) of POC in surface waters. This interpretation is plausible, and I don't necessary disagree, but I wonder if the authors can rule out the following alternative interpretation.

Specifically, could the contrasting conditions observed in January and May reflect variable storage of POC in surface waters which, in turn, is regulated by physical aggregation and sedimentation? As noted in the text (p.2, line 2 and p. 10 line 11) one generally thinks that turbulence increases particle coagulation by increasing the rate of particle-particle encounter. While this may be true for aggregation of small particles, turbulence may lead to fragmentation of fragile large aggregates. Indeed, this may explain the absence of large aggregates in the samples collected by the BUOY-OSR. Furthermore, although the weather conditions were characterized as "calm" for all deployments of the CFE, conditions were the most quiescent in January. Therefore, is it possible that ultra-quiescent conditions facilitate the physical aggregation of POC into particles large enough (marine snow) to be exported with much greater efficiency than for the fragmented pieces of marine snow? I have no evidence to suggest that this alternative hypothesis is preferable to the one offered by the authors. Rather, I simply suggest that the authors consider physical aggregation as an alternative hypothesis to account for the unexpected inverse relationship between surface ocean POC inventory and the flux of exported POC collected by the CFE.

Also, the authors speculate that larger size classes of organisms dominated the grazing during January. Can this be verified using collections of historical data available from some of the programs that have been monitoring the region for decades, such as CalCOFI, the California Current System LTER, or the Central and Northern California Ocean Observing System?

DETAILS and EDITORIAL COMMENTS in their order of appearance:

p. 2 line 13 "nearby" as one word

p. 5 line 5 delete "were"

p. 5 line 21 insert "with" between linearly and PIC.

p. 6 line 10 delete "a" after estimate.

p. 6 lines 18-20: Here the authors stress, appropriately, that the conversion to POC flux is based on very little observational evidence. I suggest that the authors add a new section to the Discussion with recommendations for future studies that would reduce the uncertainty in this conversion factor.

p. 6 line 21 delete the comma after "above".

p. 8 lines 9-12: Here the authors describe the unexpected finding that in some cases the PIC/POC ratio decreases with depth. This is unexpected because the paradigm

is that POC is regenerated much more rapidly than PIC. The authors attribute the PIC/POC decrease with depth to temporal variability of the PIC/POC production ratio. Could other (potentially more interesting biogeochemistry) factors be involved?

p. 8 line 23 insert "in January 2013" between collected and by.

p. 8 line 30 "lower" is misspelled.

p. 10 lines 20-26: Plot the Martin curve on Figure 6C to provide readers with a visual illustration of the difference between Martin's export attenuation (b value) and the b values derived in this study.

p. 11 line 9 change "high" to "higher"

Fig 1 caption: insert "place" after "deployments took"

Fig 4 caption: explain the small circles, similar to the explanation offered in the caption of Fig. 5.

Fig 9: placing a white vertical bar between the two images will make it easier for readers to compare the figure with the caption.

---

## Author Comment (AC1) · 20 Apr 2016

*Below we provide a response to Reviewer 1 in Italics.*

¶Following the abstract and introduction, the methods were described in great detail. Sometimes I feel that this is more a methodological manuscript. This can be seen in the high number of figures describing the methods. Even in the result and discussion sections, methodological aspects can be found (e.g. results: chapter 3.3., third paragraph, chapter 3.4...). I wonder whether and how the methods described here are different from the ones presented in earlier papers, e.g. in Bishop et al., 2004 ('Robotic observations ...). This should be clarified and then, the method sections could be shortened and the focus concentrated on the gathered data. If the methods differ, the changes/improvements could be briefly outlined. Some figures could also be moved to

the Appendix.

—— *Sorry for the confusion. The papers, Bishop et al. 2004, 2009... "Robotic Observations..." describe our first robot, the Carbon Explorer. The CE was comprised of a SOLO float with interfaced Transmissometer and scattering sensors. The primary mission of the CE was to measure particle concentration profiles (beam attenuation coefficient and turbidity), we also used the transmissometer to register particle accumulation on the upward looking optic as the float drifted at depth, prior to profiling we cleaned the upward looking optic and measured transmission again, thus enabling us to derive the Carbon Flux Index by difference. This is well documented in the referenced publications.*

*The current paper describes an entirely new robot, the Carbon Flux Explorer, it was necessary to describe in detail our methodology for the Optical Sedimentation Recorder (OSR) as it was deployed both on the CFE and as a heavier than water package tethered below surface floatation. This is the BUOY-OSR system.*

*The operational details of both CFE and BUOY-OSR systems had to be documented in detail as this is the first paper and because the findings from the two systems were so different. Also described are key environmental data (ADCP results in particular) used for identifying the cause of the differences between the CFE and BUOY-OSR observations. We think that it is therefore appropriate to retain methodology as is and to not relocate figures to the appendix.*

¶My major concern, however, is the comparison between optically- and trap-derived (surface tethered and moored traps) fluxes. Both approaches are fundamentally different and comparisons are limited by seasonal, interannual and regional variability.

(1) It would be helpful to have more information on the locations of the different types of sediment traps used for comparison (e.g. in Fig. 1) and the exact seasons/years and the water depths. In chapter 4.1. the CFE fluxes from the Santa Cruz Basin were compared to fluxes from the nearby Santa Barbara and San Pedro Basins which were derived from different water depths and different sediment trap types.

(2) Flux data from Martin et al. (1987) were captured from more open ocean sites in the far north of the Pacific in different years rather than from a more local basin. To facilitate reading and to follow the presented argumentation and evaluate this comparison more critically, the authors should show a table with optical and sediment trap fluxes but with detailed information on locations, seasons, years, water depths, surface or bottom-tethered arrays with traps, type of traps and openings.

*—— (1) and (2)We will add to figure 1 – locations of the two studies compared (see attached graphic which shows a larger view of the southern California bight) and modify the text to describe the periods of observations and methods used in more detail. The methodology for the comparative studies is well documented in Thunell 1998 (Parflux Trap 50 mab at 540 m; August 1993- August 1996 at 2 week sample resolution), and Collins et al. 2011 (McLane Mark V sediment traps Jan. 2004 through Dec. 2007, weekly resolution; 550m and 800 m. Shallow traps (PITs - 24 hour deployments, various months 2005-2008). We believe that adding a detailed tabulation of already published data is not needed.*

¶(3) Further, the authors cannot explain the large differences (10-20 times) of fluxes in a satisfactory way. It is hard to believe that the CFE's optical sedimentation recorded (OSR) with 1 cm opening was able to better sample large marine snow aggregates of several mm size (which are rather rare in the water column) than a cylindrical tube of about a decimeter or larger in size of a free-drifting (surface tethered) sediment trap. I would expect this to be the other way round. It is hard to believe that the baffles of sedimentation tubes are so small to destroy fragile marine snow aggregates of a few mm in size (page 11, uppermost chapter). By the way, the opening of the tube of the OSR (1 cm) is not larger than the cm-sized trap baffle openings (page 11 lines 6-9).

*—— The reviewer stated "Obviously, the traps [BUOY-OSR] under sample larger marine snow aggregates of the size class larger than 1mm". The OSR on both CFE and Buoy-OSR systems is identical. There may have been some confusion.*

*The analysis of particle size distributions in the images and the analysis of ADCP results in-*

*dicate that the large aggregates are likely not destroyed by the BUOY-OSR baffles, but simply bounce off the baffles back into the flow; during the bounce, some fragments of the aggregates get into the trap. We think the text is quite clear on this.*

*We agree that PIT traps have almost the same baffle opening as used with the OSR. PITs have been deployed similarly to our BUOY-OSR. We would expect that similar biases would be found. Stukel's work with PIT traps appears to confirm this. During the rewrite of the article we will look to clarify the text.*

¶Sediment traps have their limitations as well, depending for instance on the type of array (surface vs bottom tethered). In particular, bottom-fixed moorings with shallow water sediment traps seem to be critical to record fluxes in the upper few hundred meters of the water column. Shallow traps may not provide an accurate measure of particle fluxes and differ by a factor of 3-10 (Buesseler 1991).

*——The 3-10 range of Buesseler 1991 is based on the 234Th method combined with uncertainties of assumptions regarding the C:Th ratio of particulates.*

¶Part of the discrepancies discussed here may be attributed to these uncertainties in trap-derived fluxes (e.g. page 9, chapter 4.1. of discussion).

*—— We deployed identical OSRs on a Lagrangian platform and surface-tethered. We feel that the manifestation of hydrodynamically forced undersampling of mm sized aggregates by the BUOY-OSR is very clear.*

*In addition, we do discuss factors leading to uncertainty of our POC flux estimates in the methods. These are the best estimates we can make. Bishop et al. 1978 paper is the most closely applicable. Please see our response to Reviewer 2 in the separate reply.*

¶Particle fluxes from sediment traps (NBST, surface tethered, moored) on the other hand, measure carbon fluxes more directly and apply less assumptions than the optical methods. There are less than a factor of two differences of fluxes between NBST and surface tethered traps as the authors mention at the end of page 10.

––– *The finding of "factor of two differences" of NBST and surface tethered PIT traps has been based on work performed in oligotropic waters near Bermuda. While Stanley et al., show that POC numbers are about a factor of two different, the PIC/POC ratios vary by a factor of four.*

*Comparisons have not been done in the coastal environment.*

*Also the flume studies (summarized in Buesseler et al. 2007) on efficiency of traps experiencing horizontal currents utilized clay and silt sized particle suspensions. While the trap size was scaled down in models used, the baffle openings were cm sized in the models with baffles. Consequently, the scale of the particle size to baffle opening was small. In our case, the aggregate size is the same magnitude as the baffle opening.*

*The flume studies also pointed out the importance of controlling trap tilt. Hence our effort to achieve near zero tilt on the OSR systems deployed tethered to the surface.*

*The present study took place in a coastal environment where large aggregates are abundant, as we mention >97 percent of the flux was carried by aggregates > 1.5 mm in size in January. Thus the factor of 20 difference found in Jan. 2013 is inescapable. Size analysis suggests a minimum factor of 3 bias in the other seasons.*

*Knauer, Martin, and Bruland (1979) say, "Each tube had an inside diameter of 7.39 cm and was equipped with a baffle system (Soutar et al., 1977) that consisted of 16 smaller tubes (length 7.6cm). The top ends of the baffle tubes had been milled to a wall thickness of 0.06 mm to minimize surface area (about 5 percent of the cylinder mouth area which is 43 cm2). We assume that materials hitting these edges fall into the collectors and contribute to the total flux. GARDNER (1977) has shown that open cylinders with a length-to-width ratio of approximately 2 or greater will yield representative fluxes. With our use of a baffle system, an adequate length-to-width ratio (8.4) and density gradients (see below [in their paper]), we assume that our traps sample the vertical flux" of particulate matter with reasonable accuracy. We also have 210Pb data (see below) supporting our assumptions. However, like other investigators attempting to measure vertical fluxes, we presently have no way of definitely knowing whether our supposition is correct."*

*What Knauer et al. (1979) did not anticipate is the possibility of large aggregates bouncing back into the current flowing horizontally over the trap.*

*We feel that the discussion is more than adequate on explaining why and how the differences between BUOY-OSR and CFE arise.*

¶Summarizing this, I suggest to be more careful with this kind of comparison and the conclusions. The problems associated with this comparison of fluxes (optical vs traps) should be clearly mentioned in the discussion.

—— *We feel that we have been as careful as we can in the discussion. We had hardly anticipated the result. We will look to see if the text can be clarified further*

¶Minor issues

(A) page 1 , line 28:... by grazers or [add] settle down as larger marine snow particles.

—— *will correct*

(B) The optical methods to estimate carbon fluxes which are described here need several assumptions, e.g. conversation factors (chapter 2.3.). What are the errors of the individual methods and the potential cumulative errors? Is there any estimation/ quantification? Something written in earlier papers?

—— *we have done all that can be done regarding a discussion of uncertainties in the derivation of our attenuance – POC conversion factor. Further work on calibration at sea is scheduled for August 2016.*

- page 5, chapter 2.2.4. Do the authors only hypothesize that attenuance is the best proxy for POC? What is the basis for this assumption, please clarify

—— *The logic is explained in the text. Transmissometer beam attenuation coefficient is a well documented and best proxy for POC concentration; it is superior to scattering based methods which are subject more to refractive index effects. By analogy we hypothesize that Volume Attenuance flux (as opposed to darkfield counts) is a proxy for POC flux.*

(C) page 8, chapter 3.3., sometimes hard to read due to many abbreviations.

—— *we will look at this*

(D) Fig. 1, show surroundings of the Santa Cruz Basin to provide more info on the general setting of the study site and the other sites used for comparison

*Will do. See attached figure as an example of added detail for fig 1.*
* * *
*Interactive comment on Biogeosciences Discuss., doi:10.5194/bg-2016-62, 2016.*

[Figure]

Data SIO, NOAA, U.S. Navy, NGA, GEBCO
Image Landsat
Data MBARI
Data LDEO-Columbia, NSF, NOAA

**Fig. 1.** Added figure element for Fig 1.

---

## Author Comment (AC2) · 20 Apr 2016

—— *and italicized text indicates author response to the review.*

Anonymous Referee 2

¶General comments: This manuscript describes the first use of a powerful new observational technology. Pending some revisions detailed below, it will make an important contribution to the literature on methods for observing the ocean's biological carbon pump. These data are the first of their kind, and if the CFE technology continues to be robust in the field, these types of measurements could revolutionize our understanding of the ocean biological pump.

¶Aside from the new method, this paper's scientific findings include high sinking POC

flux under apparent low-productivity conditions in an upwelling, coastal system, and undersampling of sinking POC by surface-tethered relative to neutrally-buoyant sediment traps.

¶Most of the specific comments below pertain to instances where strong, rather general statements are not fully supported by the data as presented. In some cases, the data just need to be clarified or more technical detail added to the text. In other cases, more nuance should be added to the interpretations.

¶A theme through several of the comments below is that more attention should be paid to spatial variability in the measured C fluxes, particularly in comparisons of in situ observations to satellite data. In general, the necessary changes should be straightforward for the authors to address.

—— *The conditions encountered are well documented with both remote sensing and CTD data.*

Specific comments

¶(1) p. 2, L. 4. The specificity of the sentence about foraminifera shells, etc. suggests that perhaps a reference is necessary.

*Be et al. Armstrong et al.*

¶(2) p. 2, L. 36. Can the CFE really operate for years? It would be more useful to most readers to point out the demonstrated length of deployment so far (months?) or give the nominal number of profile cycles that can generally be achieved.

—— *The CFEs have had field deployments lasting as long as 40 days with no ill effects. We have demonstrated in the lab that they can operate at hourly frequency for 8 months. By extension, 16 months at 2 hour frequency. . . etc. The CFE is independently powered and thus has no impact on the profiling lifetime of the float.*

*Bishop, J.K.B. (2012) Autonomous Exploration of Sedimentation Dynamics in the California Current System. Proc. 15th Biennial Challenger Conference for Marine Sciences. 3-6 Sept*

*2012. University of East Anglia, Norwich, UK. p36.*

¶(3) Methods section. I suggest you tabulate the cruise numbers, deployment locations/times/depths, and retrieval locations/times/depths to help the reader keep track of the different observations.

—— *will add this to supplemental materials).*

¶(4) Section 2.1. It may be a good idea to move the basic trap funnel and stage dimensions from Appendix A to here. Otherwise, you don't state the collection area anywhere. I noticed that another reviewer has assumed the OSR has a 1 cm diameter trap opening, which is not the case and may have led to some misunderstanding.

—— *will clarify the text although the Appendix is formally part of the paper.*

¶(5) P. 3, L. 20. Use "thickness" instead of "length" to describe the vertical dimension of the baffle.

—— *we will look at this. Given that the OSR is shown in Appendix 1, the aspect ratio of the baffle will be obvious*

¶(6) P. 3 L. 28-30. If any of the OSR data are transmitted, they must be pre-processed on board prior to doing so, correct? If this is something that has been implemented, a brief description of the on-board processing steps should be given here. Otherwise, please remove "OSR data" from the list of things that are transmitted during surfacing cycles.

—— *We say that images are recorded on board the CFE. We will implement processing onboard the CFE at the level described here. The OSR data transmitted include engineering status and image metadata. We will clarify the text.*

¶(7) P. 3 L. 29. State the actual dive depth rather than "considerably deeper than planned."

—- *Will state depths were 300-400 m deeper than planned in May 2012.*

¶(8) P. 4, L. 12-13: While it is ultimately the author's call, I suggest a change from base-10 to base-e. I believe that this is the convention for most of the optical oceanography community when describing optical properties.

—– *We have explained our rationale for this. 2.303\*log10 = ln.*

¶(9) P. 4, L. 20 (and related discussion in Section 2.3): How have you dealt with particles that overlap? Would stepwise subtraction lead to a possible underestimation of flux?

—– *Overlapping particles are additive in attenuance units as explained in the text. The only complexity is determination of particle size distributions. We provide the series of images as supplemental materials that were used in the estimation of cumulative attenuance and POC conversion factor.*

¶(10) P. 4, L. 30: Is there a poster or meeting presentation which has shown the elimination of the stress polarization interference, which you can refer to here? If not, this sentence should be reworded to indicate that the problem is surmountable, but the discussion of later CFE builds should be removed as they are not relevant to the data presented here.

—– *We will retain this discussion as it is something we have demonstrated through deployment experience.*

¶(11) Section 2.2.3 and P. 7 L. 7-8, and supplemental videos: Presumably swimmer interference should also affect analysis of the POC flux data, not just the PIC flux. Please revise this discussion so that it applies to both proxies. The supplemental videos clearly show several instances of "disappearing/moving particles" (e.g., halfway through 1204 and maybe again at the end; also about halfway through 1301). Unlike direct sediment traps, you can actually detect and correct for the presence of swimmers with the OSR, and I think this should be discussed in more detail.

—– *We plan to implement processing codes that detect movement. This is outside of the scope*

*of the current paper. The interference by swimmers was minor in Attenuance units.*

¶(12) P. 5, L. 21: Please add a reference for the statement that birefringence scales linearly with PIC concentration.

—– *We did (Guay and Bishop, 2002). The two points about birefringence are related.*

¶(13) P. 6, L. 6-16: Please add an equation or two summarizing the calculation you have described here in words. It will greatly clarify the procedure.

—– *The text is clear. We do not think an equation will simplify the text.*

¶(14) P. 6, L. 18-28: It would be illustrative to also compute fluxes using models in the literature for aggregate carbon content as a function of size. For instance, Alldredge (1998) contains useful relationships for several categories of marine snow. At the very least, you should mention the existence of such models and their relevance to interpretation of image data such as that collected by the OSR.

—– *Thank you for pointing out this reference. We will add discussion. Alldredge (1998) collected marine snow by scuba at depths of 10 to 20 m. If we use our 2d analyses of min and max dimension to compute ESV following Alldredge (1998) we get volumes that are 8.5 times higher. If we used their conversion of ESV to POC/aggregate and sum for particles > 1 mm in size then, the POC estimate falls 17 times lower than our estimate. Their estimate yields a POC density for marine snow equal to 0.00020 g/cm3; in our case numbers are 0.05 g/cm3. The carbon density of aggregates derived from their equations differs by a factor of 250.*

*I'm not sure whether or not there is a units error in Alldredge (1998) or if the identified marine snow particles sampled in the euphotic zone are mostly empty of material. There are no images of collected marine snow in the 1988 paper. Ours are clearly loaded. Bishop et al. (1978) worked with aggregates sampled from 100-400 m. We feel that these data are more reasonable. Obviously, there is a need for further work on calibration. We will add some discussion of Alldredge (1988) to the paper.*

¶(15) P. 6, L. 33-34 and Figure 1. Add a distance scale bar to Fig. 1c.

*—– yes*

You state that all satellite data points used for comparison were within a 2km radius of 33.72N, 119.5W, but that appears to be the center of a "150 km2" study region, and does not correspond to the actual surfacing locations of the CFE or optics cast locations. If Chl and POC were patchy, then the changes seen in the surface optical properties at a fixed point in the center of the box are unlikely to correspond with the observations on a quasi-Lagrangian platform drifting tens of kilometers from this point. Comparisons to satellite data should attempt to match up with the actual locations of the CFE tracks. Once this is done, please adjust the text accordingly.

*—– 150 km2 is the rectangular area of the study region. The actual area is smaller. This is an effort that will not change the interpretation of the data. With 3 surfacings of the CFE per day and only one satellite image per day at best, a drifting matchup is not as simple as it seems. The single point reference is adequate.*

*We had provided imagery at 4 km (Modis aqua) as a supplemental document. We worked with 1 km results and report means and standard deviations. We have attached similar images from Mati Kahru (Scripps) with area of the Reference circle plotted. They are also shown at 4 km resolution. The big picture story of the imagery is that prior to the expedition, there were higher levels of Chlorophyll at the surface in the Santa Cruz Basin in general. From day to day they fluctuate. By the time the ship arrived, the Chlorophyll had decreased. Interestingly, a bloom developed in the Santa Barbara Basin by the time we had left. Animation of the imagery shows that there is little coherent structure. This is the best that we can do to document conditions.*

*Further analysis of patchiness is beyond the scope of this paper and our funding.*

¶(16) Section 3.2: How close in time and space were the different depths measured on each deployment? These details should be made clear in the text or in Figure 1. That is, it is possible the CFEs were sampling different sinking particle pools at each of the different depths? It may not be correct to assume a single attenuation model.

*—— There will always be a temporal / spatial effect seen in particle profiles. We have analyzed hydrographic data for evidence of intrusions and found none. The water column is influenced by tidal currents and there is a net westward drift.*

¶(17) p. 10 L. 11-14. You invoke certain physical conditions here in order to support the hypothesis that low-biomass conditions were caused by consumer-driven export and not by physical aggregation or advection.

However, the minimal wind and current shears you describe are inconsistent with your other major finding that your surface-tethered BUOY-OSR undersampled by a factor of 20 relative to the CFE due to strong hydrodynamic effects felt by the different platforms. Please reconcile these ideas.

*—— We described conditions minimal wind and swell at the surface. In January 2013, there is no evidence for frontal features in the area of our observations. The whole water column is influenced by semi-diurnal tidal currents varying from 0-20 cm/sec. It is also influenced by internal waves.*

*The difference in shear is that experienced by the BUOY-OSR vs, the Lagrangian CFE at depth. We have quantified the horizontal motions leading to the observed bias.*

¶(18) p. 10, L 16-17 and Figure 6: The trend is hard to pick out from Figure 6. You need to add an inset that shows the January 2013 period. Otherwise it looks like the 1-week-prior points are scattered, not necessarily decreasing, and in any case they cover up the running-mean line so it cannot be seen.

*—— will review this and improve the figure.*

¶(19) P. 10 L. 18-19. "Satellite imagery from Jan 2013 shows a patchy POC/chlorophyll distribution without obvious eddy structures or fronts near by." This statement is not possible to evaluate from the satellite chlorophyll images included in the Supplementary Information. It certainly appears as though there are potential eddy structures and fronts in these images. However, the images do not have latitude and longitude

marked, there is no color bar (is it log or linear? What are the scale limits?) nor are the CFE deployment locations marked. These images need to be clearly annotated so it is obvious that they support the claim that there were no nearby eddy structures or fronts. Otherwise it is not possible to differentiate a rapid temporal change in POC from a rapid spatial change, and this assertion should be removed.

—— *see our response to point (15) above.*

¶(20) P. 10, L. 22-23: What are the uncertainties on your derived "Martin" b values? Are they even significant? (If not, add a statement to this effect – it adds strength to your finding that the Martin curve is an inappropriate model for these data). However, you should also mention again the time and distance separations among the different depth measurements – if export was patchy, then it could be that each sampled depth is too far from the others to infer a continuous attenuation profile.

—— *The data are presented as clearly as can be. The depth occupations of the CFE occur approximately 7 hours apart as described in the methods. The time series is not long enough to average out day/night effects; thus there may be a temporal effect. In early versions of the manuscript, Martin curves were not drawn. Reviewers requested this discussion. Reviewer 3 requests contrast with the typical curve. We will do this. We will provide a table with times and positions of CFE transmissions and BUOY-OSR positions. The figures graphically compare time series. We feel that a lot of additional discussion is unwarranted.*

¶(21) Section 4.1: The difference between surface tethered and neutrally buoyant traps may be more pronounced in the presence of large aggregates such as the ones you have observed here. The studies you cite comparing PITS and NBST traps were conducted in an oligotrophic region where in situ camera profiles showed low concentrations of particles larger than 1500 um (McDonnell and Buesseler, 2012). Your findings in coastal California are quite striking, but there may be site-specific differences in the relative efficiencies of tethered/neutrally-buoyant traps at collecting aggregates. Please revise lines 34-36 to address the differences between the different types of

environments.

*——— I looked at the text and don't understand what the reviewer wants us to say beyond the fact that the factor of two differences are found in oligotrophic waters. The text seems clear but I will look at it again and modify if it can be improved. That said, Stanley et al. found a factor of four difference in PIC/POC ratio. So surface tethered traps are collecting a different quality of particle than NBSTs even in oligotrophic waters.*

¶(22) P. 11, L. 5-6: Near-horizontal approach of particles to tethered traps has been described in detail by Siegel et al. 2008; I suggest you include a citation to this reference.

*——— we see no reason to add the reference for particle trajectories. This has been a point of discussion since Gardner's 1977 PhD thesis.*

¶(23) P. 11 L. 21: Replace "the single profile 234Th/238U method" with "the 234Th/238U method with a steady-state assumption", which is clearer.

*——— The text seems clear as written. Will review.*

¶(24) P. 11 L. 23-24: Similarly, replace "time series sampling" (which is less specific) with something like, "multiple reoccupations of a water parcel assuming non steady-state conditions".

*——— the text seems clear as written. Will review.*

¶(25) P. 11 L. 22: Replace "is not applicable" with "may not be applicable". If it can be established that a coastal system is in steady state and advection is minimal, then the steady-state assumption can be used.

*——— the text seems clear as written. Will review. The conditions satisfying steady state the Reviewer indicates are unlikely to occur in any coastal environment.*

¶(26) Figure 9: Please put a thin margin between these panels.

*—– will do.*

¶(27) Movies in Supplement. Do these represent multiple depths and profile cycles?

*—– yes. Imaging logic described in the methods.*

¶(28) Can you make this information clearer in the "readme" file?

*—– yes. However imaging logic is described in the methods.*

¶(29) If there are multiple cycles represented in the videos, please insert "marker" frames so it's clear where the breaks are.

*—– the video is provided as an example of CFE deployment results. They are there primarily to contrast the kinds of particles encountered by CFE in the three seasons. A second purpose to document the contrast of CFE and BUOY-OSR collections in January. We don't have funding to do any more with movie production at this time.*

¶(30) Also, the movies occasionally show "disappearing" particles. Are these zooplankton? How are they treated in the flux estimation calculation? (see also the comment on section 2.2.3)

*—– Yes the ones that move around and/or disappear are swimmers. We describe interferences in the PIC records due to barnacle larvae. The effects are relatively small in the attenuance data. We mention cases where corrections have been made. We are developing codes to detect movement and remove the contributions of these relatively rare swimmers.*

Technical comments ¶(31) p. 1, L. 14, change "monitor" to "monitored"

*—– yes*

¶(32) p. 1, L. 19, Break into two sentences. Start 2nd sentence with "Multiple lines of evidence indicate :::"

*—– yes.*

¶(33) p. 1., L. 19. Remove space from "under sampling"

—— *yes.*

¶(34) p. 1, L. 20. Change "compared to" to "than the"

—— *yes.*

¶(35 36) p. 2, L. 3. "coccoliths" should be singular p. 2 L. 13, change "near by" to "nearby"

—— *yes yes.*

¶(37) P. 2 L. 13, should be "strong, recent weakening"

—— *yes.*

¶(38) P. 2 L. 18. It is unnecessary to abbreviate Eppley and Peterson 1979 because you only cite it once more.

—— *the abbreviation works as written.*

¶(39) p.2 L.34. Insert "that" before "we have developed".

—— *yes.*

¶(40) p.3 L. 4. "gain detail of the" is awkwardly-worded.

—— *will clarify.*

¶(41) p. 8 L. 5. "artifact" is misspelled.

—— *will correct.*

¶(40) P. 10 L. 17: Should refer to Figure 6, not Figure 4.

—— *will correct.*

¶References:

Alldredge, A. 1998. The carbon, nitrogen and mass content of marine snow as a function of aggregate size. Deep Sea Research Part I: Oceanographic Research Papers 45: 529–541.

McDonnell, A. M. P., and K. O. Buesseler. 2012. A new method for the estimation of sinking particle fluxes from measurements of the particle size distribution, average sinking velocity, and carbon content. Limnol. Oceanogr. Methods 10: 329–346

——— end of response
* * *
[Figure]

4 Km Modis Aqua/Terra
And VIIRS processed data
From Mati Kahru
mkahru@ucsd.edu

http://spg.ucsd.edu/Satellite_Data/California_Current/

Chlorophyll-a

0.05  0.1   0.2   0.4   1   2   4  6 8

**Fig. 1.** Imagery from Mati Kahru (Scripps) January 2013

---

## Author Comment (AC3) · 20 Apr 2016

—— *author response in italics.*

Anonymous Referee 3

¶Bishop and coworkers describe a novel technological development (carbon Flux Explorer, CFE) that promises to revolutionize research on the biogeochemical cycling of carbon within the ocean thermocline, while also offering new insights into the factors that control the spatial and temporal patterns of the flux of particulate organic carbon (POC) exported from the euphotic zone. This publication is timely in that the CFE will be a valuable asset to the emerging EXPORTS program (Siegel et al., 2016. Prediction of the Export and Fate of Global Ocean Net Primary Production: The EXPORTS Science Plan. Frontiers in Marine Science, 3, doi 10.3389/fmars.2016.00022).

[Figure]

**BGD**

Fluxes and transformation of carbon in the ocean thermocline have long been a topic of discussion and debate, and the need for more quantitative characterization of these fluxes, and of the processes that regulate them, has been recommended in a number of community planning documents: e.g. OCTET http://www.somas.stonybrook.edu/1999/10/18/octet/ and OCCC http://www.us-ocb.org/documents/occc$_i s_2 004.pdf$.

Motivating factors include both the desire to understand the supply of nutrition that fuels the mesopelagic ecosystem and the need to characterize the environmental factors within the thermocline that control the fraction of carbon exported from the euphotic zone that ultimately reaches abyssal depths before being regenerated, where it may be sequestered from the atmosphere for centuries to millennia. The CFE will enable investigators to characterize and quantify these POC fluxes with unprecedented accuracy and resolution, both spatial and temporal.

The manuscript is well written, and it could be published with only modest revision, as already described by Referees 1 and 2.

¶(1) However, in the spirit of exploring multiple working hypotheses, I would ask the authors to consider an alternative interpretation of their principal scientific finding described in the paper. Bishop et al. interpret the high flux of POC in January, a time of low surface biomass and low POC concentration, to reflect the rapid loss of POC by grazing and export.

By contrast, they interpret the opposite end-member condition of high surface biomass and low export, in May, to reflect the much greater efficiency of biological recycling (consumption and regeneration) of POC in surface waters. This interpretation is plausible, and I don't necessary disagree, but I wonder if the authors can rule out the following alternative interpretation.

Specifically, could the contrasting conditions observed in January and May reflect variable storage of POC in surface waters which, in turn, is regulated by physical aggrega-

tion and sedimentation?

—— *The transmissometer profiles show that there is not an accumulation of stored POC in the water column at the time of the January expedition. We don't know anything about DOC pools, however as the study lacked these measurements.*

¶(2) As noted in the text (p.2, line 2 and p. 10 line 11) one generally thinks that turbulence increases particle coagulation by increasing the rate of particle-particle encounter. While this may be true for aggregation of small particles, turbulence may lead to fragmentation of fragile large aggregates. Indeed, this may explain the absence of large aggregates in the samples collected by the BUOY-OSR.

—— *The aggregates we encountered in January looked structurally robust. (See 5 supplemental images and animations of imagery). The particles arrive on the sample stage and do not dissaggregate. That said, there are cases during the March 2013 and May 2012 deployments where aggregates arrive and then fall apart over the period of two hours. I will clarify the text and see if I can add some discussion. Everything about the observations suggests that baffle bounce is the major reason that the bias is observed.*

¶(3) Furthermore, although the weather conditions were characterized as "calm" for all deployments of the CFE, conditions were the most quiescent in January. Therefore, is it possible that ultra-quiescent conditions facilitate the physical aggregation of POC into particles large enough (marine snow) to be exported with much greater efficiency than for the fragmented pieces of marine snow?

I have no evidence to suggest that this alternative hypothesis is preferable to the one offered by the authors. Rather, I simply suggest that the authors consider physical aggregation as an alternative hypothesis to account for the unexpected inverse relationship between surface ocean POC inventory and the flux of exported POC collected by the CFE.

—— *I think the only way this will be resolved is with comprehensive process studies and physical*

*characterization of the turbulence levels present.*

¶(4) Also, the authors speculate that larger size classes of organisms dominated the grazing during January. Can this be verified using collections of historical data available from some of the programs that have been monitoring the region for decades, such as CalCOFI, the California Current System LTER, or the Central and Northern California Ocean Observing System?

––– *The big surprise for us was the finding of intense feeding activity of the large creatures (dolphins, seabirds, squid. . .) in the water column. It was remarkable.*

¶DETAILS and EDITORIAL COMMENTS in their order of appearance: (a) p. 2 line 13 "nearby" as one word

––– *yes.*

¶(b) p. 5 line 5 delete "were"

––– *yes.*

¶(c) p. 5 line 21 insert "with" between linearly and PIC.

––– *yes.*

¶(d) p. 6 line 10 delete "a" after estimate.

––– *yes.*

¶(e) p. 6 lines 18-20: Here the authors stress, appropriately, that the conversion to POC flux is based on very little observational evidence. I suggest that the authors add a new section to the Discussion with recommendations for future studies that would reduce the uncertainty in this conversion factor.

––– *yes, good suggestion will add recommendations. We are building a sample collecting CFE with expectation of deploying it in August 2016.*

¶(f) p. 6 line 21 delete the comma after "above".

[Figure]

*—— yes.*

¶(g) p. 8 lines 9-12: Here the authors describe the unexpected finding that in some cases the PIC/POC ratio decreases with depth. This is unexpected because the paradigm is that POC is regenerated much more rapidly than PIC. The authors attribute the PIC/POC decrease with depth to temporal variability of the PIC/POC production ratio. Could other (potentially more interesting biogeochemistry) factors be involved?

*—— yes. In the Pacific, we have seen evidence of relatively fast remineralization of PIC – shallower than the carbonate saturation horizon (Bishop and Wood, 2008).*

¶(g) p. 8 line 23 insert "in January 2013" between collected and by.

*—— yes.*

¶(h) p. 8 line 30 "lower" is misspelled.

*—— yes.*

¶(i) p. 10 lines 20-26: Plot the Martin curve on Figure 6C to provide readers with a visual illustration of the difference between Martin's export attenuation (b value) and the b values derived in this study.

*—— yes.*

¶(j) p. 11 line 9 change "high" to "higher"

*—— yes.*

¶(k) Fig 1 caption: insert "place" after "deployments took"

*—— yes.*

¶(l) Fig 4 caption: explain the small circles, similar to the explanation offered in the caption of Fig. 5.

*—— yes.*

¶(m) Fig 9: placing a white vertical bar between the two images will make it easier for readers to compare the figure with the caption.

*––– yes.*

*End of review response.*

––––––––––––––––––––––––––––––––

*Interactive comment on Biogeosciences Discuss., doi:10.5194/bg-2016-62, 2016.*

---

## Author Response (AR1)

**Author response and line references to changed text.**

I wonder whether and how the methods described here are different from the ones presented in earlier papers, e.g. in Bishop et al., 2004 ('Robotic observations ...). This should be clarified and then, the method sections could be shortened and the focus concentrated on the gathered data. If the methods differ, the changes/improvements could be briefly outlined. Some figures could also be moved to the appendix.

— Sorry for the confusion. The papers, Bishop et al. 2004, 2009... "Robotic Observations..." describe our first robot, the Carbon Explorer. The CE was comprised of a SOLO float with interfaced Transmissometer and scattering sensors. The primary mission of the CE was to measure particle concentration profiles (beam attenuation coefficient and turbidity), we also used the transmissometer to register particle accumulation on the upward looking optic as the float drifted at depth, prior to profiling we cleaned the upward looking optic and measured transmission again, thus enabling us to derive the Carbon Flux Index by difference. This is well documented in the referenced publications.

**[p3 LINES 7-14 rewitten to clarify]**

The current paper describes an entirely new robot, the Carbon Flux Explorer, it was necessary to describe in detail our methodology for the Optical Sedimentation Recorder (OSR) as it was deployed both on the CFE and as a heavier than water package tethered below surface floatation. This is the BUOY-OSR system. The operational details of both CFE and BUOY-OSR systems had to be documented in detail as this is the first paper and because the findings from the two systems were so different. Also described are key environmental data (ADCP results in particular) used for identifying the cause of the differences between the CFE and BUOY-OSR observations. We think that it is therefore appropriate to retain methodology as is and to not relocate figures to the appendix.

**[no changes to text]**

My major concern, however, is the comparison between optically- and trap-derived (surface tethered and moored traps) fluxes. Both

approaches are fundamentally different and comparisons are limited by seasonal, interannual and regional variability.

(1) It would be helpful to have more information on the locations of the different types of sediment traps used for comparison (e.g. in Fig. 1) and the exact seasons/years and the water depths. In chapter 4.1. the CFE fluxes from the Santa Cruz Basin were compared to fluxes from the nearby Santa Barbara and San Pedro Basins which were derived from different water depths and different sediment trap types.

(2) Flux data from Martin et al. (1987) were captured from more open ocean sites in the far north of the Pacific in different years rather than from a more local basin.

To facilitate reading and to follow the presented argumentation and evaluate this comparison more critically, the authors should show a table with optical and sediment trap fluxes but with detailed information on locations, seasons, years, water depths, surface or bottom-tethered arrays with traps, type of traps and openings.

— (1) and (2). We will add to figure 1 – locations of the two studies compared (see attached graphic which shows a larger view of the southern California bight) and modify the text to describe the periods of observations and methods used in more detail. The methodology for the comparative studies is well documented in Thunell 1998 (Parflux Trap 50 mab at 540 m; August 1993-August 1996 at 2 week sample resolution), and Collins et al. 2011 (McLane Mark V sediment traps Jan. 2004 through Dec. 2007, weekly resolution; 550m and 800 m. Shallow traps (PITs - 24 hour deployments, various months 2005-2008).

**[Modified Figure 1 to show deployment locations for other studies, Text p 10 lines 20-21 modified to include dates]**

**We believe that adding a detailed tabulation of already published data is not needed**

(3) Further, the authors cannot explain the large differences (10-20 times) of fluxes in a satisfactory way. It is hard to believe that the CFE's optical sedimentation recorded (OSR) with 1 cm opening was able to better sample large marine snow aggregates of several mm size (which

are rather rare in the water column) than a cylindrical tube of about a decimeter or larger in size of a free-drifting (surface tethered) sediment trap. I would expect this to be the other way round. It is hard to believe that the baffles of sedimentation tubes are so small to destroy fragile marine snow aggregates of a few mm in size (page 11, uppermost chapter). By the way, the opening of the tube of the OSR (1 cm) is not larger than the cm-sized trap baffle openings (page 11 lines 6-9).

The reviewer stated "Obviously, the traps [BUOY-OSR] under sample larger marine snow aggregates of the size class larger than 1mm". The OSR on both CFE and Buoy-OSR systems is identical.

The analysis of particle size distributions in the images and the analysis of ADCP results indicate that the large aggregates are likely not destroyed by the BUOY-OSR baffles, but simply bounce off the baffles back into the flow; during the bounce, some fragments of the aggregates get into the trap.

We think the text is quite clear on this.

The PIT traps have almost the same baffle opening, and have been deployed similarly to our BUOY-OSR, thus we would expect that similar biases may hold.

During the rewrite of the article we will look to clarify the text.

**[Clarified text p 9. Lines 1-5]**

\sf

Sediment traps have their limitations as well, depending for instance on the type of array (surface vs bottom tethered). In particular, bottom-fixed moorings with shallow water sediment traps seem to be critical to record fluxes in the upper few hundred meters of the water column. Shallow traps may not provide an accurate measure of particle fluxes and differ by a factor of 3-10 (Buesseler 1991).

*The 3-10 range of Buesseler 1991 is based on the* 234*Th method combined with assumptions regarding the C:Th ratio of particulates.*

[No changes to text]

Part of the discrepancies discussed here may be attributed to these uncertainties in trap-derived fluxes (e.g. page 9, chapter 4.1. of discussion).

We deployed identical OSRs on a Lagrangian platform and surface-tethered. We feel that the hydrodynamic bias we have seen is very clear.

We tried to discuss factors leading to uncertainty of our POC flux estimates in the methods – these are the best estimates we can make. Bishop et al. 1978 paper is the most closely applicable. Please see our response to Reviewer 2 in the separate reply.

**[clarified text]**

Particle fluxes from sediment traps (NBST, surface tethered, moored) on the other hand, measure carbon fluxes more directly and apply less assumptions than the optical methods. There are less than a factor of two differences of fluxes between NBST and surface tethered traps as the authors mention at the end of page 10.

The reviewer is correct regarding the factor of two differences; however, the comparisons of NBST and surface tethered PIT traps have been mostly done in waters near Bermuda. Comparisons have not been done in the coastal environment. While Stanley et al., show that POC numbers are about a factor of two different, the PIC/POC ratios vary by a factor of four. Hydrodynamics or surface tethered traps can bias in both directions.

Furthermore, the early flume studies (e.g. Garner 1977) on the performance of traps utilized clay and silt sized particle suspensions; while the trap geometry was scaled down, the baffle openings were cm sized in models with baffles. The scale of the particle size to baffle opening scale was tiny. In our case, the aggregate size is the same magnitude as the baffle opening.

The flume studies also pointed out the importance of controlling trap tilt. Hence our effort to achieve near zero tilt on the OSR systems deployed tethered to the surface.

The present study took place in a coastal environment when large aggregates are abundant, as we mention >97% of the flux was carried by aggregates > 1.5mm in size in January. Thus the factor of 20 difference found in Jan. 2013 is inescapable. Size analysis suggests a factor of 3 bias in the other seasons. I'd like to repeat a quotation from Knauer, Martin, and Bruland. 1979

"Each tube had an inside diameter of 7.39 cm and was equipped with a baffle system (SOUTAR et al., 1977) that consisted of 16 smaller tubes (length 7.6cm). The top ends of the baffle tubes had been milled to a wall thickness of 0.06 mm to minimize surface area (about 5% of the cylinder mouth area which is 43 cm 2). We assume that materials hitting these edges fall into the collectors and contribute to the total flux. GARDNER (1977) has shown that open cylinders with a length-to-width ratio of approximately 2 or greater will yield representative fluxes. With our use of a baffle system, an adequate length-to-width ratio (8.4) and density gradients (see below), **we assume** that our traps sample the vertical flux" of particulate matter with reasonable accuracy. We also have 210pb data (see below) supporting our assumptions. However, like other investigators attempting to measure vertical fluxes, we presently have no way of definitely knowing whether our supposition is correct."

What Knauer et al. (1979) did not anticipate is the possibility of large aggregates bouncing back into the current flowing by the trap.

We feel that the discussion is more than adequate as is on this topic..

**[No modification of Text, we could add comments above] \sf**

Summarizing this, I suggest to be more careful with this kind of comparison and the conclusions. The problems associated with this comparison of fluxes (optical vs traps) should be clearly mentioned in the discussion.

*{We feel that we have been as careful as we can in the discussion. We had hardly anticipated the result. We will look to see if the text can be clarified further}*

**[No change in text beyond minor word changes]**

Minor issues (A) page 1 , line 28:... by grazers or settle down as larger marine snow particles.

**[corrected p 1 line 29]**

(B) The optical methods to estimate carbon fluxes which are described here need several assumptions, e.g. conversation factors (chapter 2.3.). What are the errors of the individual methods and the potential cumulative errors? Is there any estimation/quantification? Something written in earlier papers?

{we have done all that can be done regarding a discussion of uncertainties in the derivation of our attenuance – POC conversion factor. Further work on calibration at sea is scheduled for August 2016}.

[In response to R1 and R2, We added discussion to Sec 2.3. p 6 lines 26-37. Alldridge (1998) describes carbon content – volume relationships for fresh Marine Snow sampled in the euphotic zone. Summary P 13 lines 7-9.]

- page 5, chapter 2.2.4. Do the authors only hypothesize that attenuance is the best proxy for POC? What is the basis for this assumption, please clarify

{The logic is explained in the text. Transmissometer beam attenuation coefficient is a well documented and best proxy for POC concentration; it is superior to scattering based methods which are subject more to refractive index effects. By analogy we hypothesize that Volume Attenuance flux (as opposed to darkfield counts) is a proxy for POC flux.}

**[no changes]**

(C) page 8, chapter 3.3., sometimes hard to read due to many abbreviations *{we will look at this}*

**[p 9 1-5 modiified for clarity]**

(D) Fig. 1, show surroundings of the Santa Cruz Basin to provide more info on the general setting of the study site and the other sites used for comparison *{will do}*

**[Done]**

Anonymous Referee #2 Received and published: 13 April 2016

Reviewer comment in black. *Our original Response in blue;* [Our action taken and line references in green.]

Specific comments

(1) p. 2, L. 4. The specificity of the sentence about foraminifera shells, etc. suggests that perhaps a reference is necessary.

Be et al. Armstrong et al.

**[already referenced]**

(2) p. 2, L. 36. Can the CFE really operate for years? It would be more useful to most readers to point out the demonstrated length of deployment so far (months?) or give the nominal number of profile cycles that can generally be achieved.

----- The CFEs have had field deployments lasting as long as 40 days with no ill effects. We have demonstrated in the lab that they can operate at hourly frequency for 8 months. By extension, 16 months at 2 hour frequency... etc. The CFE is independently powered and thus has no impact on the profiling lifetime of the float.

**[Added text at p 13 lines 12-17]**

Bishop, J.K.B. (2012) Autonomous Exploration of Sedimentation Dynamics in the California Current System. Proc. 15th Biennial Challenger Conference for Marine Sciences. 3-6 Sept 2012. University of East Anglia, Norwich, UK. p36.

(3) Methods section. I suggest you tabulate the cruise numbers, deployment locations/times/depths, and retrieval locations/times/depths to help the reader keep track of the different observations.

----- will add this to supplemental materials.

[Added Table 1.]

(4) Section 2.1. It may be a good idea to move the basic trap funnel and stage dimensions from Appendix A to here. Otherwise, you don't state the collection area anywhere. I noticed that another reviewer has assumed the OSR has a 1 cm diameter trap opening, which is not the case and may have led to some misunderstanding.

----- will clarify the text although the Appendix is formally part of the paper.

**[OSR funnel and stage diameter information added p 3 line 29]**

(5) P. 3, L. 20. Use "thickness" instead of "length" to describe the vertical dimension of the baffle.

----- we will look at this. Given that the OSR is shown in Appendix 1, the aspect ratio of the baffle will be obvious

**[no change]**

(6) P. 3 L. 28-30. If any of the OSR data are transmitted, they must be pre-processed on board prior to doing so, correct? If this is something that has been implemented, a brief description of the on-board processing steps should be given here. Otherwise, please remove "OSR data" from the list of things that are transmitted during surfacing cycles.

----- We say that images are recorded on board the CFE. We will implement processing onboard the CFE at the level described here. The OSR data transmitted include engineering status and image metadata. We will clarify the text.

**[p 3 line 35 and p 4 lines 2&3]**

(7) P. 3 L. 29. State the actual dive depth rather than "considerably deeper than planned."

---- Will state depths were 300-400 m deeper than planned in May 2012.

**[p4 line 2]**

(8) P. 4, L. 12-13: While it is ultimately the author's call, I suggest a change from base-10 to base-e. I believe that this is the convention for

most of the optical oceanography community when describing optical properties.

----- We have explained our rationale for this. 2.303 \* log10 = ln.

**[p 4 line 20-21]**

(9) P. 4, L. 20 (and related discussion in Section 2.3): How have you dealt with particles that overlap? Would stepwise subtraction lead to a possible underestimation of flux?

----- Overlapping particles are additive in attenuance units as explained in the text. The only complexity is determination of particle size distributions. We provide the series of images as supplemental materials that were used in the estimation of cumulative attenuance and POC conversion factor.

**[no change to text]**

(10) P. 4, L. 30: Is there a poster or meeting presentation which has shown the elimination of the stress polarization interference, which you can refer to here? If not, this sentence should be reworded to indicate that the problem is surmountable, but the discussion of later CFE builds should be removed as they are not relevant to the data presented here.

----- We will retain this discussion as it is something we have demonstrated through deployment experience.

**[text remains unchanged. This is simply a fact]**

(11) Section 2.2.3 and P. 7 L. 7-8, and supplemental videos: Presumably swimmer interference should also affect analysis of the POC flux data, not just the PIC flux. Please revise this discussion so that it applies to both proxies. The supplemental videos clearly show several instances of "disappearing/moving particles" (e.g., halfway through 1204 and maybe again at the end; also about halfway through 1301). Unlike direct sediment traps, you can actually detect and correct for the presence of swimmers with the OSR, and I think this should be discussed in more detail. ----- We plan to implement processing codes that detect movement. This is outside of the scope of the current paper. The interference by swimmers was minor in Attenuance units.

**[Added line p 5 18-19]**

\P (12) P. 5, L. 21: Please add a reference for the statement that birefringence scales linearly with PIC concentration.

----- We did (Guay and Bishop, 2002). The two points about birefringence are related.

**[No change]**

(13) P. 6, L. 6-16: Please add an equation or two summarizing the calculation you have described here in words. It will greatly clarify the procedure.

----- The text is clear. We do not think an equation will simplify the text.

**[No change]**

(14) P. 6, L. 18-28: It would be illustrative to also compute fluxes using models in the literature for aggregate carbon content as a function of size. For instance, Alldredge (1998) contains useful relationships for several categories of marine snow. At the very least, you should mention the existence of such models and their relevance to interpretation of image data such as that collected by the OSR.

----- Thank you for pointing out this reference. We will add discussion. Alldredge (1998) collected marine snow by scuba at depths of 10 to 20 m. If we use our 2d analyses of min and max dimension to compute ESV following Alldredge (1998) we get volumes that are 8.5 times higher. If we used their conversion of ESV to POC/aggregate and sum for particles > 1 mm in size then, the POC estimate falls 17 times lower than our estimate. Their estimate yields a POC density for marine snow equal to 0.00020 g/cm3; in our case numbers are ~0.03 g/cm3. The carbon content derived from their equations differs by a factor of 150. [changed 0.05 to 0.03 as used OM% and not C %]

I'm not sure whether or not there is a units error in Alldredge (1998) or if the identified marine snow particles sampled in the euphotic zone are mostly empty

of material. There are no images of collected marine snow in the 1988 paper. Ours are clearly loaded. Bishop et al. (1978) worked with aggregates sampled from 100-400 m. We feel that these data are more reasonable. Obviously, there is a need for further work on calibration. We will add some discussion of Alldredge (1988) to the paper.

**[Added Discussion p 6. lines 30 to p 7 line 2]**

(15) P. 6, L. 33-34 and Figure 1. Add a distance scale bar to Fig. 1c.

----- Yes

**[done]**

You state that all satellite data points used for comparison were within a 2km radius of 33.72N, 119.5W, but that appears to be the center of a "150 km2" study region, and does not correspond to the actual surfacing locations of the CFE or optics cast locations. If ChI and POC were patchy, then the changes seen in the surface optical properties at a fixed point in the center of the box are unlikely to correspond with the observations on a quasi-Lagrangian platform drifting tens of kilometers from this point. Comparisons to satellite data should attempt to match up with the actual locations of the CFE tracks. Once this is done, please adjust the text accordingly.

 $-150 \text{ km}^2$  is the rectangular area of the study region. The actual area is smaller. This is an effort that will not change the interpretation of the data. With 3 surfacings of the CFE per day and only one satellite image per day at best, a drifting matchup is not as simple as it seems. The single point reference is adequate. We had provided imagery at 4 km (Modis aqua) as a supplemental document. We worked with 1 km results and report means and standard deviations. We have attached similar images from Mati Kahru (Scripps) with area of the Reference circle plotted. They are also shown at 4 km resolution. The big picture story of the imagery is that prior to the expedition, there were higher levels of Chlorophyll at the surface in the Santa Cruz Basin in general. From day to day they fluctuate. By the time the ship arrived, the Chlorophyll had decreased. Interestingly, a bloom developed in the Santa Barbara Basin by the time we had left. Animation of the imagery shows that there is little coherent structure. This is the best that we can do to document conditions. Further analysis of patchiness is beyond the scope of this paper and our funding.

**[Added text p7 lines 20-21; and added Figure A1.3 to describe spatial gradients of Chlorophyll a in the study area].**

(16) Section 3.2: How close in time and space were the different depths measured on each deployment? These details should be made clear in the text or in Figure 1. That is, it is possible the CFEs were sampling different sinking particle pools at each of the different depths? It may not be correct to assume a single attenuation model.

----- There will always be a temporal / spatial effect seen in particle profiles. We have analyzed hydrographic data for evidence of intrusions and found none. The water column is influenced by tidal currents and there is a net westward drift.

**[The distance of separation stated in the caption for Fig 1. Spatial gradients of chlorophyll were less than 20% over this distance in almost all cases and all expeditions. Even though the CFE and BUOY-OSR were separated, CFE data at the same location as the BUOY-OSR one day earlier showed no difference in flux (Fig. 8)]**

(17) p. 10 L. 11-14. You invoke certain physical conditions here in order to support the hypothesis that low-biomass conditions were caused by consumer-driven export and not by physical aggregation or advection.

However, the minimal wind and current shears you describe are inconsistent with your other major finding that your surface-tethered BUOY-OSR undersampled by a factor of 20 relative to the CFE due to strong hydrodynamic effects felt by the different platforms. Please reconcile these ideas.

----- We described conditions minimal wind and swell at the surface. In January 2013, there is no evidence for frontal features in the area of our observations. The whole water column is influenced by semi-diurnal tidal currents varying from 0-20 cm/sec. It is also influenced by internal waves.

The difference in shear is that experienced by the BUOY-OSR vs, the Lagrangian CFE at depth. We have quantified the horizontal motions leading to the observed bias.

**[The text is clear on meaning, no change]**

(18) p. 10, L 16-17 and Figure 6: The trend is hard to pick out from Figure 6. You need to add an inset that shows the January 2013 period. Otherwise it looks like the 1-week-prior points are scattered, not necessarily decreasing, and in any case they cover up the runningmean line so it cannot be seen.

----- will review this and improve the figure.

**[Added section A1.3 and Fig. A3 to the Appendix]**

(19) P. 10 L. 18-19. "Satellite imagery from Jan 2013 shows a patchy POC/chlorophyll distribution without obvious eddy structures or fronts near by." This statement is not possible to evaluate from the satellite chlorophyll images included in the Supplementary Information. It certainly appears as though there are potential eddy structures and fronts in these images. However, the images do not have latitude and longitude marked, there is no color bar (is it log or linear? What are the scale limits?) nor are the CFE deployment locations marked. These images need to be clearly annotated so it is obvious that they support the claim that there were no nearby eddy structures or fronts. Otherwise it is not possible to differentiate a rapid temporal change in POC from a rapid spatial change, and this assertion should be removed.

**----- see our response to point (15) above.**

**[no further text modifications]**

(20) P. 10, L. 22-23: What are the uncertainties on your derived "Martin" b values? Are they even significant? (If not, add a statement to this effect – it adds strength to your finding that the Martin curve is an inappropriate model for these data). However, you should also mention again the time and distance separations among the different depth measurements – if export was patchy, then it could be that each sampled depth is too far from the others to infer a continuous attenuation profile.

— The data are presented as clearly as can be. The depth occupations of the CFE occur approximately 7 hours apart as described in the methods. The time series is not long enough to average out day/night effects; thus there may be a

temporal effect. In early versions of the manuscript, Martin curves were not drawn. Reviewers requested this discussion. Reviewer 3 requests contrast with the typical curve. We will do this. We will provide a table with times and positions of CFE transmissions and BUOY-OSR positions. The figures graphically compare time series. We feel that a lot of additional discussion is unwarranted.

**[we added text p 11. lines 14-16; Table 1(locations) and Table 3]**

(21) Section 4.1: The difference between surface tethered and neutrally buoyant traps may be more pronounced in the presence of large aggregates such as the ones you have observed here. The studies you cite comparing PITS and NBST traps were conducted in an oligotrophic region where in situ camera profiles showed low concentrations of particles larger than 1500 um (McDonnell and Buesseler, 2012). Your findings in coastal California are quite striking, but there may be site-specific differences in the relative efficiencies of tethered/neutrally-buoyant traps at collecting aggregates. Please revise lines 34-36 to address the differences between the different types of environments.

----- I looked at the text and don't understand what this reviewer wants us to say beyond the fact that the factor of two differences are found in oligotrophic waters. The text seems clear but I will look at it again and modify if it can be improved. That said, Stanley et al. found a factor of four difference in PIC/POC ratio. So surface tethered traps are collecting a different quality of particle than NBSTs even in oligotrophic waters.

**[No changes to the text; We caution against assuming > 1mm sized particles are not important components of flux in oligotrophic waters; See e.g. particle size distributions at Bermuda (in Bishop et al., 2012)]**

(22) P. 11, L. 5-6: Near-horizontal approach of particles to tethered traps has been described in detail by Siegel et al. 2008; I suggest you include a citation to this reference.

----- we see no reason to add the reference for particle trajectories. This has been a point of discussion since Garner's 1977 thesis.

**[no further text modifications]**

(23) P. 11 L. 21: Replace "the single profile 234Th/238U method" with "the 234Th/238U method with a steady-state assumption", which is clearer.

----- The text seems clear as written. Will review.

[no further text modifications]

(24) P. 11 L. 23-24: Similarly, replace "time series sampling" (which is less specific) with something like, "multiple reoccupations of a water parcel assuming non steady-state conditions".

----- The text seems clear as written. Will review.

[page 12, line 18; "of a water parcel"]

(25) P. 11 L. 22: Replace "is not applicable" with "may not be applicable". If it can be established that a coastal system is in steady state and advection is minimal, then the steady-state assumption can be used.

----- The text seems clear as written. Will review. The conditions satisfying steady state the reviewer indicates are unlikely to occur in any coastal environment.

[no further text modifications]

(26) Figure 9: Please put a thin margin between these panels.

----- *will do*.

**[modified Figure 9 with line]**

(27) Movies in Supplement. Do these represent multiple depths and profile cycles?

----- yes. Imaging logic described in the methods.

**[no further modifications]**

(28) Can you make this information clearer in the "readme" file?

----- *yes*

**[will modify on final document upload]**

(29) If there are multiple cycles represented in the videos, please insert "marker" frames so it's clear where the breaks are.

— the video is provided as an example of CFE deployment results. They are there primarily to contrast the kinds of particles encountered by CFE in the three seasons. A second purpose is to document the contrast of CFE and BUOY-OSR collections in January. We don't have funding to do any more with movie production at this time.

**[no further modifications]**

(30) Also, the movies occasionally show "disappearing" particles. Are these zooplankton? How are they treated in the flux estimation calculation? (see also the comment on section 2.2.3)

— Yes the ones that move around and/or disappear are swimmers. We describe interferences in the PIC records due to barnacle larvae. The effects are relatively small in the attenuance data. We mention cases where corrections have been made. We are developing codes to detect movement and remove the contributions of these relatively rare swimmers.

**[no further modifications beyond our reply to comment 11 above ]**

Technical comments (31) p. 1, L. 14, change "monitor" to "monitored"

----- yes [Done]

(32) p. 1, L. 19, Break into two sentences. Start 2nd sentence with "Multiple lines of evidence indicate ..."

----- yes [Done]

(33) p. 1., L. 19. Remove space from "under sampling"

**----- yes [Done]**

(34) p. 1, L. 20. Change "compared to" to "than the"

----- yes [Done]

(35 & 36) p. 2, L. 3. "coccoliths" should be singular p. 2 L. 13, change "near by" to "nearby"

----- yes [Done]

(37) P. 2 L. 13, should be "strong, recent weakening"

----- yes [Done]

(38) P. 2 L. 18. It is unnecessary to abbreviate Eppley and Peterson 1979 because you only cite it once more.

----- the abbreviation works as written.

**[No change]**

(39) p.2 L.34. Insert "that" before "we have developed".

----- yes [Done]

(40) p.3 L. 4. "gain detail of the" is awkwardly-worded.

----- yes [Done]

(41) p. 8 L. 5. "artifact" is misspelled.

----- yes [Done]

(40) P. 10 L. 17: Should refer to Figure 6, not Figure 4.

----- will correct. [Done]

Anonymous Referee #3 Received and published: 16 April 2016

(1) However, in the spirit of exploring multiple working hypotheses, I would ask the authors to consider an alternative interpretation of their principal scientific finding described in the paper. Bishop et al. interpret the high flux of POC in January, a time of low surface biomass and low POC concentration, to reflect the rapid loss of POC by grazing and export.

By contrast, they interpret the opposite end-member condition of high surface biomass and low export, in May, to reflect the much greater efficiency of biological recycling (consumption and regeneration) of POC in surface waters. This interpretation is plausible, and I don't necessary disagree, but I wonder if the authors can rule out the following alternative interpretation.

Specifically, could the contrasting conditions observed in January and May reflect variable storage of POC in surface waters which, in turn, is regulated by physical aggregation and sedimentation?

— The transmissometer profiles show that there is not an accumulation of stored POC in the water column at the time of the January expedition. We don't know anything about DOC pools, however as the study lacked these measurements.

**[no changes to document]**

(2) As noted in the text (p.2, line 2 and p. 10 line 11) one generally thinks that turbulence increases particle coagulation by increasing the rate of particle-particle encounter. While this may be true for aggregation of small particles, turbulence may lead to fragmentation of fragile large aggregates. Indeed, this may explain the absence of large aggregates in the samples collected by the BUOY-OSR.

----- The aggregates we encountered in January looked structurally robust. (See 5 supplemental images and animations of imagery). The particles arrive on the sample stage and do not disaggregate. That said, there are cases during the March 2013 and May 2012 deployments where aggregates arrive and then fall apart over the period of two hours. I will clarify the text and see if I can add some discussion. Everything about the observations suggests that baffle bounce is the major reason that the bias is observed.

**[page 11 line 4 most consistent with...]**

Furthermore, although the weather conditions were characterized as "calm" for all deployments of the CFE, conditions were the most quiescent in January. Therefore, is it possible that ultra-quiescent conditions facilitate the physical aggregation of POC into particles large enough (marine snow) to be exported with much greater efficiency than for the fragmented pieces of marine snow?

I have no evidence to suggest that this alternative hypothesis is preferable to the one offered by the authors. Rather, I simply suggest that the authors consider physical aggregation as an alternative hypothesis to account for the unexpected inverse relationship between surface ocean POC inventory and the flux of exported POC collected by the CFE.

**----- I think the only way this will be resolved is with serious process studies and physical characterization of the turbulence levels present.**

Also, the authors speculate that larger size classes of organisms dominated the grazing during January. Can this be verified using collections of historical data available from some of the programs that have been monitoring the region for decades, such as CalCOFI, the California Current System LTER, or the Central and Northern California Ocean Observing System?

----- I think the biggest surprise was the finding of intense feeding activity of the large creatures (dolphins, seabirds, squid...) in the water column. It was remarkable.

[Investigation of CalCOFI - CCS LTER data sets is a future activity]

DETAILS and EDITORIAL COMMENTS in their order of appearance: (a) p. 2 line 13 "nearby" as one word ----- yes. [Done]

(b) p. 5 line 5 delete "were"

----- yes. [Done]

(c) p. 5 line 21 insert "with" between linearly and PIC.

----- yes. [Done]

(d) p. 6 line 10 delete "a" after estimate.

----- yes. [Done]

(e) p. 6 lines 18-20: Here the authors stress, appropriately, that the conversion to POC flux is based on very little observational evidence. I suggest that the authors add a new section to the Discussion with recommendations for future studies that would reduce the uncertainty in this conversion factor.

----- yes, good suggestion will add recommendations. We are building a sample collecting CFE with expectation of deploying it in August 2016.

Page 13 – lines 6-10.

(f) p. 6 line 21 delete the comma after "above".

----- yes. [Done]

(g) p. 8 lines 9-12: Here the authors describe the unexpected finding that in some cases the PIC/POC ratio decreases with depth. This is unexpected because the paradigm is that POC is regenerated much more rapidly than PIC. The authors attribute the PIC/POC decrease with depth to temporal variability of the PIC/POC production ratio. Could other (potentially more interesting biogeochemistry) factors be involved?

----- yes. In the Pacific, we have seen evidence of relatively fast remineralization of PIC – shallower than the carbonate saturation horizon (Bishop and Wood, 2008). [Added p 11 line 24-25].

(g) p. 8 line 23 insert "in January 2013" between collected and by.

----- yes. [Done]

(h) p. 8 line 30 "lower" is misspelled.

----- yes. [Done]

(i) p. 10 lines 20-26: Plot the Martin curve on Figure 6C to provide readers with a visual illustration of the difference between Martin's export attenuation (b value) and the b values derived in this study.

**----- yes. [Done]**

(j) p. 11 line 9 change "high" to "higher"

**----- yes. [Done]**

(k) Fig 1 caption: insert "place" after "deployments took"

**----- yes. [Done]**

(I) Fig 4 caption: explain the small circles, similar to the explanation offered in the caption of Fig. 5.

**----- yes. [Done]**

(m) Fig 9: placing a white vertical bar between the two images will make it easier for readers to compare the figure with the caption.

----- yes. [Done]

End of review response.

**Robotic observations of high wintertime carbon export in California coastal waters**

J. K. B. Bishop1,2, M. B. Fong3, T. J. Wood2

1Department of Earth and Planetary Science, UC Berkeley, Berkeley, CA 94720, USA. 2Earth Sciences Division, Lawrence Berkeley National Laboratory, Berkeley, CA, 94720, USA 3Department of Chemistry, Scripps Institution of Oceanography, La Jolla, CA, USA.

Correspondence to: J. K. B. Bishop (jkbishop@berkeley.edu)

**Abstract.**

5

Biologically mediated particulate organic and inorganic carbon (POC and PIC) export from surface waters is the
principal determinant of the vertical oceanic distribution of pH and dissolved inorganic carbon and thus sets the conditions for air sea exchange of CO2; exported organic matter also provides the energy fuelling communities in the mesopelagic zone. However, observations are temporally and spatially sparse. Here we report first hourly-resolved optically-quantified POC and PIC sedimentation rate time series from autonomous Lagrangian Carbon Flux
Explorers (CFEs), which monitored particle flux using an imaging optical sedimentation recorder (OSR) at depths
below 140 m in the Santa Cruz Basin, CA in May 2012, and in January and March 2013. Highest POC vertical flux (~100-240 mmol C m-2 d-1) occurred in January, when most settling material was mm to cm-sized aggregates, but when surface biomass was low; fluxes were ~18 and ~6 mmol C m-2 d-1, respectively in March and May, under high surface biomass conditions. An unexpected discovery was that January 2013 fluxes measured by CFE were 20 times

- higher than that measured by simultaneously deployed surface-tethered OSR; multiple lines of evidence indicate strong undersampling of aggregates larger than 1 mm in the later case. Furthermore, the Jan 2013 CFE fluxes were about 10 times higher than observed during multi year sediment trap observations in the nearby Santa Barbara and San Pedro Basins. The strength of carbon export in biologically dynamic California coastal waters is likely underestimated by at least a factor of 3 and at times by a factor of 20.
- 25 Key words. Biological Carbon Pump, Ocean Carbon Cycle, Twilight Zone Export, Coastal Sedimentation, Optical Sedimentation Recorder, Carbon Flux Explorer.

**1. Introduction**

30

Phytoplankton account for half of global net photosynthesis (Field et al., 1988), or about 50 Pg C y-1, yet they live for a week before being removed from the euphotic zone through grazing or abiotic aggregation processes. While most phytoplankton carbon is recycled in the surface layer, recent model and observation based estimates (Henson *et al.*, 2011; Yao and Schlitzer 2013; Siegel *et al.*, 2014) suggest that globally 5 to 12 Pg C y-1 is exported below the euphotic zone as sinking particulate organic and inorganic carbon (POC and PIC) in fecal pellets, amorphous large aggregates, and as independently sinking carapaces and calcareous shells (e.g. Bishop, Ketten, and Edmond, 1978; Alldredge and Silver 1988; Turner, 2015). Many large amorphous aggregate particles (e.g. greater in size than 0.5

| James Bishop 4/28/2016 9:31 AM        |
|---------------------------------------|
| Deleted: sediment traps               |
| James Bishop 4/28/2016 9:32 AM        |
| Deleted:                              |
| James Bishop 4/28/2016 9:34 AM        |
| Deleted: compared to highest previous |
| James Bishop 4/28/2016 8:40 PM        |
| Deleted: nearby                       |
| James Bishop 4/28/2016 9:32 AM        |
| Deleted: between                      |
|                                       |

James Bishop 4/28/2016 9:38 AM Deleted: consumed by grazers

[revised manuscript text omitted]

Bishop et al. (2004, 2009) and Estapa et al. (2013) have deployed transmissometer equipped profiling floats to observe the high-frequency variability of the systematics of sedimentation. When the floats yo-yo from depth to the surface, transmissometers record the concentration variability of particles in the water column. As the floats drift at

- 10 depth between profiles, there is a measurable transmission loss (or attenuance increase) as settled particles accumulate on the upward looking transmissometer window. These data, when normalized by the time at depth, yield a simple metric, or index, of sedimentation. In contrast, with direct high-resolution imaging of the settled particles, the CFE not only is able to quantify optical metrics of particle flux, but also is able to gain detail on kinds of particles and thus the specifics of the process governing carbon export.
- 15 Below we present observations from CFE development testing during three expeditions off the coast of southern California aboard R/V New Horizon: May 29-June 4 2012 (NH1204), Jan. 18-21 2013 (NH1301) and March 27-31 2013 (NH1304). These observations, like past ship-tended studies using neutrally buoyant and surface-tethered sediment traps, are of relatively short duration (days); however, they are the first carried out in a productive coastal environment. CFE data are analysed below in the framework of remotely sensed surface chlorophyll and POC, *in-situ*
- 20 water column optical properties, and subsurface currents. The study site is a 150 km2 region centred at 33° 43'N 119° 33'W in the 1900 m deep Santa Cruz Basin (SCB) near the California coast (Figs. 1B & C).

**2. Methods**

**2.1 Carbon Flux Explorer (CFE).**

The CFE is comprised of an Optical Sedimentation Recorder (OSR) interfaced to a CTD-equipped Sounding Oceanographic Lagrangian Observer (SOLO) float (Davis, Sherman and Dufor, 2001). The CFE dives repeatedly (daily in this study) below the surface to obtain OSR observations at three target depths. The CFE's Optical Sedimentation Recorder (OSR; Figure A1, Appendix Sec. 1.1) awakes when the CFE reaches a target depth. Particles settle through a hexagonal celled baffle (1 cm opening, 5 cm length) into a high-aspect titanium funnel and settling column before depositing on a 2.54 cm diameter glass sample stage; funnel diameter is 15.4 cm, Particles are imaged

in three lighting modes: transmitted, transmitted-cross-polarized, and dark field. On first wake up of a given dive, the sample stage is flushed with water and images of the particle-free stage are obtained. At timed intervals (~20 min in data described here) the OSR repeats image sets, which register the sequential build-up of particles. After the predetermined number of image sets over ~1.8 hours, cleaning occurs and a new reference image set is obtained. After ~5 hours at a target depth the OSR performs a final image/cleaning cycle and reference image set, and the SOLO
 surfaces to report GPS position, CTD profile data, and OSR engineering data, and dives to its next target depth.

3

James Bishop 4/28/2016 9:49 AM Deleted: . James Bishop 4/28/2016 9:50 AM Deleted:

James Bishop 4/28/2016 10:42 PM Deleted: W James Bishop 4/28/2016 10:43 PM Deleted: f James Bishop 4/28/2016 10:06 AM Deleted: the biological process of James Bishop 4/28/2016 10:16 AM Deleted: sedimentation, thus it extends significantly beyond the simple Carbon Flux Index (CFI) optical sedimentation proxy from transmissometer-equipped Lagrangian floats (Bishop et al. 2004, 2009; Estapa et al., 2013)

James Bishop 5/2/2016 12:27 PM Deleted:

Every 3 dive cycles, the depth order is repeated; in the current study, target depths were 150, 300 and 500m. A ballasting error in May 2012 led to the CFE operating 300 to 400m deeper than planned. In the CFE described here, all images were stored aboard the OSR for post recovery analysis.

**2.2 OSR Image data reduction.**

[revised manuscript text omitted]

- 15 A total of 127 aggregates from 500 µm to 8.3 mm in size were enumerated in all five images. A size cutoff of 800 µm was used to minimize contamination of the analysis due to the slight expansion of aggregate dimension over time as they rested on the stage. Although, size distributions were initially quantified to a lower limit of 20 µm, it is important to note that 97% of the volume attenuance of the sample was accounted for by particles >800 µm in this sample. For the remaining 63 particles larger than 800 µm, we calculated an aggregate volume of 0.113 cm3. To estimate the POC flux, we assume an aggregate dry weight
- density of 0.087 g cm3 (e.g. From Fig. 22 in Bishop et al. 1978) and thus calculate the weight of aggregate matter as 0.0098 g. 20 Organic matter at depths between 100 m and 500 m is typically 60% (range 50-70%) of dry weight (Fig. 11 in Bishop et al. 1978). Thus, organic matter weight is 0.0059g. We estimate POC (as carbon) by dividing this mass by the OM:C conversion factor, 1.88 from Hedges et al. (2002). We divide by 12.011, the atomic weight of carbon, to yield a POC loading of  $0.26 \times 10^{-3}$ moles. The opening of the OSR funnel has a diameter of 15.4 cm, which yields an assumed trap collection area of 186.3 cm2.
- 25 The time interval for collection of this sample was 1.84 hours, or 0.0766 days. Combining this information yields a carbon flux of 183 mmol m-2 d-1. The average POCATN flux for all particles >13  $\mu$ m in size during the same time interval was 66.2 mATN $cm^2 cm^2 d^{-1}$ . As mentioned above, over 97% of cumulative sample attenuance was in the >800  $\mu$ m size fraction. The conversion factor for POCATN flux to POC flux is the ratio of 183 to 66.2\*0.97 = 2.8. This conversion factor allows estimation of POC fluxes observed for compare with other data from near by waters.
- Alldredge (1998) collected marine snow by scuba at euphotic depths of 10 to 20 m in the nearby Santa Barbara Basin. They 30 imaged particles in the laboratory and derived maximum and minimum dimensions, and assumed an elliptical volume. From this they calculated equivalent spherical volume (ESV in mm3). Samples were analyzed for POC and a regression formula was derived (POC (µg) = 0.99\*ESV0.52). If we use the ImageJ 2-dimensional analyses of minimum and maximum aggregate dimensions in our images to compute equivalent spherical volume (ESV) following Alldredge's methods we get ESV values that
- are 8.5 times higher than obtained using our approach. If we use the Alldredge (1988) conversion of ESV to POC/aggregate and 35 sum over all particles >1 mm in size, then the amount of POC is 17 times lower than our estimate. The Alldredge method yields a POC density for aggregates equal to 0.00020 g C cm3 while our method yields ~0.028 g C cm3. The carbon density of aggregate particles derived from their equations differs from ours by a factor of 140. We are not sure if there is a unit error in Alldredge (1998), or if the marine snow particles sampled in shallow waters were mostly empty of material, or if they deflate as

6

James Bishop 4/28/2016 10:29 AM Deleted: a

ames Bishop 4/28/2016 9:04 PM Formatted: Superscript James Bishop 4/28/2016 10:44 PM Formatted: Font color: Text 1

James Bishop 4/28/2016 10:40 AM Formatted: Superscript James Bishop 4/28/2016 10:40 AM Formatted: Superscript

**they sink below the euphotic zone. Bishop et al. (1978) describe aggregates sampled from 100-400 m that appeared similarly loaded to those described here; for this reason believe that our method is internally most consistent.**

[revised manuscript text omitted]

**3.2 Particle Flux Profiles**

25

Averaged POCATN and PICPOL flux profiles provide systematic quantification of depth trends of sedimentation. In Jan. 2013, POCATN fluxes at depths 150, 300 and 500 m were 67, 85, and 35 mATN-cm2 cm-2 d-1, respectively; in Mar. 2013 at similar depths, flux was between 6 to 7 mATN-cm2 cm-2 d-1; values in May 2012 for the 450 - 900 m interval fell between 2 and 3 mATN-cm2 cm-2 d-1 with no depth trend. PICPOL flux trends roughly followed those for

- $POC_{ATN}$  flux. PICPOL fluxes and at 150, 300 and 500 m depths were 96, 122, and 63 ppm- cm2 d-1 (Jan. 2013) and 33, 15, and 16 ppm-cm2 cm-2 d-1 (Mar. 2013); The apparent decrease of PIC flux with depth from 2.8 to 0.5 ppm-cm2 d-1 in May 2012 may be a consequence of the low flux condition and high scatter of points.
- For both POCATN and PICPOL flux, it is also notable that the 1.8 hour averaged fluxes for each depth ranged over a 30 factor of 10 relative to their average for the May and March periods, but ranged by less than a factor of two in the high flux period January 2013. This suggests that sedimentation is more uniform when large aggregates dominate. The profile of PICPOL/POCATN ratio (Fig. 7) shows a depth dependent increase from 1.4 to 2.4 in January, indicating preferential loss of POC with sinking. In March, PICPOL/POCATN ratios slightly decreased with depth from 4.6 to 3.7. In May 2012, PIC/POC ratio decreased from 1.5 to 0.3 between 430 and 900 m. The fact that the PIC/POC can 35

decrease with depth suggests preferential carbonate dissolution can and does occur in shallow waters, a finding

8

mes Bishop 4/28/2016 Deleted: A

James Bishop 4/28/2016 10:48 PM Deleted: )

James Bishop 4/28/2016 9:31 PM Deleted: supplemental

lames Bishop 4/28/2016 Deleted: an artefact

consistent with Bishop and Wood (2008). These observations confirm that the relative contribution of carbonates to aggregate ballasting varies with time.

**3.3 Comparison of Surface Tethered Optical Sedimentation Recorder (OSR) flux with CFE Flux**

In January 2013, an OSR system, identical to the OSR on the CFE, was deployed at 237 m tethered below a buoy system at the surface (Appendix, Section 1.4, Fig. A4). This BUOY-OSR system was engineered to collect samples needed to calibrate the POCATN and PICPOL data; however, this approach was a failure. The POCATN time series from CFE and BUOY-OSR are compared in Figure 8. The BUOY-OSR yielded POCATN and PICATN fluxes of 3.6 mATN- $cm^2 cm^{-2} d^{-1}$  and 2.2 ppm- $cm^2 cm^{-2} d^{-1}$ , respectively – approximately 25 fold and 50 fold lower than CFE fluxes at 144 and 320 m depths. At the same time, BUOY-OSR and CFE images showed that the two systems collected aggregates

10 of completely different morphology and size distribution (Fig. 9). The CFE images showed abundant 5 mm- to cmsized marine snow aggregates. The BUOY-OSR collected only fragments of these aggregates and few particles larger than 2 mm.

Figure 10 shows cumulative volume attenuance size distributions corresponding to samples collected in January 2013

- by the CFE and BUOY-OSR. We selected CFE images for depths 144 m (Fig. 7), 320, and 507 m. We also selected the BUOY-OSR image corresponding to Fig. 7 and another taken at the exact same time as the first CFE image.
- Other BUOY-OSR images were analysed for the rest of the time series. CFE images were either the first or second image after stage cleaning since subsequent images had multiple cases of particle overlap. BUOY-OSR images analysed were from just before stage cleaning since no particles were overlapping. Size distribution analysis was performed using ImageJ software, with an attenuance threshold >0.02, and a size threshold of at least two pixels for a
- 20 particle. Particles were classified according to their equivalent circular diameter (ECD), with the lower size limit of 20 μm. Volume attenuance for each particle was computed (the product of particle area and attenuance). Results were summed cumulatively from largest to smallest particle size and then normalized by total volume attenuance (Table 2; Fig. 10). In January 2013, >96 % of cumulative volume attenuance (CVA) was found in the >1000 μm size range in CFE data at all depths. In contrast, BUOY-OSR results typically show less than half of the cumulative
- 25 volume attenuance in aggregates larger than 1000 μm; this is consistent with the particles being fragments of the larger aggregates and consistent with the factor of 20 difference in observed attenuance flux. If we were to eliminate all aggregates larger than 1500 μm from January CFE data, then there is close agreement with the CVA size-distributions derived from BUOY-OSR data. During the period when current flow across the BUOY-OSR fell below 2 cm s-1, analysis of the BUOY-OSR image shows that nearly 50% of the aggregate volume attenuance was in the
- 2000-5000 μm, i.e. large, size range. In other words, the surface tethered OSR began sampling larger aggregates.
   Although, BUOY-OSR and CFE were operating at different depths, and separated by ~9 km in our study area (Fig. 1), there was no indication in CTD/optics profiles that there was any discontinuity or advective feature at 237 m that would lead to biasing the BUOY-OSR results low. Nor was there any indication in remote sensing imagery (Supplemental materials) of nearby fronts or eddy structures. Furthermore, spatial chlorophyll variability was less

9

35 than 20% in our study area (Appendix, Section 1.3).

| James Bishop 4/28/2016 11:35 AM              |
|----------------------------------------------|
| Deleted: sample-collecting                   |
| James Bishop 4/28/2016 11:37 AM              |
| Deleted: ,                                   |
| James Bishop 4/28/2016 11:36 AM              |
| Deleted: with the same                       |
| James Bishop 4/28/2016 11:36 AM              |
| Deleted: model as the one in                 |
| James Bishop 4/28/2016 11:37 AM              |
| Deleted: ,                                   |
| James Bishop 4/28/2016 11:37 AM              |
| Deleted: free-drifting                       |
| James Bishop 4/28/2016 5:31 PM               |
| Deleted: 3                                   |
| James Bishop 4/28/2016 5:31 PM               |
| Deleted: 3                                   |
| James Bishop 4/29/2016 9:05 AM               |
| Deleted: to gain samples of particulates for |
| comparison with                              |
| James Bishop 4/29/2016 9:06 AM               |
| Deleted:                                     |
| James Bishop 4/28/2016 2:00 PM               |
| Formatted: Subscript                         |
| James Bishop 4/29/2016 9:08 AM               |
|                                              |
| James Bisnop 4/28/2016 2:02 PM               |
| Deleted: of volume attenuance                |
| James Bisnop 4/29/2016 9:11 AM               |
| Deleted: Two cases are contrasted. First, w  |
| James Bishop 4/28/2016 11:43 AM              |
|                                              |
| James Bishop 4/28/2016 10:55 PM              |
| Deleted: 1                                   |
| James Bishop 4/28/2016 2:04 PM               |
| Deleted: for                                 |
| James Bishop 4/28/2016 2:04 PM               |
| Deleted:                                     |

**James Bishop 4/28/2016 2:50 PM Deleted: at slightly different locations**

James Bishop 4/28/2016 1:05 PM Deleted: James Bishop 4/28/2016 3:34 PM Deleted: Figure 11, shows, CFE CVA – size distributions, from all three expeditions compared with January BUOY-OSR data and further demonstrates that the CFE captured a greater loading of large particles than the BUOY-OSR, All lines of evidence indicate a major bias against aggregates larger than millimetre in size by the surface-tethered BUOY-OSR.

**3.4 POC flux estimates**

[revised manuscript text omitted]

   Satellite imagery from Jan 2013 (Supplemental materials) shows a patchy POC/chlorophyll distribution without obvious eddy structures or fronts nearby.

Our observations show that POCATN flux profiles (Fig. 6) attenuate with depth, albeit not following the classic Martin

- 15 et al. (1987) curve fit ( $\Phi z = \Phi_{\underline{zref}}/(Z|\underline{Zref})^b$ , Z is depth,  $\underline{Zref} = 100 \text{ m}$  and  $\Phi$  is flux, and b = 0.858). Using light based euphotic depths of 50, 20 and 25 m for January, March, and May as the  $\underline{Zref}$  values for particle remineralisation (Buesseler and Boyd (2009), we derive Martin's 'b' factors of 0.36, 0.22, and 0.17, respectively, far lower than the accepted b value (Table 3). There are multiple limitations to this exercise; these include likely temporal variation of flux at a particular depth with time of day (see. e.g. Fig 15 in Bishop et al. 1987), depths sampled being 100 m deeper
- 20 than the base of the euphotic zone, and the limited time of the study. This said, it is clear that in January 2013, the export may have been high enough to overwhelm the capacity of detritus feeders to reduce the flux. In May and March, all evidence is consistent with high recycling of the sinking POC in waters shallower than our 400 and 125 m observation points, respectively.

The fact that the PIC/POC ratio can decrease with depth suggests that carbonate dissolution may occur in shallow

[revised manuscript text omitted]

James Bishop 4/28/2016 11:35 PM

James Bishop 4/28/2016 11:36 PM **Deleted:** evidence

James Bishop 5/2/2016 5:39 PM Deleted:

sediment traps in waters close to our study site. Analysis of the images of accumulated particles suggest that surfacetethered sediment traps have under-sampled a major class of large aggregates contributing to carbon export from surface waters. At this writing, it is premature to revise the carbon export attributable to coastal waters up by a factor of 20 as we do not yet know the year-round contribution of large aggregates to sedimentation. In other seasons, our

POC flux estimates are closer - but still elevated by greater than a factor of two compared with moored traps, and 5 likely a minimum of a factor of three based on discussion of size frequency distribution results presented in Sec. 4.1. Missing contributions of large aggregates in sediment trap collections may explain the reported imbalance between POC flux to the seafloor and benthic respiratory demand seen at nearby station M (34° 50'N, 123° 00'W; Smith et al., 2013). We also show that particulate fluxes estimated using remotely sensed surface biomass may be overestimated 10

sometimes - or underestimated at other times, depending on the population of grazers and the efficiency of export. The CFE can provide 'ocean truth' for such estimates. There is uncertainty in the conversion of POCATN flux to POC flux, and we have no data on the conversion of PICPOL to PIC flux; however, the optical metrics of POC and PIC flux are robustly defined for the CFE and the resulting

15 calibration can only be achieved by at-sea 24-hour deployments of CFEs modified to collect samples; we hope to report on upcoming deployments soon.

images of settled particles clearly provide evidence of the food web mechanisms governing carbon export. Better

While CFE and BUOY-OSRs were not directly compared with simultaneously measured fluxes with PIT traps, thorium methods, large particle abundances sampled by large volume in-situ filtration and other methods, such a study should be undertaken in biologically dynamic waters and augmented by a framework of contemporaneous

remote sensing observations, water column biology and physics. To date, two CFEs have each completed missions lasting 40 days offshore of California and another 10 days in the waters in the subarctic Pacific (mission duration set by ship availability) in bad weather and sea states, and without biofouling issues; our lab tests show that a CFE can perform hourly observations of carbon export in the open ocean for 8 months (2 hourly frequency for 16 months, etc...). More work to enable on board image data analysis is

20

- 25 required to render the CFE fully autonomous. Near shore, horizontal currents can displace a CFE from the area of intended operation in several days time; thus longer time series observations are difficult without a means to reposition the CFE. As the cost of a CFE is equivalent to the cost of operating a research vessel for ~2-3 days, the job of tending CFEs for seasonal or year round studies in the coastal environment would need to be done using autonomous surface or underwater vehicles.
- In summary, we show that autonomous sampling technologies may be deployed to address many questions about the 30 workings of the biological pump both offshore and onshore, including whether or not its strength has changed over the past several decades and whether or not it will change in the future in the face of anthropogenic CO2 warming and acidification, and exploitation of living resources of the ocean.
- Data Availability. Data for plots, representative images used in this paper, and animations of image time series (at 1/2 35 resolution) are available as supplemental materials and are available from the author on request. Shipboard data sets: including

James Bishop 4/28/2016 11:40 PM Deleted: i

James Bishop 5/2/2016 4:21 PM Deleted: James Bishop 4/28/2016 11:42 PM Formatted: Subscript ames Bishop 5/2/2016 5:39 PM

James Bishop 4/28/2016 11:42 PM Deleted: W

meteorological, surface hydrography, CTD cast data, are available through the Go Ship (http://www.go-ship.org) archives or in the case of ADCP data, are available through the University of Hawaii.

**Acknowledgements.**

The CFE was developed in close collaboration with Mike McClune and Russ Davis of the Scripps Institution of Oceanography Instrument Development Group. OSR design and construction had critical contributions from the 5 Electrical and Mechanical engineering groups and machine shop facilities at Lawrence Berkeley National Laboratory. Alex Morales (LBNL) and 25 UC Berkeley undergraduates facilitated CFE and BUOY-OSR deployments and recoveries at sea. In particular, Gabrielle Weiss, Amelia Weiss, Andrew Bower, and Christina Hamilton contributed both at sea and in the laboratory; we also thank the resident technicians, crews and captains of the R/V New Horizon for assistance. Jules Hummon (U Hawaii) assisted with ADCP data reprocessing. Hannah 10 Bourne (UC Berkeley) cross-checked ADCP and BUOY-OSR current shear calculations. We thank Mati Kahru (SIO) for access to kilometre scale chlorophyll data for our study area. We thank the three anonymous reviewers for

[revised manuscript text omitted]

**Appendix A. Methods Detail**

10

40 A1.1 Optical Sedimentation Recorder on CFE.

The Optical Sedimentation Recorder (OSR; *Fig. A1*) is independently powered from the SOLO float to which it is attached and the two communicate via a hard-wired link. It thus does not impact the number of profiles that the float can make beyond additional telemetry needs.

16

James Bishop 4/28/2016 11:55 PM Formatted: Font:Italic James Bishop 4/28/2016 11:51 PM Formatted: Font:(Default) +Theme Body

| James Bishop 5/2/2016 3:32 PM |
|-------------------------------|
| Deleted: K.O.                 |
| James Bishop 5/2/2016 3:32 PM |
| Deleted: C.H.                 |
| James Bishop 5/2/2016 3:32 PM |
| Deleted: J.                   |
| James Bishop 5/2/2016 3:31 PM |
| Deleted: M.W.                 |
| James Bishop 5/2/2016 3:31 PM |
| Deleted: R.J.                 |
| James Bishop 5/2/2016 3:31 PM |
| Deleted: D.K.                 |
| James Bishop 5/2/2016 3:31 PM |
| Deleted: D.A.                 |

The SOLO dives to its target depth and signals the OSR to begin operation. During time at depth, particles settle into a baffled (5 cm thickx x 1 cm opening hex cell) high aspect polished titanium funnel (height 23.6 cm; slope 75°; 15.4 cm diameter opening) and settling column (height ~11 cm) before depositing on a 2.5 cm diameter glass sample stage. The baffle is typical to the dimensions used in surface tethered Particle Interceptor Traps (e.g. Martin et al.

5 1987, Collins et al., 2011, Stukel et al., 2013).

A downward focused 5 cm diameter white (color temperature 2700 K) light emitting diode (LED) source with bonded linear polarizer (Meadowlark Optics, Frederick, CO) is axially aligned with the funnel and provides transmitted light (or backlit) illumination of the sample.

An annular array of 12 white LEDs provides dark field illumination. This light source is integrated within the body of a 3-D printed form designed to provide both a hydrodynamic cleaning function and to secure a 27 mm diameter

10 a 3-D printed form designed to provide both a hydrodynamic cleaning function and to secure a 27 mm diameter dashed cross-hair reticle (Edmond Optics Inc., Barrington, NJ) to the pressure case window. A Seabird Inc. (Bellevue WA) pump outfitted with an isolation valve is used to flush the sample stage during cleaning.

A substage polarizer rotator is rotated either parallel or perpendicular to the axis of the downlight polarizer. In the case of CFE001, this polarizer rotator was housed within the instrument pressure case.

- 15 Under computer control, particles are imaged with a 5M pixel Sumix (SMX-11M5C; Oceanside, CA) colour imager in transmitted, transmitted – cross polarized, and dark field lighting modes which are designated as TRA, POL, DRK, respectively. Gain settings for each of the red, green, blue (RGB) channels were adjusted so that images of the downlight yield similar count value histogram averages (to within several %). The same gain settings were used for crosspolarized transmitted light illumination. For dark field, RGB channel balance was achieved by imaging salt crystals
- 20 dispersed across the sample stage. The camera lens was set to F16. Shutter timing was set for each of the three modes to prevent image saturation.

**A 1.2 Surface conditions, Hydrography, and Transmissometer data.**

25

35

Sea and wind conditions during the three study periods were benign. For example, In January and March 2013, winds monitored by the ship were less 5 m s-1 and averaged 2.5 m s-1. In May 2012, winds were less than 10 m s-1 and averaged 3 m s-1. Surface waves and swell were almost completely absent in January and March. In January 2013, the

weather was completely cloud free which contributed to excellent satellite imagery of the area. During all expeditions, a CTD/rosette system with a C-Star (WETLabs, Inc. Philomath, OR) transmissometer and Seapoint Inc. scattering sensor was repeatedly deployed to full water column depth. Approximately 20 casts were completed during each cruise. Only transmissometer data are reported here. The transmissometer cleaning,

30 calibration and data processing to particle beam attenuation coefficient ( $c_p$ ) follow protocols we have published (Bishop and Wood, 2008, Bishop, Lam and Wood, 2012). POC was calculated by multiplying particle beam attenuation coefficient by 27 (Bishop and Wood, 2008).

Figure A2 (A-C) shows  $c_p$  profiles for the three expeditions. Also shown is 0-20 m averaged POC from these casts. The time line for plot (D) is relative to the start of deployment of the CFE. Average and standard deviation values of 0-20 m averaged POC are quoted in the text.

**James Bishop 5/2/2016 4:31 PM Deleted: long James Bishop 5/2/2016 4:31 PM Deleted: wide**

The effects of internal waves with periods of 20-30 minutes were usually seen in CTD cast data. In March 2013, internal wave amplitudes of approximately 40 m were observed during CTD tow-yo experiments. During other expeditions, wave amplitudes were less than 20 m.

A 1.3 Spatial gradients of surface chlorophyll in the study area.

- 5 To examine the detailed spatial gradients of chlorophyll in our study area, we chose locations at 33.73°N 119.50°W and 33.69°N 119.58°W and retrieved chlorophyll data at 1 km spatial resolution within a 2 km search radius of the two locations. These positions were 9 km apart. To achieve greater temporal and spatial coverage than possible with the single MODIS/Aqua product, we analysed merged data from MODIS on the Aqua and Terra satellites and from the Visible Infrared Imaging Radiometer Suite (VIIRS) on the National Polar-orbiting Partnership Suomi satellite
- 10
   (Kahru et al. 2012; http://spg.ucsd.edu/Satellite\_Data/California\_Current/). Images at 1 km resolution are provided in

   Supplemental material.

Figure A3 shows time series of daily mean and standard deviation of chlorophyll plotted for both locations overthirty day periods encompassing the days that the ship was on station in the Santa Cruz Basin and during CFE

15 deployments. Also plotted is the relative difference in chlorophyll between the two locations. The two locations were almost always <20% different from one another. From this we conclude that a separation of 9 kilometers is not a significant cause of differences in our data. In other words, spatial gradients of biomass in our study area were small.

**A 1.4 BUOY-OSR configuration, January 2013.**

25

Figure A4 depicts the deployment configuration of the sample collecting BUOY-OSR used in this study. A SBE 19plus logging CTD (Sea Bird Electronics, Bellevue WA) recorded pressure, temperature and salinity at 5 second intervals. One OSR (Buoy instrument 84) was configured identically to CFE001. The funnel assembly of the second OSR (Buoy instrument 88) had a sample stage leak enabling strong water circulation through the funnel leading to particle loss as the system was pumped up and down due to wave action. Buoy instrument 84 is considered reliable as particles remained in place where they fell as they accumulated.

James Bishop 4/28/2016 10:39 PM Deleted:

James Bishop 4/28/2016 9:29 PM Formatted: Font:10 pt

James Bishop 4/28/2016 11:57 PM Formatted: Font:11 pt

James Bishop 4/28/2016 3:55 PM Formatted: Normal

James Bishop 4/28/2016 9:29 PM Deleted: A3

| Table 1. | Carbon | Flux  | Ex   | plorer  | and    | BUO | Y-O | SR   | positions |
|----------|--------|-------|------|---------|--------|------------|------------|------|-----------|
|          | 0      | arbon | Ebre | v Evolo | ror Si | urfacin    | a Doo      | itio | -         |

|            | curbonn            | Tax Explorer | Surrueing i Ositio | 15              |
|------------|--------------------|--------------|--------------------|-----------------|
| Dive       | Date               | Time (UTC)   | Longitude          | Latitude |
| 123        | 1-Jun-2012  | 3:15         | -119.475           | 33.738          |
| 124 | 1-Jun-2012  | 11:49 | -119.479           | 33.731   |
| 125        | 1-Jun-2012  | 21:14        | -119.484           | 33.736          |
| 126 | 2-Jun-2012  | 4:52         | -119.480           | 33.744          |
| 127        | 2-Jun-2012         | 13:28        | -119.498           | 33.744          |
| 128        | 2-Jun-2012         | 22:48        | -119.492           | 33.750          |
| 129 | 3-Jun-2012  | 6:24         | -119.493           | 33.737          |
| 130 | 3-Jun-2012  | 15:10 | -119.508           | 33.740          |
| 132 | 3-Jun-2012  | 20:45        | -119.512           | 33.735          |
|            |                    |              |                    |                 |
| 143        | 19-Jan-2013 | 8:16         | -119.479           | 33.744          |
| 144 | 19-Jan-2013 | 9:13  | -119.484           | 33.744          |
| 145        | 19-Jan-2013        | 16:31        | -119.514           | 33.744          |
| 146        | 19-Jan-2013 | 23:45        | -119.522           | 33.722          |
| 147 | 20-Jan-2013        | 8:21         | -119.575    | 33.714          |
| 148        | 20-Jan-2013        | 14:52        | -119.600           | 33.695          |
| 150 | 20-Jan-2013        | 20:46        | -119.605           | 33.667          |
|            |                    |              |                    |                 |
| 161 | 28-Mar-2013        | 10:55        | -119.537           | 33.709          |
| 162 | 28-Mar-2013        | 18:19 | -119.529    | 33.709          |
| 163 | 29-Mar-2013        | 4:08         | -119.555           | 33.705          |
| 164        | 29-Mar-2013        | 10:47        | -119.586           | 33.687          |
| 165 | 29-Mar-2013        | 18:05 | -119.576           | 33.693          |
| 166        | 30-Mar-2013        | 3:02         | -119.607           | 33.696          |
| 167 | 30-Mar-2013        | 9:46  | -119.614           | 33.673   |
|            |                    |              |                    |                 |

| BUC                | OY-OSR Positio | ns January 2013 |               |
|--------------------|----------------|-----------------|---------------|
| Date               | Time (UTC)     | Longitude       | Latitude      |
| 20-Jan-2013        | 9:00    | -119.490        | 33.740        |
| 20-Jan-2013        | 10:00          | -119.496        | 33.739        |
| 20-Jan-2013 | 11:00   | -119.501 | 33.739 |
| 20-Jan-2013        | 12:00          | -119.507        | 33.739        |
| 20-Jan-2013 | 13:00          | -119.513 | 33.740        |
| 20-Jan-2013 | 14:00          | -119.519 | 33.741        |
| 20-Jan-2013 | 15:00   | -119.524 | 33.742        |
| 20-Jan-2013        | 16:00          | -119.530        | 33.744        |
| 20-Jan-2013 | 17:00          | -119.534 | 33.745        |
| 20-Jan-2013 | 18:00          | -119.538 | 33.746        |
| 20-Jan-2013 | 19:00   | -119.542 | 33.747        |
| 20-Jan-2013        | 20:00          | -119.545        | 33.748        |
| 20-Jan-2013        | 21:00          | -119.546        | 33.749        |
| 20-Jan-2013 | 22:00          | -119.548        | 33.748        |
| 20-Jan-2013        | 23:00          | -119.550        | 33.747        |
|                    |                |                 |               |
|                    |                |                 |               |
|                    |                |                 |               |

| 1 | James Bishop 4/28/2016 10:37 PM |
|---|---------------------------------|
|   | Formatted Table                 |
|   | James Bishop 4/28/2016 10:37 PM |
|   | Formatted: Right: -0.07"        |
|   | James Bishop 4/28/2016 10:37 PM |
|   | Formatted: Right: -0.07"        |
|   | James Bishop 4/28/2016 10:37 PM |
|   | Formatted: Right: -0.07"        |
|   | James Bishop 4/28/2016 10:37 PM |
|   | Formatted: Right: -0.07"        |
|   | James Bishop 4/28/2016 10:37 PM |
|   | Formatted: Right: -0.07"        |
|   | James Bishop 4/28/2016 10:37 PM |
|   | Formatted: Right: -0.07"        |
|   | James Bishop 4/28/2016 10:37 PM |
|   | Formatted: Right: -0.07"        |
|   | James Bishop 4/28/2016 10:37 PM |
|   | Formatted: Right: -0.07"        |
|   | James Bishop 4/28/2016 10:37 PM |
|   | Formatted: Right: -0.07"        |
|   | James Bishop 4/28/2016 10:37 PM |
|   | Formatted: Right: -0.07"        |
|   | James Bishop 4/28/2016 10:37 PM |
|   | Formatted: Right: -0.07"        |
|   | James Bishop 4/28/2016 10:37 PM |
|   | Formatted: Right: -0.07"        |
|   | James Bishop 4/28/2016 10:37 PM |
|   | Formatted: Right: -0.07"        |
|   | James Bishop 4/28/2016 10:37 PM |
|   |                                 |
|   | James Bishop 4/28/2016 10:37 PM |
|   | Formatieu. Right0.07     |

**Comparison of the state of the sta**

| Diameter 1 | 144                | m CFE              | 320    | m CFE    | 507 m   | n CFE   | 2     | 37 m BU | OY 2 | 237 m B | UOY 3 |
|-----------------------|--------------------|--------------------|--------|----------|---------|---------|-------|---------|-----------------|---------|------------------|
| (µm)                  | 20130              | 120 11:11          | 201301 | 20 16:42 | 2013012 | 0 05:56 | 20    | 130120  | AVG             | 2013012 | 0 18:08          |
| >5000                 | 0.462 4 | 0.462 5 | 0.000  | 0.000    | 0.776   | 0.776   | 0.000 | 0.000   | ±0.000          | 0.000   | 0.000            |
| 2000-5000             | 0.441              | 0.903              | 0.978  | 0.978    | 0.150   | 0.926   | 0.065 | 0.065   | ±0.130          | 0.494   | 0.494            |
| 1000-2000             | 0.059              | 0.962              | 0.000  | 0.978    | 0.055   | 0.981   | 0.353 | 0.418   | ±0.090          | 0.169   | 0.662            |
| 500-1000              | 0.014              | 0.976              | 0.007  | 0.984    | 0.003   | 0.983   | 0.250 | 0.668   | ±0.070          | 0.153   | 0.815            |
| 200-500               | 0.015              | 0.992              | 0.005  | 0.989    | 0.011   | 0.994   | 0.245 | 0.913   | ±0.022          | 0.144   | 0.959            |
| 100-200               | 0.005              | 0.996              | 0.007  | 0.996    | 0.003   | 0.997   | 0.064 | 0.977   | ±0.006          | 0.030   | 0.989            |
| 50-100                | 0.003              | 0.999              | 0.003  | 0.999    | 0.002   | 0.999   | 0.017 | 0.994   | ±0.001          | 0.008   | 0.997            |
| 20-50                 | 0.001              | 1.000              | 0.001  | 1.000    | 0.001   | 1.000   | 0.006 | 1.000   | ±0.000          | 0.003   | 1.000            |

1. Diameter is Equivalent Circular Diameter of particle with the same area;

2. Averages (excluding 18:08 data) for BUOY data, third column (±) is s.d. of cumulative data.

3. Currents below 3 cm/sec for 60 min.

4. Volume Attenuance for indicated size interval.

5. Cumulative volume attenuance for particles greater than lower limit of size interval

**10**

5

**Table 3. Martin curve fit parameters**

| Time        | Zref (m)               | ⊕Zret            | b     | Std. Dev. 1 | P Zref | b    | Std. Dev. |  |
|-------------|------------------------|------------------|-------|------------------------|---------------|------|-----------|--|
| Jan. 2013   | 50                     | 252.0            | 0.858 | 34.7                   | 115.0         | 0.36 | 23.3      |  |
| Mar. 2013   | 20                     | 57.5             | 0.858 | 2.3                    | 11.4          | 0.22 | 0.4       |  |
| May 2012    | 25                     | 35.0             | 0.858 | 1.0                    | 3.8           | 0.17 | 0.6       |  |
| 1 Units are | in $m\Delta TN_{cm}^2$ | cm -2 |       |                        |               |      |           |  |

James Bishop 4/28/2016 10:36 PM Deleted: 1

| Formatted: Font:(Default) Arial, 9 pt, Not
James Bishop 4/29/2016 10:50 AM
Formatted Table
James Bishop 4/29/2016 10:44 AM
Formatted                                                                                                                      |
|-----------------------------------------------------------------------------------------------------------------------------------------------------------------------------------------------------------------------------------------------------------------------|
| James Bishop 4/29/2016 10:50 AM
Formatted Table
James Bishop 4/29/2016 10:44 AM
Formatted                                                                                                                                                                    |
| Formatted Table
James Bishop 4/29/2016 10:44 AM
Formatted[1]
James Bishop 4/29/2016 10:45 AM
Formatted[2]
James Bishop 4/29/2016 10:45 AM
Formatted[4]
James Bishop 4/29/2016 10:44 AM
James Bishop 4/29/2016 10:44 AM |
| James Bishop 4/29/2016 10:44 AM
Formatted[1]
James Bishop 4/29/2016 10:45 AM
Formatted[2]
James Bishop 4/29/2016 10:45 AM
Formatted[4]
James Bishop 4/29/2016 10:44 AM
Formatted[3]                                                              |
| Formatted[1]
James Bishop 4/29/2016 10:45 AM
Formatted[2]
James Bishop 4/29/2016 10:45 AM
Formatted[4]
James Bishop 4/29/2016 10:44 AM
James Bishop 4/29/2016 10:44 AM                                                       |
| James Bishop 4/29/2016 10:45 AM
Formatted                                                                                                                                                                                                                          |
| Formatted                                                                                                                                                                                                                                                             |
| James Bishop 4/29/2016 10:45 AM
Formatted                                                                                                                                                                                                                          |
| Formatted                                                                                                                                                                                                                                                             |
| James Bishop 4/29/2016 10:44 AM
James Bishop 4/29/2016 10:44 AM
Formatted                                                                                                                                                                |
| Formatted: Centered James Bishop 4/29/2016 10:44 AM Formatted                                                                                                                                                                                                         |
| James Bishop 4/29/2016 10:44 AM                                                                                                                                                                                                                                       |
| Formatted                                                                                                                                                                                                                                                             |
| Lamaa Diahan 4/20/2016 10:44 AM                                                                                                                                                                                                                                       |
| James Bishop 4/29/2016 10.44 AM                                                                                                                                                                                                                                       |
| Formatted                                                                                                                                                                                                                                                             |
| James Bishop 4/29/2016 10:44 AM                                                                                                                                                                                                                                       |
| Formatted: Centered                                                                                                                                                                                                                                                   |
| James Bishop 4/29/2016 10:44 AM                                                                                                                                                                                                                                       |
| Formatted: Font. (Default) Anal, 9 pt, Not                                                                                                                                                                                                                            |
| James Bisnop 4/29/2016 10:44 AM                                                                                                                                                                                                                                       |
| James Diebon 4/20/2016 10:44 AM                                                                                                                                                                                                                                       |
| Formatted: Font: (Default) Arial 9 nt Not                                                                                                                                                                                                                             |
| Jamos Rishon 4/20/2016 10:44 AM                                                                                                                                                                                                                                       |
| Formatted: Centered                                                                                                                                                                                                                                                   |
| James Rishon 4/20/2016 10:44 AM                                                                                                                                                                                                                                       |
| Formatted: Font: (Default) Arial 9 pt Not                                                                                                                                                                                                                             |
| James Bishon 4/29/2016 10:44 AM                                                                                                                                                                                                                                       |
| Formatted: Font: (Default) Arial, 9 pt. Not                                                                                                                                                                                                                           |
| James Bishon 4/29/2016 10:44 AM                                                                                                                                                                                                                                       |
| Formatted: Centered                                                                                                                                                                                                                                                   |
| James Bishop 4/29/2016 10:44 AM                                                                                                                                                                                                                                       |
| Formatted: Font: (Default) Arial, 9 pt, Not                                                                                                                                                                                                                           |
| James Bishop 4/29/2016 10:47 AM                                                                                                                                                                                                                                       |
| Formatted [6]                                                                                                                                                                                                                                                         |

**Figure Captions.**

Figure 1. Left. Carbon Flux Explorer showing the SOLO float and interfaced Optical Sedimentation Recorder. Top Right: Relief Map for Southern California Bight (Google Earth) centered on the Santa Cruz Basin (SCB) and our study area (shown in yellow). Also shown are locations of moored sediment trap deployments in the Santa Barbara Basin (SBB, Thunnel, 1998), and San Pedro Basin (SPB, Collins et al. 2011). Lower Right: Operations within study area. Surfacing positions of CFE001 during May 2012 (red triangles), Jan. 2013 (blue circles) and Mar. 2013 (Green box) - Also shown is track for the BUOY-OSR system (blue line) in Jan. 2013; separation of CFE and BUOY-OSR was approximately 9 km. Small symbols denote locations of CTD Casts. All motion was to the southwest. CFE and BUOY-OSR deployments took near 33.75°N 119.5°W. CTD profiles were taken within several km of CFE and BUOY-OSR positions.

[revised manuscript text omitted]

James Bishop 5/2/2016 10:14 PM Deleted: grey

James Bishop 4/29/2016 12:21 AN Formatted: Not Highlight

circles). Top to bottom graphs correspond to times of sampling. Dashed blue lines denote the periods that the ship was on station in the Santa Cruz Basin study area. The black horizontal line denotes duration of CFE deployments. Right panels show the fractional difference in mean chlorophyll between the two locations. There was no systematic difference between locations.

Figure A4. Photograph of the surface tethered BUOY-OSR as deployed in January 2013. Also shown is the mooring
 configuration used in this study. The seven small floats spaced 1 m apart are intended to minimize the effects of surface wave action on the up and down motion of the subsurface sediment trap. During the January 2013 deployment depth was 237 m (±0.25 m S.D.) – Maximum peak to trough vertical motion was 1 m.

**10**